# Taming Heavy-Tailed Losses in Adversarial Bandits and the Best-of-Both-Worlds Setting

**Duo Cheng**
Virginia Tech
duocheng@vt.edu

**Xingyu Zhou**
Wayne State University
xingyu.zhou@wayne.edu

**Bo Ji**
Virginia Tech
boji@vt.edu

## Abstract

In this paper, we study the multi-armed bandits problem in the best-of-both-worlds (BOBW) setting with heavy-tailed losses, where the losses can be negative and unbounded but have $(1 + v)$-th raw moments bounded by $u^{1+v}$ for some known $u > 0$ and $v \in (0, 1]$. Specifically, we consider the BOBW setting where the underlying environment can be either (oblivious) adversarial (i.e., the loss distribution can change arbitrarily over time) or stochastic (i.e., the loss distribution is fixed over time), which is unknown to the decision-maker a prior. We propose an algorithm and prove that it achieves a $T^{\frac{1}{1+v}}$-type worst-case (pseudo-)regret in the adversarial regime and a $\log T$-type gap-dependent regret in the stochastic regime, where $T$ is the time horizon. Compared to the state-of-the-art results, our algorithm offers stronger *high-probability* regret guarantees (vs. expected regret guarantees), and more importantly, relaxes a strong technical assumption on the loss distribution, which is generally hard to verify in practice. As a byproduct, relaxing this assumption leads to the first near-optimal regret result for heavy-tailed bandits with Huber contamination in the adversarial regime (vs. the easier stochastic regime studied in all previous works). Our result also implies a high-probability BOBW regret guarantee when the bounded true losses are protected with pure Local Differential Privacy (LDP), while the existing work ensures the (weaker) *approximate* LDP with the regret bounds in expectation only.

## 1 Introduction

Consider the multi-armed bandits (MAB) problem (Auer et al., 2002a,b), which is a useful framework for sequential decision-making under uncertainty and can be formulated as repeated interactions between the environment and a (learning) algorithm. In each of the total $T$ rounds indexed by $t$, the algorithm plays an action $a_t$ from a fixed set of $K$ actions (assuming $K \leqslant T$). Simultaneously, the environment determines the losses of all actions $\ell_t \in \mathbb{R}^K$. The algorithm observes and suffers the loss associated with $a_t$ (denoted by $\ell_{t,a_t}$). The goal of the algorithm is to minimize the cumulative loss over $T$ rounds, or equivalently, to minimize the *regret*, defined as the difference between its cumulative loss and that incurred by playing the best-fixed action (in hindsight) all the time. Without observing the losses of the other actions, the algorithm must "infer" the optimal action through interactions on the fly, facing the well-known trade-off between exploration and exploitation.

Depending on how the losses are determined, MAB problem is typically studied in two regimes: 1) the stochastic regime, where the loss of each action is drawn from a fixed (but unknown) distribution over time; 2) the adversarial regime, where the losses can be arbitrary (within some known class). Typically, the losses are assumed to have a support on a bounded interval (e.g., $[0, 1]$), and the fundamental limits have been well understood: 1) in the stochastic regime, a number of optimism-

38th Conference on Neural Information Processing Systems (NeurIPS 2024).

based algorithms achieve the $\Theta(\sqrt{KT})$[1] worst-case (pseudo-)regret and $\Theta(\sum_{i:\Delta_i>0}(1/\Delta_i)\log T)$ gap-dependent regret, where $\Delta_i$ (formally defined in Section 2) is the sub-optimality gap between action $i$ and the optimal action (Lai & Robbins, 1985; Auer et al., 2002a; Agrawal & Goyal, 2017); 2) in the adversarial regime, the $\Theta(\sqrt{KT})$ worst-case regret can be achieved by classic Online Learning algorithms, such as Follow-the-Regularized-Leader (FTRL), Online-Mirror-Descent (OMD), and Follow-the-Perturbed-Leader (FTPL) (Audibert & Bubeck, 2009; Lee et al., 2024).

Despite these progresses, the optimal algorithms in the two regimes fall into different frameworks (i.e., optimism-based algorithms for the stochastic regime vs. Online Learning algorithms for the adversarial regime). Moreover, while the former ones enjoy a logarithmic regret (i.e., $O(\log T)$) in the stochastic regime, even their $\widetilde{O}(\sqrt{KT})$ worst-case guarantees no longer hold when the environment deviates from the stochastic regime with empirical evidence provided in Zimmert & Seldin (2021). On the other hand, while Online Learning algorithms always preserve worst-case optimality, they could be too "conservative" to enjoy logarithmic regrets in stochastic environments.

These performance discrepancies motivated the study of the *Best-of-Both-Worlds (BOBW)* setting. That is, one single algorithm preserves the optimal worst-case regret in the adversarial regime and adapts to the stochastic regime with a logarithmic regret, *without knowing the type of the regime* in advance. Bubeck & Slivkins (2012) initiated the study by proposing a *detect-switch* framework, which preserves the optimal $\widetilde{O}(\sqrt{KT})$ regret in the adversarial regime and enjoys $O\left((\log T)^2 K/\Delta\right)$ regret in the stochastic regime, where $\Delta := \min_{i:\Delta_i>0}\Delta_i$ is the smallest sub-optimality gap. Under this framework, Auer & Chiang (2016) improved the gap-dependent term from $K/\Delta$ to $\sum_{i:\Delta_i>0}(1/\Delta_i)$.

Another line of work showed that without explicit detect-switch, OMD, originally designed for the adversarial regime, can automatically adapt to stochastic environments with a logarithmic regret (Wei & Luo, 2018; Zimmert & Seldin, 2021). In particular, Zimmert & Seldin (2021) achieved optimal regrets in the BOBW setting. Following these works, the power of Online Learning algorithms towards BOBW has been extended to various setups (Ito, 2021; Ito et al., 2022; Kong et al., 2023; Ito & Takemura, 2023; Dann et al., 2023; Jin et al., 2024; Lee et al., 2024; Tsuchiya et al., 2024).

While the aforementioned works require bounded losses, real-world data from application domains such as finance (Cont, 2001) and imaging (Hamza & Krim, 2001) often exhibits a heavy-tailed distribution. Intuitively, heavy-tailed losses make learning problems harder (compared to bounded losses) as they are "noisier" and "less informative" (Zhang et al., 2020). When losses are unbounded but have $(1+v)$-th raw moment bounded by $u^{1+v}$ for some known $u > 0$ and $v \in (0,1]$, Bubeck et al. (2013) showed $\widetilde{\Theta}(uK^{\frac{v}{1+v}}T^{\frac{1}{1+v}})$ worst-case regret and $\Theta\left(\sum_{i:\Delta_i>0}(u^{1+1/v}/\Delta_i)^{1/v}\log T\right)$ gap-dependent regret[2] in the stochastic regime by integrating robust mean estimators (e.g., trimmed mean and median-of-means) into optimism-based algorithms.

To address heavy-tailed losses in the adversarial regime and BOBW setting, Huang et al. (2022) showed that with calibrated adaptive loss trimming thresholds, FTRL with Tsallis entropy regularizer (Audibert & Bubeck, 2009) enjoys the optimal BOBW expected regrets under the "truncated non-negative losses" assumption (see Assumption 2). Without this strong assumption, it is unclear whether the near-optimal worst-case regret can still be achieved in the adversarial regime, let alone the BOBW setting. The key technical challenge here is that heavy-tailed losses can be both negative and unbounded, which is known to break the regret guarantees of the Online Learning algorithms. Extensive discussions and insights for our solution are provided in Section 3.

Given this challenge, the gap raises an interesting question: *In heavy-tailed MAB, are there any fundamental barriers to the worst-case optimality in the adversarial regime and the BOBW guarantees?*

We offer a positive answer to the above question, which implies that there is no such barrier. Our main contributions are as follows:

- In the adversarial regime, we propose an OMD-based algorithm achieving the (near-)optimal $\widetilde{O}(uK^{\frac{v}{1+v}}T^{\frac{1}{1+v}})$ pseudo-regret *with high probability*. Our approach relaxes the undesired "truncated non-negative losses" assumption, which is needed in the state-of-the-art results

---

[1]We use the standard notations $O(\cdot)$, $\Omega(\cdot)$, and $\Theta(\cdot)$. Those with tilde hide poly-log terms in $T$ and $K$. We use $\log(\cdot)$ to denote the natural logarithm ($\log_e(\cdot)$), otherwise the base $x$ is explicitly specified as $\log_x(\cdot)$.

[2]In the stochastic heavy-tailed MAB, the gap-dependent logarithmic bounds omit some $O(\exp(1/v))$ factors; this is also the case in all previous works (Bubeck et al., 2013; Tao et al., 2022; Huang et al., 2022).

Table 1: Comparison with related results in heavy-tailed MABs. In this table, column "$(u,v)$-free" indicates whether the algorithm ensures the stated guarantee for unknown $(u,v)$; "Assumption-2-free" indicates whether the stated guarantee holds without Assumption 2; "Sto." and "Adv." are abbreviations for "Stochastic" and "Adversarial", respectively; "High-prob." indicates whether the stated expected bound is implied by some (stronger) high-probability bound.

| Algorithm | $(u,v)$-free | Assumption-2-free | Regime | Expected Regret | High-prob. |
|:---:|:---:|:---:|:---:|:---:|:---:|
| Lower bounds (Bubeck et al., 2013) | N/A | N/A | Sto./Adv. | $\Omega(uK^{\frac{v}{1+v}}T^{\frac{1}{1+v}})$ | N/A |
| | | | Sto. | $\Omega\left(\sum_{i:\Delta_i>0}(\frac{u^{1+\frac{1}{v}}}{\Delta_i})^{\frac{1}{v}}\log T\right)$ | |
| RobustUCB (Bubeck et al., 2013) | ✗ | ✓ | Sto. | $O(uK^{\frac{v}{1+v}}T^{\frac{1}{1+v}}\log T)$ | ✓ |
| | | | | $O\left(\sum_{i:\Delta_i>0}(\frac{u^{1+\frac{1}{v}}}{\Delta_i})^{\frac{1}{v}}\log T\right)$ | |
| HTINF (Huang et al., 2022) | ✗ | ✗ | Adv. | $O(uK^{\frac{v}{1+v}}T^{\frac{1}{1+v}})$ | ✗ |
| | | | Sto. | $O\left(\sum_{i\neq i^*}(\frac{u^{1+\frac{1}{v}}}{\Delta_i})^{\frac{1}{v}}\log T\right)$ | |
| AdaTINF (Huang et al., 2022) | ✓ | ✗ | Adv. | $O(uK^{\frac{v}{1+v}}T^{\frac{1}{1+v}}+\sqrt{KT})$ | ✗ |
| AdaR-UCB (Genalti et al., 2024) | ✓ | ✗ | Sto. | $O\left(uK^{\frac{v}{1+v}}T^{\frac{1}{1+v}}+\sum_{i:\Delta_i>0}\frac{\Delta_i}{\mathbb{P}_{\ell\sim P_i}(\ell\neq 0)}\log T\right)$ | ✗ |
| | | | | $O\left(\sum_{i:\Delta_i>0}\left((\frac{u^{1+\frac{1}{v}}}{\Delta_i})^{\frac{1}{v}}+\frac{\Delta_i}{\mathbb{P}_{\ell\sim P_i}(\ell\neq 0)}\right)\log T\right)$ | |
| uniINF (Chen et al., 2024) | ✓ | ✗ | Adv. | $O\left(uK^{\frac{v}{1+v}}T^{\frac{1}{1+v}}((\log T)^{1.5}+\log u)\right)$ | ✗ |
| | | | Sto. | $O\left(K(\frac{u^{1+\frac{1}{v}}}{\Delta})^{\frac{1}{v}}\log(T)\log\frac{u^{1+v}}{\Delta}\right)$ | |
| OMD-LB-HT (**Algorithm 1**) | ✗ | ✓ | Adv. | $O(uK^{\frac{v}{1+v}}T^{\frac{1}{1+v}}(\log T)^3)$ | ✓ |
| SAO-HT (**Algorithm 2**) | ✗ | ✓ | Adv. | $O(uK^{\frac{v}{1+v}}T^{\frac{1}{1+v}}\log(K)(\log T)^4)$ | ✓ |
| | | | Sto. | $O\left(K\log(K)(\frac{u^{1+\frac{1}{v}}}{\Delta})^{\frac{1}{v}}(\log T)^4\right)$ | |

even for the weaker expected regret guarantee (Huang et al., 2022). Relaxing it also allows us to obtain the near-optimal worst-case guarantee against the Huber contamination, which, to our best knowledge, was only studied for the stochastic regime in the literature. This suggests broader implications of our approach.

- On top of the above advance in the adversarial regime, by leveraging the detect-switch framework, we further extend the (near-)optimal regret guarantees to the BOBW setting. Specifically, our algorithm preserves the optimal $\widetilde{O}(uK^{\frac{v}{1+v}}T^{\frac{1}{1+v}})$ regret in the adversarial regime and enjoys $O(K(u^{1+1/v}/\Delta)^{1/v}\log(K)(\log T)^4)$ gap-dependent logarithmic regret in the stochastic regime, both *with high probability*, which implies that *there is no fundamental barrier to achieving the BOBW guarantees when the loss distributions are heavy-tailed*. This result also immediately imply the first high-probability BOBW regret guarantees with *pure* Local Differential Privacy (LDP) protection on the true losses, while the existing result ensures the weaker *approximate* LDP protection with expected regret guarantees only.

- Technique-wise, we leverage the inherent stronger stability of log-barrier to relax a strong technical assumption made in previous works and utilize the *increasing-learning-rates* trick (Lee et al., 2020) to obtain the stronger high-probability guarantee in the adversarial regime. Moreover, we adapt the detect-switch framework by Bubeck & Slivkins (2012), originally designed for BOBW in the bounded-loss case, to the heavy-tailed setup. The adaptation introduces non-trivial challenges in the analysis due to the history-dependent trimmed estimator. In particular, to obtain the desired concentration rate in the adversarial regime, the proof does not follow its existing counterpart in the stochastic regime. Beyond addressing these challenges, we identify a novel use (i.e., handling history-dependent trimming in martingale concentrations) of an adaptive variant of Freedman's inequality (originally proposed by Lee et al. (2020) and improved by Zimmert & Lattimore (2022) for high-probability regret in adversarial bandits), which may be of independent interest.

We refer the readers to Table 1 for a summary of the most relevant results, in which we also include adaptive results on the case when $u,v$ are unknown. We also present a comprehensive discussion about related work, which is deferred to Appendix A due to the page limit.

## 2 Problem Setup

In this section, we formally introduce the problem setup and define needed notations. To formulate heavy-tailed losses, we let $\ell_{t,i}$ denote the loss of action $i$ in round $t$, which is drawn from distribution $P_{t,i}$ satisfying the following assumption:

**Assumption 1.** The $(1 + v)$-th (raw) moments of losses (which have potentially unbounded support in $\mathbb{R}$) are bounded by $u^{1+v}$ for some constants $u > 0$ and $v \in (0, 1]$, i.e., $\mathbb{E}_{\ell_{t,i} \sim P_{t,i}} \left[ |\ell_{t,i}|^{1+v} \right] \leqslant u^{1+v}, \forall t \in [T], i \in [K]$, where $[n]$ denotes set $\{1, \ldots, n\}$ for any integer $n \geqslant 1$.

In the heavy-tailed MAB problem, the (learning) algorithm and environment perform the following interactions repeatedly in round $t = 1, \ldots, T$:

1. The algorithm samples action $a_t$ from $[K]$ via $a_t \sim w_t := (w_{t,1}, \ldots, w_{t,K})$ in the probability simplex $\Omega := \{x \in [0,1]^K \mid \sum_{i=1}^K x(i) = 1\}$, i.e., action $i \in [K]$ is sampled with probability $w_{t,i}$. The environment draws loss $\ell_{t,i} \sim P_{t,i}$ for every action $i \in [K]$.

2. The algorithm observes $\ell_{t,a_t}$ only; the losses of all other actions are unrevealed.

3. The algorithm determines $w_{t+1}$ based on the history $(w_1, a_1, \ell_{1,a_1}, \ldots, w_t, a_t, \ell_{t,a_t})$.

*Remark* 1. All of our algorithms and their regret bounds allow moment order $(1 + v) \in (1, 2]$ only. That is, one may not obtain any regret guarantee by running our algorithms with $v > 1$. If the losses have higher-order moments ($v > 1$), one can simply run our algorithms with $v = 1$ and obtaining the corresponding regret bounds (since bounded higher-order moments imply lower-order ones). Note that Bubeck et al. (2013) showed that for all $v \geqslant 1$, the lower bounds in terms of both worst-case regret and gap-dependent regret are the same as the case of $v = 1$.

We assume that heavy tail parameters $u$ and $v$ and time horizon $T$ are known to the algorithm a priori.[3] The objective of the algorithm is to minimize the *pseudo-regret* $R_T$, defined as

$$R_T := \sum_{t=1}^T \left( \mu_{t,a_t} - \mu_{t,i^*} \right), \tag{1}$$

where $\mu_{t,i} := \mathbb{E}_{\ell_{t,i} \sim P_{t,i}} [\ell_{t,i}]$ denotes the mean loss of action $i \in [K]$ in round $t \in [T]$, and $i^* \in \operatorname{argmin}_{i \in [K]} \sum_{t=1}^T \mu_{t,i}$ denotes any best-fixed action in hindsight.

Depending on how loss distributions are determined, we further define the following two regimes:

- **Stochastic regime:** For every action $i \in [K]$, the loss distributions are identical in all rounds. That is, we have $P_{1,i} = \cdots = P_{T,i} = P(i)$, implying $\mu_{1,i} = \cdots = \mu_{T,i} = \mu(i)$ and $i^* \in \operatorname{argmin}_{i \in [K]} \mu(i)$. We also define $\Delta_i := \mu(i) - \mu(i^*)$ and $\Delta := \min_{i:\Delta_i > 0} \Delta_i$.

- **(Oblivious) Adversarial regime:** All loss distributions are chosen arbitrarily (by some adversary, with the full knowledge of the algorithm) before the interaction begins. Our regret definition and adversarial model are also considered in Huang et al. (2022).

*Remark* 2. It is not hard to see that the stochastic regime is a special (and "easy") case of the adversarial regime. There are two differences between our heavy-tailed setup and the bounded-loss setup in the adversarial bandits literature: 1) Typically in the (bounded-loss) adversarial regime, the losses are considered to be deterministic rather than randomized (and this difference also leads to some new challenges in the analysis when we adopt the detect-switch framework from the bounded case (Bubeck & Slivkins, 2012) as we will show later in Section 4.2), and 2) a stronger notion of regret, defined as $\overline{R}_T := \sum_{t=1}^T \ell_{t,a_t} - \min_{i \in [K]} \sum_{t=1}^T \ell_{t,i}$, is considered; this is stronger than pseudo-regret since $\mathbb{E}[\overline{R}_T] \geqslant \mathbb{E}[R_T]$. However, when adapted to the heavy-tailed case, it is natural to still consider randomized losses and pseudo-regret as in the stochastic regime: The "easiness" of the regime boils down to how heavy-tailed distributions are chosen. Moreover, a low *stronger regret* depends not only on playing good actions (with low loss means), but also on the realization of potentially unbounded losses, which loses the standard meaning in evaluating the algorithm. Therefore, when switching from stochastic to adversarial regime in heavy-tailed bandits, we keep pseudo-regret as the metric.[4]

---

[3]We summarize (negative) results from the literature when $u, v$ are unknown in Appendix A.

[4]Starting from this point, we always refer to "pseudo-regret" ($R_T$ in Eq. (1)) as "regret" in both regimes.

Our goal is to design one single algorithm that can achieve the near-optimal $\widetilde{O}(uK^{\frac{v}{1+v}}T^{\frac{1}{1+v}})$ worst-case regret in the adversarial regime and enjoys $\log T$-type regret when the regime is stochastic, without being informed of the regime type in advance.

A closely related work by Huang et al. (2022) studied the same BOBW setup (i.e., achieving BOBW guarantee when $u, v$ are known). They proposed an FTRL-based algorithm with Tsallis entropy regularizer and carefully-chosen history-dependent trimming threshold for loss magnitude control and showed $O\left(\sum_{i:\Delta_i > 0}(u^{1+1/v}/\Delta_i)^{1/v}\log T\right)$ regret in the stochastic regime and $O(uK^{\frac{v}{1+v}}T^{\frac{1}{1+v}})$ regret in the adversarial regime, both of which are in *expectation* and optimal. However, their regret guarantees rely heavily on a strong technical assumption:

**Assumption 2** (Truncated non-negative losses (Huang et al., 2022)). Given any fixed $M > 0$, the loss distributions of the optimal action $i^*$ satisfy $\mathbb{E}_{\ell_{t,i^*} \sim P_{t,i^*}}[\ell_{t,i^*} \cdot \mathbb{I}\{|\ell_{t,i^*}| > M\}] \geqslant 0, \forall t \in [T]$.

In the following sections, we first show that by resorting to the log-barrier regularizer, we obtain the near-optimal regret bound in the adversarial regime without Assumption 2 and naturally extend it to the stronger high-probability guarantee (Section 3). On top of that, we further adapt the detect-switch framework in Bubeck & Slivkins (2012) and obtain high-probability bounds in BOBW (Section 4).

**Additional Notations.** For any round $t$ and action $i$, we let $I_{t,i} := \mathbb{I}\{a_t = i\}$ denote whether action $i$ is pulled in round $t$ and $N_{t,i} := \sum_{s=1}^{t} I_{s,i}$ denote the number of times when action $i$ is pulled before the end of round $t$. We use $\widehat{\ell}_{t,i} := \ell_{t,i}I_{t,i}\mathbb{I}\{|\ell_{t,i}| \leqslant M_{t,i}\}/w_{t,i}$ to denote the (trimmed) IW estimate with respect to some threshold $M_{t,i}$ and $\widehat{\mu}_{t,i} := \sum_{s=1}^{t} \ell_{s,i}I_{s,i}\mathbb{I}\{|\ell_{s,i}| \leqslant B_{s,i}\}/N_{t,i}$ to denote the (trimmed) empirical average with respect to some threshold $B_{t,i}$.[5] We also use $\mu'_{t,i} := \mathbb{E}_{\ell_{t,i} \sim P_{t,i}}[\ell_{t,i}\mathbb{I}\{|\ell_{t,i}| \leqslant M_{t,i}\}]$ to denote the mean of the trimmed loss and $\widehat{L}_{t,i} := \sum_{s=1}^{t} \widehat{\ell}_{s,i}$ to denote the cumulative IW estimate.

## 3  High-probability Near-optimal Regret in the Adversarial Regime

This section is dedicated to high-probability regret in the adversarial regime. We first present detailed discussions on why Assumption 2 is needed in Huang et al. (2022) and how we get rid of it (Section 3.1), followed by the description of our algorithm design and regret guarantee (Section 3.2).

### 3.1  Technical Challenges and Insights

The main technical challenge we encounter comes from the potentially-unbounded negative losses (even for regret bounds *in expectation* only). We illustrate this challenge using the previous work of Huang et al. (2022) as an example. Their algorithm is running standard FTRL with $(1 + v)^{-1}$-Tsallis entropy regularizer over the trimmed loss estimate sequence $\widehat{\ell}_1, \ldots, \widehat{\ell}_T$ with respect to some trimming threshold $(M_{t,i})_{t \in [T], i \in [K]}$ to be introduced later in this subsection.

To bound the expected regret, Huang et al. (2022) first rewrite it as

$$
\begin{aligned}
\mathbb{E}\left[R_T\right] &= \mathbb{E}\left[\sum_{t=1}^{T}\langle w_t - y^{i^*}, \mu_t - \mu'_t\rangle\right] + \mathbb{E}\left[\sum_{t=1}^{T}\langle w_t - y^{i^*}, \mu'_t\rangle\right] \\
&= \underbrace{\mathbb{E}\left[\sum_{t=1}^{T}\sum_{i=1}^{K} w_{t,i}(\mu_{t,i} - \mu'_{t,i})\right]}_{\text{Part I}} + \underbrace{\mathbb{E}\left[\sum_{t=1}^{T}(\mu'_{t,i^*} - \mu_{t,i^*})\right]}_{\text{Part II}} + \underbrace{\mathbb{E}\left[\sum_{t=1}^{T}\langle w_t - y^{i^*}, \widehat{\ell}_t\rangle\right]}_{\text{Part III}}, \quad (2)
\end{aligned}
$$

where $y^i$ is the $K$-dim vector such that the $i$-th entry is one and all the others are zero.

The analyses begin with Part III. The desired upper bound on Part III holds only under the well-known "stability condition" associated with Tsallis entropy (Jin et al., 2024, Lemma C.5.3):

$$
\eta_{t,i}(w_{t,i})^{1-\frac{1}{1+v}}\widehat{\ell}_{t,i} = \eta_{t,i}(w_{t,i})^{1-\frac{1}{1+v}}(w_{t,i})^{-1}\ell_{t,i}I_{t,i}\mathbb{I}\{|\ell_{t,i}| \leqslant M_{t,i}\} \geqslant -C(u,v), \quad (3)
$$

where constant $C(u,v) > 0$ depends on $u$ and $v$ only and the learning rate $\eta_{t,i}$ is chosen as $u^{-1}t^{\frac{-1}{1+v}}$. While this condition is trivially satisfied when losses are non-negative, due to potentially-unbounded

---

[5]The values of thresholds may change from time to time in this work but will be made clear given the context.

**Algorithm 1** OMD with log-barrier and increasing learning rates for heavy-tailed MABs (`OMD-LB-HT`)
1: **Input:** failure probability $\zeta$, initial learning rate $\eta$, trimming threshold $\{M_{t,i}\}_{t\in[T],i\in[K]}$
2: **Define:** learning rate increase factor $\kappa = e^{1/\log T}$; log-barrier regularizer $\phi_t(x) = -\sum_{i=1}^K \log(x(i))/\eta_{t,i}$; Bregman divergence $D_{\phi_t}(x,x') = \sum_{i=1}^d (x(i)/x'(i) - \log(x(i)/x'(i)) - 1)/\eta_{t,i}$; simplex truncation parameter $\lambda = T^{\frac{-v}{1+v}} K^{\frac{-1}{1+v}}$; truncated probability simplex $\Omega' := \{x \in \Omega : x(i) \geqslant \lambda/K, \forall i \in [K]\}$
3: **Initialization:** For every action $i \in [K]$, define $w_{1,i} = 1/K, \rho_{1,i} = 2K, \eta_{1,i} = \eta$
4: **for** $t = 1 : T$ **do**
5:     Take action $a_t$ sampled from $w_t$ and observe $\ell_{t,a_t}$
6:     Construct loss estimate $\widehat{\ell}_{t,i} = \mathbb{I}\{a_t = i\}\mathbb{I}\{|\ell_{t,i}| \leqslant M_{t,i}\}\ell_{t,i}/w_{t,i}, \forall i \in [K]$
7:     Calculate $w_{t+1} = \operatorname{argmin}_{x\in\Omega'}\left(\langle x, \widehat{\ell}_t\rangle + D_{\psi_t}(x, w_t)\right)$
8:     **for** $i \in [K]$ **do**
9:         **if** $1/w_{t+1,i} > \rho_{t,i}$, **then** $\rho_{t+1,i} = 2/w_{t+1,i}, \eta_{t+1,i} = \eta_{t,i}\kappa$
10:        **else** $\rho_{t+1,i} = \rho_{t,i}, \eta_{t+1,i} = \eta_{t,i}$
11:    **end for**
12: **end for**

negative losses, threshold $M_{t,i}$ here is chosen to be $\Theta((t \cdot w_{t,i})^{\frac{1}{1+v}})$ (in particular, to fully "cancel" the $(w_{t,i})^{-1}$ from $\widehat{\ell}_{t,i}$). Otherwise, negative losses break this condition whenever $w_{t,a_t}$ is very small.

While such a threshold suffices to bound Part I and Part III with worst-case optimality (and even in the BOBW setting), applying the analysis for Part I to Part II leads to an upper bound of form $\sum_{t=1}^T (w_{t,i^*})^{\frac{-v}{1+v}}$ on Part II, which is potentially unbounded since $w_{t,i^*}$ can be very close to zero. However, with the help of Assumption 2, Part II itself is non-positive and hence can be ignored.

**Summary.** The key issue above is that, due to unbounded and negative losses, Eq. (3) results in a threshold $M_{t,i}$ which scales with $(w_{t,i})^{\frac{1}{1+v}}$ (in particular, for $i = i^*$), rendering Part II hard to bound. *Remark* 3. This issue may not be fixed by simply shifting all loss estimates to become positive and satisfy Eq. (3). Roughly speaking, the reason is that obtaining desired regret bounds relies heavily on the well-bounded $(1 + v)$-th moment of the losses (before shifting). However, to ensure positive losses, the needed shift is too "significant" and breaks the "nice" moment conditions.

To handle this issue and relax Assumption 2, we resort to the log-barrier regularizer, which offers the standard regret guarantee with the following stability condition (Agarwal et al., 2017):

$$\eta_{t,i}w_{t,i}\widehat{\ell}_{t,i} \geqslant -0.5. \tag{4}$$

Importantly, this condition itself already provides a $w_{t,i}$ (rather than the previous $(w_{t,i})^{1-\frac{1}{1+v}}$ in Eq. (3)) to "cancel" the $(w_{t,i})^{-1}$ from $\widehat{\ell}_{t,i}$, meaning that we do not need any additional $w_{t,i}$ contributed from threshold $M_{t,i}$. As a result, we can choose a different $M_{t,i}$ that scales with $K^{\frac{-1}{1+v}}$ rather than the previous $(w_{t,i})^{\frac{1}{1+v}}$ such that all three parts are bounded by $\widetilde{O}(uK^{\frac{v}{1+v}}T^{\frac{1}{1+v}})$ *without Assumption 2*.

We further adopt the increasing-learning-rates trick (Lee et al., 2020) together with an adaptive variant of Freedman's inequality (stated in Lemma 12) to obtain the stronger *high-probability* regret.

### 3.2 Algorithm Overview and Regret Guarantee

We now discuss the algorithm design and present the full pseudo-code in Algorithm 1. At a high level, our algorithm is running standard OMD over loss sequence $\widehat{\ell}_1, \ldots, \widehat{\ell}_T$ with respect to some fixed threshold $M_{t,i} = u \cdot (T/K)^{\frac{1}{1+v}}$. The key ingredients (to obtain high-probability regret) are the special learning rate schedule and probability simplex truncation (in OMD update) (Wei & Luo, 2018; Lee et al., 2020), which we briefly introduce below.

**Increasing Learning Rates.** We use vectors $\{\rho_t\}_{t\in[T]}$ to keep track of smallest sampling probabilities throughout the interaction. To be more specific, for every action $i$, if $w_{t+1,i}$ is so small that $1/w_{t+1,i} > \rho_{t,i}$ holds, we increase the learning rate by a factor of $\kappa > 1$ and set $\rho_{t+1,i} = 2/w_{t+1,i}$ (Line 9). Otherwise, we keep the learning rate unchanged and set $\rho_{t+1,i} = \rho_{t,i}$ (Line 10). In the analysis, such

a schedule introduces some negative term in the upper bound, which cancels out a positive term that could potentially be very large due to the variance of loss estimates (Lee et al., 2020).

**Probability Simplex Truncation.** We perform the OMD update over the *truncated* probability simplex $\Omega' := \{x \in \Omega : x(i) \geqslant \lambda/K, \forall i \in [K]\}$ (where $\lambda$ controls the degree of truncation), of which the purpose is to ensure that $w_{t,i}$ is always at least $\lambda/K$ and hence to control the variance of the loss estimates. The value of $\lambda$ here is $T^{\frac{-v}{1+v}}K^{\frac{-1}{1+v}}$ adapted to the heavy-tailed case and differs from the original value $1/T$ for the bounded-loss case in Lee et al. (2020).

The regret guarantee of Algorithm 1 is formally stated below with full proofs presented in Appendix B.

**Theorem 1.** *In the adversarial regime, for any failure probability $\zeta$, by choosing initial learning rate $\eta = (40M \log(T) \log(8KT/\zeta))^{-1}$ and trimming threshold $M_{t,i} = u \cdot (T/K)^{\frac{1}{1+v}}$, Algorithm 1 ensures that with probability at least $1 - \zeta$, $R_T = O(uK^{\frac{v}{1+v}}T^{\frac{1}{1+v}}(\log T)^2 \log(T/\zeta))$. By further choosing $\zeta = 1/T$, Algorithm 1 ensures that $\mathbb{E}[R_T] = O(uK^{\frac{v}{1+v}}T^{\frac{1}{1+v}}(\log T)^3)$.*

*Remark* 4. Removing Assumption 2 is crucial to obtain *the first and near-optimal* worst-case regret in heavy-tailed MAB when the feedback could be contaminated by the Huber model *in the adversarial regime*, in contrast to all previous works that study the (easier) stochastic regime (Guan et al., 2020; Agrawal et al., 2024; Wu et al., 2024). We provide all details in Appendix D.

## 4 High-probability Regrets in the Best-of-Both-Worlds Setting

With Algorithm 1 achieving high-probability optimal regret in the adversarial regime, we further leverage the detect-switch framework proposed by Bubeck & Slivkins (2012) named SAO to achieve high-probability bounds in the BOBW setting. We first present the BOBW guarantee, followed by an algorithm overview and analysis sketch. The complete proofs are provided in Appendix C.

**Theorem 2.** *In the adversarial regime, for any failure probability $\zeta$, by choosing constant $c_1 = 6$, Algorithm 2 ensures that with probability at least $1 - \zeta$,*

$$R_T = O\left(uK^{\frac{v}{1+v}}T^{\frac{1}{1+v}}\log(K)(\log T)^2(\log(\beta/\zeta))^2\right),$$

*which (by further choosing $\zeta = 1/T$) implies that $\mathbb{E}[R_T] = O\left(uK^{\frac{v}{1+v}}T^{\frac{1}{1+v}}\log(K)(\log T)^4\right)$. In the stochastic regime, Algorithm 2 ensures that with probability at least $1 - \zeta$,*

$$R_T = O\left(K\log(K)(u^{1+1/v}/\Delta)^{1/v}\log(T)\left(\log(T/\zeta)\right)^3\right),$$

*which implies that $\mathbb{E}[R_T] = O\left(K\log(K)(u^{1+1/v}/\Delta)^{1/v}(\log T)^4\right)$.*

*Remark* 5. High-probability bounds also are powerful tools to handle adaptive adversaries (who determine the current distribution $P_{t,i}$ based on past actions $a_1, \ldots, a_{t-1}$). Following the literature (Audibert & Bubeck, 2010; Lee et al., 2020; Zimmert & Lattimore, 2022), based on our high-probability bounds against oblivious adversaries, one may further derive both high-probability and expected regret bounds against adaptive adversaries, which we leave as future investigation.

*Remark* 6. This theorem immediately implies a (high-probability) BOBW regret guarantee when the losses are bounded (in $[0, 1]$) and protected with $\varepsilon$-Local Differential Privacy (LDP) via showing that the privatized losses (with Laplacian noise) have second moments bounded by $O(\varepsilon^{-1})$ (Agarwal & Singh, 2017; Tossou & Dimitrakakis, 2017; Zheng et al., 2020; Ren et al., 2020). To the best of our knowledge, this is the first result showing (high probability) BOBW regret guarantee with *pure* LDP protection, while the state-of-the-art result (Zheng et al., 2020) ensures *(the weaker) approximate* LDP protection, and only expected regret bounds are provided. Full details are given in Appendix E.

### 4.1 Algorithm Overview

Our algorithm design follows from the detect-switch framework by Bubeck & Slivkins (2012). The high-level idea is to keep performing statistical tests (to "identify" the environment) and carefully maintain the sampling distribution over all arms. Once some certain test fails (which implies that the environment is unlikely stochastic), we switch to Algorithm 1 and run it over the remaining rounds.

In each round after playing action $a_t$ sampled from distribution $w_t$, if any active action $i \in A_t$ satisfies the test in Eq. (5), it is deactivated, and we use $\tau_i$ and $q_i$ to store the round and the sampling

---

**Algorithm 2** SAO for heavy-tailed MABs (`SAO-HT`)

---

1: **Input:** failure probability $\zeta$; constant $c_1 \geqslant 6$

2: **Define:** $\beta = 12T^2 K \log(T)$; $M_{t,i} = u \dfrac{(t/K)^{\frac{1}{1+v}}}{(\log(\log(T)/\zeta))^{\frac{1}{3v+1}}}$; $B_{t,i} = u \left( \dfrac{N_{t,i}}{\log(2T/\zeta)} \right)^{\frac{1}{1+v}}$;

   $\text{Width}(t) = 12uK^{\frac{v}{1+v}} t^{\frac{1}{1+v}} \left( \log(\beta/\zeta) \right)^{\frac{3v}{3v+1}}$

3: **Initialization:** Play each action once; for every $i \in [K]$, let $w_{K+1,i} = \frac{1}{K}, \tau_i = T$; set $A_K = [K]$.

4: **for** $t = K+1, \ldots, T$ **do**

5:     Sample and play action $a_t \sim w_t$; observe $\ell_{t,a_t}$

6:     **for** $i \in A_{t-1}$ **do**

7:         **if**

$$\widehat{L}_{t,i} - \min_{j \in A_{t-1}} \widehat{L}_{t,j} > c_1 \text{Width}(t) \tag{5}$$

8:             **then** $A_t = A_{t-1} \backslash \{i\}, \tau_i = t$, and $q_i = w_{t,i}$

9:         **end if**

10:     **end for**

11:     **if** any of the three conditions holds **then** run Algorithm 1 for the remaining rounds, let $t_{\text{sw}} = t$, and let $q_i = w_{t,i}, \tau_i = t$ for every $i \in A_t$:

12:

$$\exists i \in [K] \text{ such that } \left| \widehat{L}_{t,i}/t - \widehat{\mu}_{t,i} \right| > 9u \left( \dfrac{\log(\beta/\zeta)}{N_{t,i}} \right)^{\frac{v}{1+v}} + \mathbb{I}\{i \in A_t\} \dfrac{\text{Width}(t)}{t}$$

$$+ \mathbb{I}\{i \notin A_t\} \dfrac{\text{Width}(t)}{\tau_i}, \tag{6}$$

$$\exists i \notin A_t \text{ such that } (\widehat{L}_{t,i} - \min_{j \in A_t} \widehat{L}_{t,j})/t > (c_1 + 4)\text{Width}(t)/(\tau_i - 1), \tag{7}$$

$$\exists i \notin A_t \text{ such that } (\widehat{L}_{t,i} - \min_{j \in A_t} \widehat{L}_{t,j})/t \leqslant (c_1 - 4)\text{Width}(t)/\tau_i \tag{8}$$

13:     **end if**

14:     $w_{t+1,i} = \mathbb{I}\{i \notin A_t\} q_i \tau_i/(t+1) + \mathbb{I}\{i \in A_t\} \left( 1 - \sum_{j \notin A_t} q_j \tau_j/(t+1) \right) / |A_t|, \forall i \in [K]$

15: **end for**

16: $q_i = w_{T,i}, \tau_i = T, \forall i \in A_T$

---

probability when it is deactivated, respectively (Line 8). After that, the algorithm performs tests in Eqs. (6)-(8) for environment identification. If any of them is satisfied, the procedure is terminated, and we instead run Algorithm 1 over the remaining rounds. We use $t_{\text{sw}}$ to denote the round when the algorithm switch happens. Otherwise, we update the distribution $w_{t+1}$ for the next round as in Line 14. To make notations and analyses well-defined, we deactivate all remaining arms in $A_{t_{\text{sw}}}$ when the algorithm switch happens.

### 4.2 Analysis Sketch

In this subsection, we present the key steps of the regret analysis in the proof of Theorem 2.

Before diving into the specific regime, we first derive a set of concentration results (good events). Informally, all of the following hold simultaneously with high probability in all rounds $t \in [T]$:

$$\left| \widehat{L}_{t,i} - \sum_{s=1}^{t} \mu_{s,i} \right| /t \leqslant \mathbb{I}\{i \in A_t\} \dfrac{\text{Width}(t)}{t} + \mathbb{I}\{i \notin A_t\} \dfrac{\text{Width}(t)}{\tau_i}, \tag{9}$$

$$\left| \widehat{\mu}_{t,i} - \dfrac{\sum_{s=1}^{t} \mu_{s,i} I_{s,i}}{N_{t,i}} \right| = \widetilde{O} \left( \dfrac{u}{(N_{t,i})^{\frac{v}{1+v}}} \right), \tag{10}$$

$$N_{t,i} = \widetilde{O} \left( q_i \tau_i \cdot (1 + \log t) \right), \tag{11}$$

where $\text{Width}(t) := 12uK^{\frac{v}{1+v}} t^{\frac{1}{1+v}} \left(\log(\beta/\zeta)\right)^{\frac{3v}{3v+1}}$. All analyses below are conditioned on these good events, and we omit "with high probability" in the arguments.

*Remark* 7. Our main non-trivial adaptation deviating from the case with bounded losses in Bubeck & Slivkins (2012) is the good event in Eq. (10) due to *jointly randomized and heavy-tailed* losses. In particular, we need the concentration result of trimmed mean $\widehat{\mu}_{t,i}$ specified in Eq. (10) *in both regimes*. In the stochastic regime, it been shown in Bubeck et al. (2013). We need Eq. (10) in adversarial regime (while Bubeck & Slivkins (2012) does not) as losses are *deterministic* therein. However, with heavy tails, the proof of Eq. (10) in the adversarial regime does not follow straightforwardly from the stochastic regime. More technical details are presented later in Remark 8. Nonetheless, this matter is specific to the nature of the trimmed estimator we use, which essentially could be replaced with any other estimator, as long as the concentration rate is preserved. However, it is unclear whether other estimators can handle non-identical distributions.

### 4.2.1 Analysis Overview in the Stochastic Regime

**Step 1: Showing that tests in Eqs. (6)-(8) are never satisfied.** We show by good events that tests in Eqs. (6)-(8) are never satisfied, implying that we never switch to Algorithm 1.

**Step 2: Building the connection between $\Delta_i$ and $\tau_i$.** From tests in Eqs. (7) and (8), we can show that for every suboptimal action $i$ with $\Delta_i > 0$, its sub-optimality gap $\Delta_i = \widetilde{O}((K/\tau_i)^{\frac{v}{1+v}})$. Intuitively, an action with a smaller sub-optimality gap stays active for a longer time.

**Step 3: Bounding the total regret.** By the definition of pseudo-regret, we now have

$$R_T = \sum_{i:\Delta_i>0} \Delta_i N_{T,i} = \widetilde{O}\left( \sum_{i:\Delta_i>0} \Delta_i q_i \tau_i \right) = \widetilde{O}\left( \sum_{i:\Delta_i>0} \Delta_i q_i K(\Delta_i)^{-1-\frac{1}{v}} \right). \tag{12}$$

We complete the proof by showing $\sum_{i:\Delta_i>0} q_i \leqslant \sum_{i=1}^{K} (1/i) = O(\log K)$.

### 4.2.2 Analysis Overview in the Adversarial Regime

In the adversarial regime, whenever we switch to Algorithm 1, it provides $\widetilde{O}(u(T - t_{\text{sw}})^{\frac{1}{1+v}} K^{\frac{v}{1+v}})$ (high-probability) regret guarantee for the remaining $(T - t_{\text{sw}})$ rounds. Therefore, it suffices to show that the cumulative regret before the switch is $\widetilde{O}(u(t_{\text{sw}})^{\frac{1}{1+v}} K^{\frac{v}{1+v}})$.

In our analysis, we trivially bound the regret in the single round $t_{\text{sw}}$ by $2u$, and it remains to show that the regret in the first $(t_{\text{sw}} - 1)$ rounds is $\widetilde{O}(u(t_{\text{sw}} - 1)^{\frac{1}{1+v}} K^{\frac{v}{1+v}})$, which is explained in the following.

**Step 1: Regret decomposition.** We first get an regret upper bound in terms of $i_t^* := \arg\min_{i \in [K]} \sum_{s=1}^{t} \mu_{s,i}$ for $t = t_{\text{sw}} - 1$:

$$\sum_{s=1}^{t_{\text{sw}}-1} \mu_{s,a_t} - \sum_{s=1}^{t_{\text{sw}}-1} \mu_{s,i^*} \leqslant \sum_{i=1}^{K} N_{t_{\text{sw}}-1,i} \left( \underbrace{\frac{\sum_{s=1}^{t_{\text{sw}}-1} \mu_{s,i} I_{s,i}}{N_{t_{\text{sw}}-1,i}} - \frac{\widehat{L}_{t_{\text{sw}}-1,i}}{t_{\text{sw}}-1}}_{\text{Part A}} + \underbrace{\frac{\widehat{L}_{t_{\text{sw}}-1,i} - \widehat{L}_{t_{\text{sw}}-1,i_{t_{\text{sw}}-1}^*}}{t_{\text{sw}}-1}}_{\text{Part B}} \right)$$

$$+ \underbrace{\widehat{L}_{t_{\text{sw}}-1,i_{t_{\text{sw}}-1}^*} - \sum_{s=1}^{t_{\text{sw}}-1} \mu_{s,i_{t_{\text{sw}}-1}^*}}_{\text{Part C}}, \tag{13}$$

and then we bound Parts A, B, and C, separately.

**Step 2: Bounding Part A.** We rewrite Part A as $\text{Part A} = \left( \frac{\sum_{s=1}^{t_{\text{sw}}-1} \mu_{s,i} I_{s,i}}{N_{t_{\text{sw}}-1,i}} - \widehat{\mu}_{t_{\text{sw}}-1,i} \right) + \left( \widehat{\mu}_{t_{\text{sw}}-1,i} - \frac{\widehat{L}_{t_{\text{sw}}-1,i}}{t_{\text{sw}}-1} \right)$, where the first term on the right-hand side is $O\left( u \left( \frac{\log(\beta/\zeta)}{N_{t_{\text{sw}}-1,i}} \right)^{\frac{v}{1+v}} \right)$ due to Eq. (10) and the second term is $O\left( u \left( \frac{\log(\beta/\zeta)}{N_{t_{\text{sw}}-1,i}} \right)^{\frac{v}{1+v}} + \frac{\text{Width}(t_{\text{sw}}-1)}{\tau_i-1} \right)$ due to test in Eq. (6).

*Remark* 8. Due to heavy tails, the losses are trimmed by some *history-dependent* threshold $B_{t,i}$ (which depends on the number of pulls $N_{t,i}$) for a rate-optimal concentration in good event (10). To

show this in the stochastic regime, one can treat the observed losses from one action as *i.i.d.* samples via the "reward tape/table" argument (Slivkins, 2019) and apply Bernstein's inequality for every fixed $N_{t,i}$ (and set the uniform upper bound as $B_{t,i}$ associated with it) as in Bubeck et al. (2013). However, one cannot simply follow the same path in the adversarial regime, since the distributions are *no longer identical*, and has to follow the martingale-based analysis (Agarwal et al., 2021, Lemma 6.2).

Now one may readily see the issue: The desired uniform upper bound $B_{t,i}$ is determined *on-the-fly*, while the standard Freedman's inequality for martingales (e.g., Lemma 11) requires a *fixed* uniform upper bound. To close this gap, we again exploit an adaptive variant of Freedman's inequality by Zimmert & Lattimore (2022) (Lemma 12; which was originally proposed for a totally different use, namely, obtaining high-probability bounds in adversarial bandits), in which we can replace the fixed uniform upper bound with the largest *realization*, satisfying our need perfectly.

**Step 3: Bounding Part B.** By considering two disjoint cases (i.e., action $i \in A_{t_{\mathrm{sw}}-1}$ or not), we can show Part B $= O\left(\frac{\mathrm{Width}(t_{\mathrm{sw}}-1)}{\tau_i - 1}\right)$ using the tests in Eqs. (5) and (7).

**Step 4: Bounding Part C.** The good event in Eq. (9) simply implies Part C $= O\left(\mathrm{Width}(t_{\mathrm{sw}} - 1)\right)$.

**Step 5: Putting all pieces together.** Combing Steps 1-4 and the good event in Eq. (11) yields

$$
\begin{aligned}
\sum_{s=1}^{t_{\mathrm{sw}}-1} \mu_{s,a_t} - \sum_{s=1}^{t_{\mathrm{sw}}-1} \mu_{s,i^*} &= O\left(\sum_{i=1}^{K} N_{t_{\mathrm{sw}}-1,i}\left(u\left(N_{t_{\mathrm{sw}}-1,i}\right)^{\frac{-v}{1+v}} + \frac{\mathrm{Width}(t_{\mathrm{sw}}-1)}{\tau_i - 1}\right) + \mathrm{Width}(t_{\mathrm{sw}}-1)\right) \\
&= \widetilde{O}\left(u\sum_{i=1}^{K}\left(N_{t_{\mathrm{sw}}-1,i}\right)^{\frac{1}{1+v}} + u\sum_{i=1}^{K} q_i K^{\frac{v}{1+v}}(t_{\mathrm{sw}}-1)^{\frac{1}{1+v}}\right) \\
&\quad + \widetilde{O}\left(u\sum_{i=1}^{K} \frac{K^{\frac{v}{1+v}}(t_{\mathrm{sw}}-1)^{\frac{1}{1+v}}}{\tau_i - 1} + \mathrm{Width}(t_{\mathrm{sw}}-1)\right) \\
&= \widetilde{O}(uK^{\frac{v}{1+v}}(t_{\mathrm{sw}}-1)^{\frac{1}{1+v}}).
\end{aligned}
\tag{14}
$$

## 5 Conclusion

In this paper, we show that there is indeed no fundamental barrier to achieving the BOBW guarantee in heavy-tailed MAB by relaxing a strong, hard-to-verify technical assumption on the loss distributions of the optimal action needed for the state-of-the-art results. We further leverage the increasing-learning-rates trick and the detect-switch framework to achieve the stronger high-probability guarantees. Our results also imply the first and near-optimal regret in the adversarial regime where the feedback could be contaminated in the Huber model, and the high-probability BOBW regret guarantee when losses are bounded and protected with pure LDP, while the state-of-the-art result only ensures the weaker approximate LDP protection with regret guarantees in expectation.

One follow-up question is whether the gap-dependent term $K(\Delta)^{-1/v}$ can be improved to the refined $\sum_{i:\Delta_i>0}(\Delta_i)^{-1/v}$, which is achieved only in the stochastic regime (Bubeck et al., 2013). Under the detect-switch framework, we tend to believe this is possible by adapting more sophisticated tests designed in Auer & Chiang (2016) for the bounded-loss case, where the gap dependency is improved from $K/\Delta$ to $\sum_{i:\Delta_i>0}(1/\Delta)$, although a higher computational complexity is expected.

It will also be interesting to understand whether canonical Online Learning algorithms (i.e., without explicit detection and switch) provably enjoy BOBW guarantees in heavy-tailed MAB (and if so, whether a refined gap dependency can be achieved). In other words, do heavy tails break the implicit adaption of Online Learning algorithms to the stochastic regime (in the worst case)? One promising direction is to still utilize log-barrier regularizer together with potentially more advance learning rate and/or trimming threshold design. Notably, one unique advantage of it over the detect-switch framework is that it typically directly extends the BOBW regret guarantee to the corrupted regime, which is an intermediate regime that smoothly extrapolates between the purely adversarial and stochastic regime (Zimmert & Seldin, 2021).

## Acknowledgements

We thank anonymous paper reviewers for their insightful feedback, especially reviewer q85G, who carefully read the proof and pointed out some gaps from the analysis in an earlier version of the manuscript. This work is supported in part by the NSF grants under CNS-2312833, CNS-2312835, and CNS-2153220, the Commonwealth Cyber Initiative (CCI), and Nokia Corporation.

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

# A  Related Work

In this section, we give a comprehensive discussion on the related work.

**Regret Minimization in the BOBW Setting (with Bounded Losses).** There are mainly two different approaches towards enjoying BOBW regret guarantees. The first approach is the detect-switch framework initially proposed by Bubeck & Slivkins (2012). The framework runs in a two-stage manner: In the first stage, the algorithm performs some tests in every round and carefully maintain the sampling distribution over all actions. Once any of the tests fails, it runs any off-the-shelf algorithm purely for the adversarial regime in the remaining rounds and inherits the regret guarantee of that algorithm. Bubeck & Slivkins (2012) proposed an algorithm achieving near-optimal regret in the adversarial regime and $O\left((\log T)^2 K/\Delta\right)$ regret in the stochastic regime, both with high-probability. Auer & Chiang (2016) further refined the tests and improved the gap-dependent term to the optimal $\sum_{i:\Delta_i>0}(1/\Delta_i)$. This framework has been generalized to setups including linear bandits (Lee et al., 2021) and bandits with feedback graph (Kong et al., 2023).

Somewhat surprisingly, since the work by Wei & Luo (2018), there have been a large group of papers showing that Online Learning algorithms (e.g., OMD), originally for the adversarial regime, implicit adapts to stochastic environments and enjoys logarithmic regrets. BOBW guarantees of Online Learning algorithms have been established in quite broad setups including MAB (Wei & Luo, 2018; Zimmert & Seldin, 2021; Jin et al., 2023), (contextual) linear bandits (Ito & Takemura, 2023; Kong et al., 2023; Kuroki et al., 2023; Kato & Ito, 2023), and (tabular) Markov Decision Processes (MDPs) (Jin & Luo, 2020; Jin et al., 2021). One desirable advantage of this approach is that, thanks to the flexibility of Online Learning framework, it is possible to additionally enjoy some other adaptive regret bounds on top of the standard BOBW guarantee (Ito, 2021; Tsuchiya et al., 2024).

**Regret Minimization in Heavy-tailed Stochastic Environments.** There are a large body of works focusing on regret minimization in the stochastic environment where the losses are potentially heavy-tailed (Bubeck et al., 2013; Shao et al., 2018; Lu et al., 2019; Ray Chowdhury & Gopalan, 2019; Zhong et al., 2021; Agrawal et al., 2021; Xue et al., 2021; Lee & Lim, 2022; Zhuang & Sui, 2021; Huang et al., 2024). The general recipe of the algorithm design is to derive concentration results for the loss mean estimation with robust estimators (e.g., median-of-means (Hsu & Sabato, 2014), truncated/trimmed estimator (Bubeck et al., 2013), and Huber estimator (Huber, 1996)), and then integrate them with celebrated optimism-based algorithms.

**High-probability Regrets in Adversarial MAB and the BOBW Setting.** The central piece towards high-probability bounds in adversarial MAB lies in balancing between the variance of the loss estimator and the total regret. In general, there are three types of approaches in the literature: 1) adding negative bonus to the Importance-Weighted (IW) estimates (Auer et al., 2002b; Zimmert & Lattimore, 2022), 2) utilizing log-barrier regularizer together with increasing learning rates (Lee et al., 2020), and 3) replacing the IW estimator with the Implicit-eXploration (IX) estimator (Neu, 2015). While the IX estimator enables a simplified analysis and is thus widely-used (e.g., Jin et al. (2020); Li et al. (2024)), it heavily exploits the non-negativity of the losses, and it is unclear to us how to adapt it to the case of potentially-negative losses. For a comparison between the first two approaches, readers are referred to Foster et al. (2021) and Zimmert & Lattimore (2022). The detect-switch framework by Bubeck & Slivkins (2012) naturally provides high-probability bounds in BOBW, whenever a subroutine algorithm achieving high-probability regret in the adversarial regime (e.g., EXP3.P (Auer et al., 2002b)) is integrated in a plug-and-play manner. While the Online Learning framework naturally provides BOBW guarantee via elegant analysis *in expectation* (Zimmert & Seldin, 2021), it is unclear how to adapt it for high-probability guarantees.

**Adaptive/Parameter-free Heavy-tailed MAB.** Throughout this work, we assume that the heavy tail parameters $u$ and $v$ are known to the learning algorithm. In this case, Bubeck et al. (2013) showed minimax lower bound $\Omega(uT^{\frac{1}{1+v}}K^{\frac{v}{1+v}})$ and gap-dependent lower bound $\Omega\left(\sum_{i:\Delta_i>0}(u/\Delta_i)^{1/v}(\log T)\right)$ using stochastic environments, so these lower bounds also apply to the adversarial regime. However, in practice one may not know the values of $u, v$ exactly in advance, therefore it is natural to ask whether it is possible to enjoy the same regret rate as if $u, v$ were known, which we refer to as "adaptation (to heavy tails) for free". On the negative side, Genalti et al. (2024) showed that there is no such an algorithm that can enjoy $\widetilde{O}(uT^{\frac{1}{1+v}}K^{\frac{v}{1+v}})$ worst-case regret for all unknown $u, v$ in the stochastic regime. In terms of gap-dependent regret, similar "no adaptation for free" effect has been shown in an earlier work by Ashutosh et al. (2021). On the positive side, in the same work by

Genalti et al. (2024), it is shown that, somewhat surprisingly, with the same "truncated non-negative losses" assumption (Assumption 2), a UCB-based algorithm proposed by them enjoys "adaptation for free" in terms of both minimax and gap-dependent regrets in the stochastic regime. With the same assumption in the adversarial regime, Huang et al. (2022) proposed a variant of FTRL, which achieves the $O(uK^{\frac{v}{1+v}}T^{\frac{1}{1+v}} + \sqrt{KT})$ worst-case regret guarantee. In a very recent work by Chen et al. (2024), an FTRL-based algorithm named uniINF is shown to enjoy the BOBW guarantee even when $u$ and $v$ are both unknown, although the gap dependency of their logarithmic regret is $K/\Delta^{1/v}$, and still Assumption 2 is needed. Exploring broader scenarios in which we can enjoy "adaptation for free" (and to what extent) is an interesting future direction.

# B   Omitted Details in Section 3

In this section, we provide complete proof for Theorem 1.

To begin with, we first define $\ell'_{t,i}$ as the trimmed value of $\ell_{t,i}$ with respect to $M$, i.e., $\ell'_{t,i} = \ell_{t,i}\mathbb{I}\{|\ell_{t,i}| \leqslant M\}$. Then, we decompose the pseudo-regret by

$$R_T = \underbrace{\sum_{t=1}^{T}\left(\mu_{t,a_t} - \ell'_{t,a_t}\right) + \sum_{t=1}^{T}\left(\ell'_{t,i^*} - \mu_{t,i^*}\right)}_{\text{TRIMERR}} + \underbrace{\sum_{t=1}^{T}\left(\ell'_{t,a_t} - \ell'_{t,i^*}\right)}_{\text{TRIMREG}}.$$

## B.1   Bounding TRIMERR

**Lemma 1.** *For any fixed trimming threshold $M_{t,i} = M > 0$ and $\zeta < 1/(e\log T)$, with probability at least $1 - \zeta$, it holds that*

$$\text{TRIMERR} \leqslant 2Tu^{1+v}M^{-v} + 8\sqrt{Tu^{1+v}M^{1-v}\log(2\log(T)/\zeta)} + 8M\log(2\log(T)/\zeta).$$

*Proof of Lemma 1.* First note that $\left(\mu'_{t,a_t} - \ell'_{t,a_t}\right)_{t=1,\dots,T}$ form a martingale difference sequence adapted to the filtration $\mathcal{F}_1,\dots,\mathcal{F}_T$ where $\mathcal{F}_t := \sigma(a_1,\ell_{1,a_1},\dots,a_{t-1},\ell_{t-1,a_{t-1}})$. Moreover, we have $\left|\mu'_{t,a_t} - \ell'_{t,a_t}\right| \leqslant 2M$ and

$$\begin{aligned}
\mathbb{E}\left[(\mu'_{t,a_t} - \ell'_{t,a_t})^2\big|\mathcal{F}_t\right] &= \sum_{i=1}^{K} w_{t,i}\mathbb{E}\left[(\mu'_{t,i} - \ell'_{t,i})^2\big|\mathcal{F}_t, a_t = i\right] \\
&= \sum_{i=1}^{K} w_{t,i}\left(\mathbb{E}\left[(\ell'_{t,i})^2\big|\mathcal{F}_t, a_t = i\right] + (\mu'_{t,i})^2 - 2\mu'_{t,i}\mathbb{E}\left[\ell'_{t,i}\big|\mathcal{F}_t, a_t = i\right]\right) \\
&\leqslant \sum_{i=1}^{K} w_{t,i}\mathbb{E}\left[(\ell'_{t,i})^2\big|\mathcal{F}_t, a_t = i\right] \\
&\leqslant \sum_{i=1}^{K} w_{t,i}\mathbb{E}\left[(\ell'_{t,i})^{1+v}M^{1-v}\big|\mathcal{F}_t, a_t = i\right] \\
&\leqslant u^{1+v}M^{1-v}, \tag{15}
\end{aligned}$$

where in the last inequality we utilize the fact that $\sum_{i=1}^{K} w_{t,i} = 1$ and Assumption 1. By Lemma 11, we have with probability at least $1 - \zeta$,

$$\sum_{t=1}^{T}\left(\mu'_{t,a_t} - \ell'_{t,a_t}\right) \leqslant 4\sqrt{Tu^{1+v}M^{1-v}\log(\log(T)/\zeta)} + 4M\log(\log(T)/\zeta). \tag{16}$$

Together with Lemma 8 which bounds $\mu_{t,a_t} - \mu'_{t,a_t}$, we arrive at

$$\sum_{t=1}^{T}\left(\mu_{t,a_t} - \ell'_{t,a_t}\right) \leqslant Tu^{1+v}M^{-v} + 4\sqrt{Tu^{1+v}M^{1-v}\log(\log(T)/\zeta)} + 4M\log(\log(T)/\zeta). \tag{17}$$

With exactly the same reasoning, we have with probability at least $1 - \zeta$ that

$$\sum_{t=1}^{T} \left( \ell'_{t,i^*} - \mu_{t,i^*} \right) \leqslant T u^{1+v} M^{-v} + 4\sqrt{T u^{1+v} M^{1-v} \log(\log(T)/\zeta)} + 4M \log(\log(T)/\zeta). \tag{18}$$

Taking a union bound over the two terms completes the proof. $\qquad\square$

## B.2 Bounding TRIMREG

For any $i \in [K]$, let $y^i$ be the $K$-dimensional one-hot vector such that the $i$-th element is 1, and all the others are zero. Define vector $y' := (1 - \lambda)y^{i^*} + \lambda w_0$, where $w_0 = (1/K, \ldots, 1/K)$ is the uniform exploration. We can rewrite TRIMREG as

$$\sum_{t=1}^{T} \left( \ell'_{t,a_t} - \ell'_{t,i^*} \right) = \sum_{t=1}^{T} \left( \langle w_t, \widehat{\ell}_t \rangle - \langle y^{i^*}, \ell'_t \rangle \right)$$

$$= \underbrace{\sum_{t=1}^{T} \langle w_t - y', \widehat{\ell}_t \rangle}_{\text{TRIMREG I}} + \underbrace{\sum_{t=1}^{T} \langle y' - y^{i^*}, \ell'_t \rangle}_{\text{TRIMREG II}} + \underbrace{\sum_{t=1}^{T} \langle y', \widehat{\ell}_t - \ell'_t \rangle}_{\text{TRIMREG III}}. \tag{19}$$

In the following subsections, we bound each of the three terms in the right-hand-sight. To give a preview:

- TRIMREG I is the standard online learning regret over loss sequence $\widehat{\ell}_1, \cdots, \widehat{\ell}_T$, and is handled using regret analysis of OMD.

- TRIMREG II is a bias term due to that we cannot directly compete with $y^{i^*}$ and have to choose a close neighbor $y'$. It is under control since $y'$ and $y^{i^*}$ are close to each other.

- TRIMREG III is the main challenge in getting high-probability bounds in adversarial bandits. While it contains some terms that could be potentially large due to the high variance of $\widehat{\ell}_t$, they would be cancelled by some negative terms in TRIMREG I introduced by the increasing learning rates so that eventually the sum of the three terms can be bounded.

### B.2.1 Bounding TRIMREG I

**Lemma 2.** *Suppose $\eta \leqslant \frac{1}{10M}$ and $\lambda = T^{\frac{-v}{1+v}} K^{\frac{-1}{1+v}}$, for any $M_{t,i} = M$, Algorithm 1 ensures that with probability at least $1 - \zeta$,*

$$\text{TRIMREG I} \leqslant \frac{2K \log T}{\eta} - \frac{\langle y', \rho_T \rangle}{10\eta \log T} + 5\eta M^{1-v} T u^{1+v}$$
$$+ 20\eta \sqrt{T u^{1+v} M^{3-v} \log(\log(T)/\zeta)} + 10\eta M^2 \log(\log(T)/\zeta).$$

*Proof of Lemma 2.* The proof is a combination of Agarwal et al. (2017, Lemma 12) and Lee et al. (2020, Lemma 2.1). For the (active) OMD update rule $w_{t+1} = \text{argmin}_{x \in \Omega'} \left( \langle x, \widehat{\ell}_t \rangle + D_{\psi_t}(x, w_t) \right)$ in Algorithm 1, it is equivalent to first obtaining some intermediate vector $\widetilde{w}_{t+1}$ such that $\nabla \psi_t(\widetilde{w}_{t+1}) = \nabla \psi_t(w_t) - \widehat{\ell}_t$, and then obtaining $w_{t+1} = \text{argmin}_{x \in \Omega'} D_{\psi_t}(x, \widetilde{w}_{t+1})$.

For any competitor $y \in \Omega'$, in each round $t \in [T]$, we have

$$\langle w_t - y, \widehat{\ell}_t \rangle = \langle w_t - y, \nabla \psi_t(w_t) - \nabla \psi_t(\widetilde{w}_{t+1}) \rangle$$
$$\overset{\text{(a)}}{=} D_{\psi_t}(y, w_t) - D_{\psi_t}(y, \widetilde{w}_{t+1}) + D_{\psi_t}(w_t, \widetilde{w}_{t+1})$$
$$\overset{\text{(b)}}{\leqslant} D_{\psi_t}(y, w_t) - D_{\psi_t}(y, w_{t+1}) + D_{\psi_t}(w_t, \widetilde{w}_{t+1}), \tag{20}$$

where in step (a) we utilize the definition of Bregman divergence (or the so-called "three-point property") and step (b) is due to the generalized Pythagorean theorem.

Throughout this proof, we choose $\lambda = T^{\frac{-v}{1+v}} K^{\frac{-1}{1+v}}$ as in Algorithm 1. Recall that we define $y' := (1-\lambda)y^{i^*} + \lambda w_0$. Therefore, for any $i \in [K]$, we have $y_i' \geqslant \lambda/K$ and hence $y' \in \Omega'$.

For any scalar $c > 0$, we define function $h(c) := c - 1 - \log c$, which is always non-negative. Taking the summation over all $T$ rounds and letting $y = y'$, we get

$$\sum_{t=1}^{T} \langle w_t - y', \widehat{\ell}_t \rangle = \sum_{t=1}^{T} \left( D_{\psi_t}(y', w_t) - D_{\psi_t}(y', w_{t+1}) + D_{\psi_t}(w_t, \widetilde{w}_{t+1}) \right)$$

$$\leqslant D_{\psi_1}(y', w_1) + \sum_{t=1}^{T-1} \left( D_{\psi_{t+1}}(y', w_{t+1}) - D_{\psi_t}(y', w_{t+1}) \right) + \sum_{t=1}^{T} D_{\psi_t}(w_t, \widetilde{w}_{t+1}), \tag{21}$$

where in the inequality we get rid of $-D_{\psi_T}(y', w_{T+1})$ since Bregman divergence is always non-negative. In the following, we bound each of these three terms in the right-hand-side.

**Step 1: bounding the first term.** Plugging in the exact expression of Bregman divergence concerning the log-barrier regularizer $\psi_t$ we define in Algorithm 1, we have

$$D_{\psi_1}(y', w_1) = \frac{1}{\eta} \sum_{i=1}^{K} h\left( \frac{y_i'}{w_{1,i}} \right) = \frac{1}{\eta} \sum_{i=1}^{K} \left( \frac{y_i'}{w_{1,i}} - 1 - \log\left( \frac{y_i'}{w_{1,i}} \right) \right)$$

$$= \frac{1}{\eta} \sum_{i=1}^{K} \log\left( \frac{1}{K y_i'} \right). \tag{22}$$

By choosing $\lambda = T^{\frac{-v}{1+v}} K^{\frac{-1}{1+v}}$, we have $y_i' \geqslant \lambda/K = T^{\frac{-v}{1+v}} K^{-1-\frac{1}{1+v}}$ and

$$\frac{1}{\eta} \sum_{i=1}^{K} \log\left( \frac{1}{K y_i'} \right) \leqslant \frac{K \log\left( T^{\frac{v}{1+v}} K^{\frac{1}{1+v}} \right)}{\eta} \leqslant \frac{K \log T}{\eta}.$$

Recall that here we utilize the assumption that $K \leqslant T$.

**Step 2: bounding the second term.** Plugging in the expression of Bregman divergence associated with log-barrier, we get

$$\sum_{t=1}^{T-1} \left( D_{\psi_{t+1}}(y', w_{t+1}) - D_{\psi_t}(y', w_{t+1}) \right) = \sum_{i=1}^{K} \sum_{t=1}^{T-1} \left( \frac{1}{\eta_{t+1,i}} - \frac{1}{\eta_{t,i}} \right) h\left( \frac{y_i'}{w_{t+1,i}} \right). \tag{23}$$

Now we look at each action $i$. Recall that, if the learning rate does not increase at round $t$, then $\left( \frac{1}{\eta_{t+1,i}} - \frac{1}{\eta_{t,i}} \right) h\left( \frac{y_i'}{w_{t+1,i}} \right)$ is simply 0. Otherwise, we have $\eta_{t+1,i} > \eta_{t,i}$, and as a result, $\left( \frac{1}{\eta_{t+1,i}} - \frac{1}{\eta_{t,i}} \right) h\left( \frac{y_i'}{w_{t+1,i}} \right) < 0$.

Therefore, if we let $n_i$ denote the total number of learning rate changes in action $i$, and let $t_{n_i}$ denote the round when the last change happens (such that $\eta_{T,i} = \eta_{t_{n_i}+1,i} = \kappa \eta_{t_{n_i},i} = \kappa^{n_i} \eta$), we have

$$\sum_{t=1}^{T-1} \left( \frac{1}{\eta_{t+1,i}} - \frac{1}{\eta_{t,i}} \right) h\left( \frac{y_i'}{w_{t+1,i}} \right) \leqslant \left( \frac{1}{\eta_{t_{n_i}+1,i}} - \frac{1}{\eta_{t_{n_i},i}} \right) h\left( \frac{y_i'}{w_{t_{n_i}+1,i}} \right)$$

$$= \left( \frac{1}{\kappa^{n_i} \eta} - \frac{\kappa}{\kappa^{n_i} \eta} \right) h\left( \frac{y_i'}{w_{t_{n_i}+1,i}} \right)$$

$$= \frac{1-\kappa}{\kappa^{n_i} \eta} h\left( \frac{y_i'}{w_{t_{n_i}+1,i}} \right). \tag{24}$$

By Lemma 9, when $\lambda = T^{\frac{-v}{1+v}} K^{\frac{-1}{1+v}}$, we have $n_i \leqslant \log_2(T^{\frac{v}{1+v}} K^{\frac{1}{1+v}}) \leqslant \log_2 T$ and $\eta_{t,i} \leqslant e^{\frac{\log_2 T}{\log T}} \eta \leqslant 5\eta$. Together with $1 - \kappa = 1 - e^{1/\log T} \leqslant -\frac{1}{\log T}$, we get

$$\frac{1-\kappa}{\kappa^{n_i} \eta} h\left( \frac{y_i'}{w_{t_{n_i}+1,i}} \right) \leqslant \frac{-h\left( \frac{y_i'}{w_{t_{n_i}+1,i}} \right)}{5\eta \log T}. \tag{25}$$

It is left to upper-bound term $-h\left(\frac{y_i'}{w_{t_{n_i}+1,i}}\right)$. Noticing that

$$\frac{y_i'}{w_{t_{n_i}+1,i}} = \frac{y_i'\rho_{T,i}}{2} \leqslant \frac{\rho_{T,i}}{2} = \frac{1}{w_{t_{n_i}+1,i}} \leqslant T, \tag{26}$$

we get

$$-h\left(\frac{y_i'}{w_{t_{n_i}+1,i}}\right) = -h\left(\frac{y_i'\rho_{T,i}}{2}\right) = \log\left(\frac{y_i'\rho_{T,i}}{2}\right) + 1 - \frac{y_i'\rho_{T,i}}{2} \leqslant \log T + 1 - \frac{y_i'\rho_{T,i}}{2}. \tag{27}$$

Taking the summation over all actions, we finish bounding the second term as

$$\sum_{i=1}^{K}\sum_{t=1}^{T-1}\left(\frac{1}{\eta_{t+1,i}} - \frac{1}{\eta_{t,i}}\right)h\left(\frac{y_i'}{w_{t+1,i}}\right) \leqslant \sum_{i=1}^{K}\frac{\log T + 1 - \frac{y_i'\rho_{T,i}}{2}}{5\eta\log T} \leqslant \frac{K\log T}{\eta} - \frac{\langle y', \rho_T\rangle}{10\eta\log T}. \tag{28}$$

**Step 3: bounding the third term.** Given the fact that $\nabla\psi_t(\widetilde{w}_{t+1}) = \nabla\psi_t(w_t) - \widehat{\ell}_t$, we have

$$\frac{-1}{\widetilde{w}_{t+1,i}\eta_{t,i}} = \frac{-1}{w_{t,i}\eta_{t,i}} - \widehat{\ell}_{t,i}, \tag{29}$$

which implies that

$$\frac{w_{t,i}}{\widetilde{w}_{t+1,i}} = 1 + \eta_{t,i}w_{t,i}\widehat{\ell}_{t,i}. \tag{30}$$

Now we have

$$\begin{aligned}\sum_{t=1}^{T}D_{\psi_t}(w_t, \widetilde{w}_{t+1}) &= \sum_{t=1}^{T}\sum_{i=1}^{K}\frac{1}{\eta_{t,i}} \cdot h\left(\frac{w_{t,i}}{\widetilde{w}_{t+1,i}}\right) \\ &= \sum_{t=1}^{T}\sum_{i=1}^{K}\frac{1}{\eta_{t,i}} \cdot h\left(1 + \eta_{t,i}w_{t,i}\widehat{\ell}_{t,i}\right) \\ &= \sum_{t=1}^{T}\sum_{i=1}^{K}\frac{1}{\eta_{t,i}} \cdot \left(\eta_{t,i}w_{t,i}\widehat{\ell}_{t,i} - \log\left(1 + \eta_{t,i}w_{t,i}\widehat{\ell}_{t,i}\right)\right).\end{aligned} \tag{31}$$

To proceed, we first show that if $\eta \leqslant \frac{1}{10M}$, it holds that

$$\eta_{t,i}w_{t,i}\widehat{\ell}_{t,i} \geqslant -0.5. \tag{32}$$

First, it is trivially true whenever $\ell_{t,i}' \geqslant 0$ as the left-hand-sight is non-negative. Since $\ell_{t,i}'$ is at most as negative as $-M$ and $\eta_{t,i} \leqslant 5\eta$, it is left to show that

$$\eta w_{t,i}\frac{M}{w_{t,i}} \leqslant 0.1, \tag{33}$$

which is clearly satisfied when $\eta \leqslant \frac{1}{10M}$.

By applying the fact that $c - \log(1 + c) \leqslant c^2, \forall c \geqslant -0.5$ to Eq. (31), we get

$$\begin{aligned}\sum_{t=1}^{T}D_{\psi_t}(w_t, \widetilde{w}_{t+1}) &\leqslant \sum_{t=1}^{T}\sum_{i=1}^{K}\frac{1}{\eta_{t,i}}\left(\eta_{t,i}w_{t,i}\widehat{\ell}_{t,i}\right)^2 \\ &= \sum_{t=1}^{T}\eta_{t,a_t}(w_{t,a_t})^2(\ell_{t,a_t}'/w_{t,a_t})^2 \\ &\leqslant 5\eta\sum_{t=1}^{T}(\ell_{t,a_t}')^2 \leqslant 5\eta M^{1-v}\sum_{t=1}^{T}\left|\ell_{t,a_t}'\right|^{1+v}.\end{aligned} \tag{34}$$

We now finish bounding the third term by deriving a high-probability bound on $\sum_{t=1}^{T} \left| \ell'_{t,a_t} \right|^{1+v}$ (and multiplying it by $5\eta M^{1-v}$).

Consider the martingale difference sequence $\left( \left| \ell'_{t,a_t} \right|^{1+v} - \mathbb{E}\left[ \left| \ell'_{t,a_t} \right|^{1+v} \right] \right)_{t \in [T]}$, we have

$$\left| \left| \ell'_{t,a_t} \right|^{1+v} - \mathbb{E}\left[ \left| \ell'_{t,a_t} \right|^{1+v} \right] \right| \leqslant M^{1+v}$$

almost surely, and

$$
\begin{aligned}
\mathbb{E}\left[ \left( \left| \ell'_{t,a_t} \right|^{1+v} - \mathbb{E}\left[ \left| \ell'_{t,a_t} \right|^{1+v} \right] \right)^2 \Big| \mathcal{F}_t \right] &= \sum_{i=1}^{K} w_{t,i} \mathbb{E}\left[ \left( \left| \ell'_{t,i} \right|^{1+v} - \mathbb{E}\left[ \left| \ell'_{t,i} \right|^{1+v} \right] \right)^2 \Big| \mathcal{F}_t, a_t = i \right] \\
&\leqslant \sum_{i=1}^{K} w_{t,i} \mathbb{E}\left[ \left| \ell'_{t,i} \right|^{2+2v} \Big| \mathcal{F}_t, a_t = i \right] \\
&\leqslant \sum_{i=1}^{K} w_{t,i} M^{1+v} \mathbb{E}\left[ \left| \ell'_{t,i} \right|^{1+v} \Big| \mathcal{F}_t, a_t = i \right] \\
&\leqslant u^{1+v} M^{1+v}.
\end{aligned}
\tag{35}
$$

By Lemma 11, we have with probability at least $1 - \zeta$ that

$$
\begin{aligned}
\sum_{t=1}^{T} \left| \ell'_{t,a_t} \right|^{1+v} &\leqslant \sum_{t=1}^{T} \mathbb{E}\left[ \left| \ell'_{t,a_t} \right|^{1+v} \right] + 4\sqrt{Tu^{1+v}M^{1+v}\log(\log(T)/\zeta)} + 2M^{1+v}\log(\log(T)/\zeta) \\
&\leqslant Tu^{1+v} + 4\sqrt{Tu^{1+v}M^{1+v}\log(\log(T)/\zeta)} + 2M^{1+v}\log(\log(T)/\zeta).
\end{aligned}
\tag{36}
$$

At this point, we have bounded all three terms, and putting them together completes the proof $\qquad \square$

### B.2.2 Bounding TRIMREG II

**Lemma 3** (Upper bound on TRIMREG II). *For any fixed $M > 0$ and $\lambda \in (0,1)$, with probability at least $1 - \zeta$, it holds that*

$$\text{TRIMREG II} \leqslant \lambda u K T + \lambda\sqrt{2KTu^{1+v}M^{1-v}\log(1/\zeta)} + \frac{4}{3}\lambda M \log(1/\zeta).$$

*Proof of Lemma 3.* By Hölder's inequality, we have

$$\sum_{t=1}^{T} \langle y' - y^{i^*}, \ell'_t \rangle \leqslant \sum_{t=1}^{T} \left\| y' - y^{i^*} \right\|_\infty \|\ell'_t\|_1 = \sum_{t=1}^{T} \max\{\lambda/K, \lambda(1-1/K)\} \cdot \|\ell'_t\|_1 \leqslant \lambda \sum_{t=1}^{T} \sum_{i=1}^{K} \left| \ell'_{t,i} \right|.
\tag{37}$$

Define $\overline{\mu}_{t,i} = \mathbb{E}_{\ell \sim P_{t,i}}\left[ |\ell| \cdot \mathbb{I}\{|\ell| \leqslant M\} \right], \forall t \in [T], i \in [K]$, we have

$$\overline{\mu}_{t,i} \leqslant \mathbb{E}_{\ell \sim P_{t,i}}\left[ |\ell| \right] \leqslant \mathbb{E}_{\ell \sim P_{t,i}}\left[ |\ell|^{1+v} \right]^{\frac{1}{1+v}} \leqslant u.
\tag{38}$$

Applying Lemma 10 to $\left( \left| \ell'_{t,i} \right| - \overline{\mu}_{t,i} \right)_{t \in [T], i \in [K]}$, since $\left| \left| \ell'_{t,i} \right| - \overline{\mu}_{t,i} \right| \leqslant 2M$ and

$$\mathbb{E}\left[ (\left| \ell'_{t,i} \right| - \overline{\mu}_{t,i})^2 \right] \leqslant \mathbb{E}\left[ \left| \ell'_{t,i} \right|^2 \right] \leqslant u^{1+v}M^{1-v},
\tag{39}$$

we get that with probability at least $1 - \zeta$,

$$\sum_{t=1}^{T}\sum_{i=1}^{K} \left| \ell'_{t,i} \right| \leqslant \sum_{t=1}^{T}\sum_{i=1}^{K} \overline{\mu}_{t,i} + \sqrt{2KTu^{1+v}M^{1-v}\log(1/\zeta)} + \frac{4}{3}M\log(1/\zeta),$$

which implies that

$$\text{TRIMREG II} \leqslant \lambda u K T + \lambda\sqrt{2KTu^{1+v}M^{1-v}\log(1/\zeta)} + \frac{4}{3}\lambda M \log(1/\zeta).$$

$\qquad \square$

### B.2.3 Bounding TRIMREG III

**Lemma 4** (Upper bound on TRIMREG III)**.** *For any fixed $M > 0$ and $\lambda \in (0,1)$, with probability at least $1 - \zeta$, it holds that*

$$\text{TRIMREG III} \leqslant 3Tu^{1+v}M^{-v} + 3M\langle y', \rho_T\rangle \log\left(\max\{\sqrt{2Tu^{1+v}M^{-1-v}K/\lambda}, 4K/\lambda\}/\zeta\right).$$

*Proof of Lemma 4.* We note that $\left(\langle y', \widehat{\ell}_t - \ell'_t\rangle/M\right)_{t\in[T]}$ [6] form a martingale sequence, and $\mathbb{E}\left[\langle y', \widehat{\ell}_t - \ell'_t\rangle/M \big| \mathcal{F}_t\right]$ is clearly finite since $w_{t,i} \geqslant \lambda/K$. Moreover, we have

$$
\begin{aligned}
\mathbb{E}\left[(\langle y', \widehat{\ell}_t - \ell'_t\rangle/M)^2 \big| \mathcal{F}_t\right] &\leqslant \mathbb{E}\left[(\langle y', \widehat{\ell}_t\rangle/M)^2 \big| \mathcal{F}_t\right] \\
&= \mathbb{E}\left[\frac{(y'_{a_t})^2(\ell'_{t,a_t}/M)^2}{(w_{t,a_t})^2} \bigg| \mathcal{F}_t\right] \\
&= \sum_{i=1}^{K} w_i \frac{(y'_i)^2}{(w_{t,i})^2} \mathbb{E}\left[(\ell'_{t,i}/M)^2 | \mathcal{F}_t, a_t = i\right] \\
&\leqslant \sum_{i=1}^{K} \frac{(y'_i)^2}{w_{t,i}} u^{1+v} M^{-1-v} \\
&\leqslant u^{1+v} M^{-1-v}\langle y', \rho_T\rangle, \hspace{2cm} (40)
\end{aligned}
$$

where in the last step we utilize the facts that $y'_i \leqslant 1$ and $1/w_{t,i} \leqslant \rho_{T,i}$.

Noting

$$\left|\langle y', (\widehat{\ell}_t - \ell'_t)\rangle\right|/M \leqslant \sum_{i=1}^{K} y'_i(\frac{1}{w_{t,i}} + 1) \leqslant \sum_{i=1}^{K} y'_i \cdot \frac{2}{w_{t,i}} \leqslant 2\langle y', \rho_T\rangle, \hspace{1cm} (41)$$

by Lemma 12, we have with probability at least $1 - \zeta$ that,

$$
\begin{aligned}
\sum_{t=1}^{T}\langle y', \widehat{\ell}_t - \ell'_t\rangle &\leqslant 3M\sqrt{Tu^{1+v}M^{-1-v}\langle y', \rho_T\rangle \iota} + 2M\max\{1, 2\langle y', \rho_T\rangle\}\iota \\
&\overset{(a)}{\leqslant} \frac{4.5Tu^{1+v}M^{-v}}{2} + \frac{2M\langle y', \rho_T\rangle\iota}{2} + 2M\max\{1, 2\langle y', \rho_T\rangle\}\iota \\
&\leqslant 3\left(Tu^{1+v}M^{-v} + M\max\{1, 2\langle y', \rho_T\rangle\}\iota\right), \hspace{1cm} (42)
\end{aligned}
$$

where $\iota := \log(2\max\{\sqrt{Tu^{1+v}M^{-1-v}\langle y', \rho_T\rangle}, 1, 2\langle y', \rho_T\rangle\}/\zeta)$ and step (a) is due to the elementary inequality $\sqrt{x_1 x_2} \leqslant \frac{x_1 + x_2}{2}, \forall x_1, x_2 \geqslant 0$.

We complete the proof by noticing that $1 \leqslant 2K \leqslant \langle y', \rho_T\rangle \leqslant \frac{2}{\lambda/K}$ and apply it to $\iota$. □

### B.2.4 Putting things together

By Lemmas 1, 2, 3, and 4, with probability at least $1 - \zeta$, we have

$$
\begin{aligned}
R_T \leqslant\ & 2Tu^{1+v}M^{-v} + 8\sqrt{Tu^{1+v}M^{1-v}\log(8\log(T)/\zeta)} + 8M\log(8\log(T)/\zeta) \\
& + \frac{2K\log T}{\eta} - \frac{\langle y', \rho_T\rangle}{10\eta\log T} + 5\eta M^{1-v}Tu^{1+v} \\
& + 20\eta\sqrt{Tu^{1+v}M^{3-v}\log(4\log(T)/\zeta)} + 10\eta M^2\log(4\log(T)/\zeta) \\
& + \lambda u KT + \lambda\sqrt{2KTu^{1+v}M^{1-v}\log(4/\zeta)} + \frac{4}{3}\lambda M\log(4/\zeta) \\
& + 3Tu^{1+v}M^{-v} + 3M\langle y', \rho_T\rangle \log\left(4\max\{\sqrt{2Tu^{1+v}M^{-1-v}K/\lambda}, 4K/\lambda\}/\zeta\right). \hspace{0.5cm} (43)
\end{aligned}
$$

---

[6](Re-)Scaling by $M$ is of course not necessary, which however simplifies the algebra calculations after applying Lemma 12.

Choosing $M = u \cdot (T/K)^{\frac{1}{1+v}}$ and $\lambda = T^{\frac{-v}{1+v}} K^{\frac{-1}{1+v}}$, we have

$$
\begin{aligned}
R_T &\leqslant 2uK^{\frac{v}{1+v}} T^{\frac{1}{1+v}} + 8\sqrt{u^2 T^{\frac{2}{1+v}} K^{\frac{v-1}{1+v}} \log(8\log(T)/\zeta)} + 8uT^{\frac{1}{1+v}} K^{\frac{-1}{1+v}} \log(8\log(T)/\zeta) \\
&\quad + \frac{2K\log T}{\eta} - \frac{\langle y', \rho_T\rangle}{10\eta\log T} + 5\eta M^{1-v} T u^{1+v} \\
&\quad + 20\sqrt{Tu^{1+v}u^{1-v}(T/K)^{\frac{1-v}{1+v}}\eta^2 M^2 \log(4\log(T)/\zeta)} + 10\eta M^2 \log(4\log(T)/\zeta) \\
&\quad + uT^{\frac{1}{1+v}} K^{\frac{v}{1+v}} + uT^{\frac{1}{1+v}} K^{\frac{v}{1+v}}\sqrt{2\log(4/\zeta)} + \frac{4}{3}uT^{\frac{1-v}{1+v}} K^{\frac{-2}{1+v}}\log(4/\zeta) \\
&\quad + 3uK^{\frac{v}{1+v}} T^{\frac{1}{1+v}} + 3M\langle y', \rho_T\rangle \log\left(4\max\{\sqrt{2K^{\frac{2+v}{1+v}} T^{\frac{v}{1+v}}}, 4K^{\frac{2+v}{1+v}} T^{\frac{v}{1+v}}\}/\zeta\right) \\
&\leqslant 7uK^{\frac{v}{1+v}} T^{\frac{1}{1+v}}\sqrt{2\log(4/\zeta)} + \frac{2K\log T}{\eta} - \frac{\langle y', \rho_T\rangle}{10\eta\log T} + 5\eta M^{1-v} T u^{1+v} \\
&\quad + 20\sqrt{u^2 T^{\frac{2}{1+v}} K^{\frac{v-1}{1+v}}\eta^2 M^2 \log(4\log(T)/\zeta)} + 10\eta M^2 \log(4\log(T)/\zeta) \\
&\quad + 3M\langle y', \rho_T\rangle \log(8KT/\zeta) + o\left(uK^{\frac{v}{1+v}} T^{\frac{1}{1+v}}\log(T/\zeta)\right).
\end{aligned}
\tag{44}
$$

Finally, choosing $\eta = \min\{\frac{1}{10M}, \frac{1}{40M\log(T)\log(8KT/\zeta)}\} = \frac{1}{40M\log(T)\log(8KT/\zeta)}$ to cancel the terms containing $\langle y', \rho_T\rangle$ ensures that

$$
R_T = O(uK^{\frac{v}{1+v}} T^{\frac{1}{1+v}}(\log T)^2 \log(T/\zeta)).
$$

## C  Omitted Details in Section 4

### C.1  Concentration Results

In this subsection, we derive concentration results needed for the analysis, adapted from Bubeck & Slivkins (2012) to suit our heavy-tailed case.

**Lemma 5** (Concentration on the trimmed importance-weighted estimator). *Suppose* $M_{t,i} = u \cdot (t/K)^{\frac{1}{1+v}} \cdot (\log(\log(T)/\zeta))^{\frac{-1}{3v+1}}$. *In both regimes, we have with probability at least* $1 - \zeta$ *that, for any* $i \in [K]$ *and* $t \in [T]$, *if* $(K+1) \leqslant t \leqslant t_{sw}$, *then*

$$
\left|\widehat{L}_{t,i} - \sum_{s=1}^{t}\mu_{s,i}\right| \leqslant 6u\left(\sqrt{(t/K)^{\frac{1-v}{1+v}}\left(\sum_{s=1}^{\min\{\tau_i, t\}}\frac{1}{w_{s,i}} + \frac{t\max\{t-\tau_i, 0\}}{q_i\tau_i}\right)} + K^{\frac{v}{1+v}} t^{\frac{1}{1+v}}\max\{t/\tau_i, 1\}\right)
$$
$$
\cdot (\log(2K\log(T)/\zeta))^{\frac{3v}{3v+1}}.
$$

*Proof.* We fix some action $i \in [K]$. We first rewrite $\widehat{L}_{t,i}$ as $\widehat{L}_{t,i} = \sum_{s=1}^{t}\frac{\ell'_{s,i}I_{s,i}}{w_{s,i}}$. Note that for any $s > \tau_i$, we have $w_{s,i} = \frac{q_i\tau_i}{s}$, and $q_i \geqslant w_{s,i} \geqslant 1/K$ for any $s \leqslant \tau_i$.

We define $X_{s,i} := \frac{\ell'_{s,i}I_{s,i}}{w_{s,i}} - \mu'_{s,i}$ for any $1 \leqslant s \leqslant t$. If $t \leqslant t_{sw}$, then $X_{1,i}, \ldots, X_{t,i}$ forms a martingale difference sequence. To have an upper bound on $|X_{s,i}|$: 1) when $s \leqslant \tau_i$, we have

$$
|X_{s,i}| \leqslant M_{s,i}K + M_{s,i} \leqslant 2KM_{s,i},
$$

and 2) when $s > \tau_i$, we have

$$
|X_{s,i}| \leqslant \frac{M_{s,i}}{q_i \cdot \frac{\tau_i}{s}} + M_{s,i} \leqslant \frac{M_{s,i}}{\frac{1}{K} \cdot \frac{\tau_i}{s}} + M_{s,i} \leqslant 2KM_{s,i}\frac{s}{\tau_i}.
$$

Combining two cases, we get

$$
|X_{s,i}| \leqslant 2KM_{s,i}\max\{\frac{s}{\tau_i}, 1\} \leqslant 2KM_{t,i}\max\{\frac{t}{\tau_i}, 1\} = \frac{2ut^{\frac{1}{1+v}} K^{\frac{v}{1+v}}\max\{\frac{t}{\tau_i}, 1\}}{(\log(\log(T)/\zeta))^{\frac{1}{3v+1}}}.
\tag{45}
$$

Moreover, we have

$$\sum_{s=1}^{t} \mathbb{E}\left[(X_{s,i})^2 \big| \mathcal{F}_s\right] = \sum_{s=1}^{t} \left(\mathbb{E}\left[\frac{(\ell'_{s,i})^2 I_{s,i}}{(w_{s,i})^2} \bigg| \mathcal{F}_t\right] + (\mu'_{s,i})^2 - 2\mu'_{s,i}\mathbb{E}\left[\frac{\ell'_{s,i} I_{s,i}}{w_{s,i}} \bigg| \mathcal{F}_t\right]\right)$$

$$= \sum_{s=1}^{t} \mathbb{E}\left[\frac{(\ell'_{s,i})^2 I_{s,i}}{(w_{s,i})^2} \bigg| \mathcal{F}_t\right] - \sum_{s=1}^{t}(\mu'_{s,i})^2$$

$$\leqslant \sum_{s=1}^{t} \frac{1}{w_{s,i}} \mathbb{E}\left[\left|\ell'_{s,i}\right|^{1+v} (M_{s,i})^{1-v} \bigg| \mathcal{F}_t\right]$$

$$\leqslant u^{1+v}(M_{t,i})^{1-v}\left(\sum_{s=1}^{\min\{\tau_i,t\}} \frac{1}{w_{s,i}} + \frac{t\max\{t-\tau_i,0\}}{q_i\tau_i}\right)$$

$$= u^2(t/K)^{\frac{1-v}{1+v}}\left(\sum_{s=1}^{\min\{\tau_i,t\}} \frac{1}{w_{s,i}} + \frac{t\max\{t-\tau_i,0\}}{q_i\tau_i}\right)(\log(\log(T)/\zeta))^{\frac{v-1}{3v+1}}, \tag{46}$$

where the last inequality is because $M_{s,i}$ is non-decreasing in $s$.

By Lemma 11 we have that with probability at least $1 - \zeta$, for any $t \leqslant t_{\text{sw}}$

$$\sum_{s=1}^{t}\left(\frac{\ell'_{s,i} I_{s,i}}{w_{s,i}} - \mu'_{s,i}\right) \leqslant 4\sqrt{u^2(t/K)^{\frac{1-v}{1+v}}\left(\sum_{s=1}^{\min\{\tau_i,t\}} \frac{1}{w_{s,i}} + \frac{t\max\{t-\tau_i,0\}}{q_i\tau_i}\right)(\log(\log(T)/\zeta))^{\frac{4v}{3v+1}}}$$

$$+ 4ut^{\frac{1}{1+v}} K^{\frac{v}{1+v}} \max\{t/\tau_i,1\} (\log(\log(T)/\zeta))^{\frac{3v}{3v+1}}. \tag{47}$$

By taking an union bound over all all actions $i \in [K]$, we have with probability at least $1 - \zeta$ that

$$\left|\widehat{L}_{t,i} - \sum_{s=1}^{t}\mu'_{s,i}\right| \leqslant 4\sqrt{u^2(t/K)^{\frac{1-v}{1+v}}\left(\sum_{s=1}^{\min\{\tau_i,t\}} \frac{1}{w_{s,i}} + \frac{t\max\{t-\tau_i,0\}}{q_i\tau_i}\right)(\log(2K\log(T)/\zeta))^{\frac{4v}{3v+1}}}$$

$$+ 4ut^{\frac{1}{1+v}} K^{\frac{v}{1+v}} \max\{t/\tau_i,1\} (\log(2K\log(T)/\zeta))^{\frac{3v}{3v+1}}, \forall t \in [T], i \in [K]. \tag{48}$$

By Lemma 8, we have almost surely that

$$\sum_{s=1}^{t}\left|\mu_{s,i} - \mu'_{s,i}\right| \leqslant \sum_{s=1}^{t} u^{1+v}(M_{s,i})^{-v} \leqslant uK^{\frac{v}{1+v}}(\log(\log(T)/\zeta))^{\frac{v}{3v+1}}\sum_{s=1}^{t} s^{\frac{-v}{1+v}}$$

$$\leqslant 2uK^{\frac{v}{1+v}} t^{\frac{1}{1+v}} (\log(\log(T)/\zeta))^{\frac{v}{3v+1}}. \tag{49}$$

Combining two parts above, we get with probability at least $1 - \zeta$ that, for any action $i$ and round $t$,

$$\left|\widehat{L}_{t,i} - \sum_{s=1}^{t}\mu_{s,i}\right| \leqslant 6u\left(\sqrt{(t/K)^{\frac{1-v}{1+v}}\left(\sum_{s=1}^{\min\{\tau_i,t\}} \frac{1}{w_{s,i}} + \frac{t\max\{t-\tau_i,0\}}{q_i\tau_i}\right)} + K^{\frac{v}{1+v}} t^{\frac{1}{1+v}} \max\{t/\tau_i,1\}\right)$$

$$\cdot (\log(2K\log(T)/\zeta))^{\frac{3v}{3v+1}}. \tag{50}$$

$\square$

**Lemma 6** (Concentration on the number of pulls). *It holds with probability at least $1 - \zeta$ that, for any $i \in [K]$ and $t \in [T]$, if $t \leqslant t_{sw}$,*

$$N_{t,i} \leqslant q_i\tau_i(1 + \log t) + 4\sqrt{q_i\tau_i(1 + \log t)\log(K\log(T)/\zeta)} + 2\log(K\log(T)/\zeta).$$

*Proof of Lemma 6.* We first fix some action $i$ and round $t$. Recall that $N_{t,i} = \sum_{s=1}^{t} I_{s,i}$. Define $X_{s,i} := I_{s,i} - w_{s,i}$. If $t \leqslant t_{\text{sw}}$ then $X_{1,i}, \ldots, X_{t,i}$ forms a martingale difference sequence such that $|X_{s,i}| \leqslant 1$. Moreover, since $w_{s,i}$ is non-decreasing in $s$ when $s \leqslant \tau_i$ (and $w_{\tau_i,i} = q_i$), we have

$$\sum_{s=1}^{t} \mathbb{E}\left[(X_{s,i})^2 \big| \mathcal{F}_s\right] \leqslant \sum_{s=1}^{t} w_{s,i} \leqslant q_i \tau_i + \sum_{s=\tau_i+1}^{t} \frac{q_i \tau_i}{s} \leqslant q_i \tau_i (1 + \log t).$$

Therefore, by Lemma 11, it holds with probability at least $1 - \zeta$ that,

$$N_{t,i} - \sum_{s=1}^{t} w_{s,i} \leqslant 4\sqrt{q_i \tau_i (1 + \log t) \log(\log(T)/\zeta)} + 2\log(\log(T)/\zeta).$$

Taking union bounds over all actions $i \in [K]$ completes the proof. $\qquad\square$

**Lemma 7** (Concentration on the trimmed empirical-mean estimator). *It holds with probability at least $1 - \zeta$ that, for any $i \in [K]$ and $t \in [K, t_{sw}]$,*

$$\left| \widehat{\mu}_{t,i} - \frac{\sum_{s=1}^{t} \mu_{s,i} I_{s,i}}{N_{t,i}} \right| \leqslant 9u \left( \frac{\log(4KT^2/\zeta)}{N_{t,i}} \right)^{\frac{v}{1+v}}.$$

*Remark 9.* Note that in the special case of stochastic regime, the term $\frac{\sum_{s=1}^{t} \mu_{s,i} I_{s,i}}{N_{t,i}}$ is simply $\mu(i)$ (which is simply a scalar denoting the loss mean of action $i$ in the stochastic regime as defined in Section 2), and this lemma has been proven in Bubeck et al. (2013). However, we are not aware of how to extend the analysis of stochastic case to adversarial case.

*Proof.* Recall that $B_{s,i}$ is defined in Algorithm 2 as $B_{s,i} := u\left(\frac{N_{s,i}}{\log(2T/\zeta)}\right)^{\frac{1}{1+v}}, \forall s \in [T]$. Then we rewrite $\widehat{\mu}_{t,i}$ as

$$\widehat{\mu}_{t,i} = \frac{\sum_{s=1}^{t} \ell_{s,i} I_{s,i} \mathbb{I}\{|\ell_{s,i}| \leqslant B_{s,i}\}}{N_{t,i}}$$

and define

$$X_{s,i} := \left(\ell_{s,i} \mathbb{I}\{|\ell_{s,i}| \leqslant B_{s,i}\} - \mathbb{E}_{\ell_{s,i} \sim P_{s,i}}\left[\ell_{s,i} \mathbb{I}\{|\ell_{s,i}| \leqslant B_{s,i}\}\right]\right) I_{s,i}/u.$$

Moreover, in this lemma we need a slightly different filtration. We define

$$\mathcal{F}'_t := \sigma(a_1, \ell_{1,a_1}, \ldots, a_{t-1}, \ell_{t-1,a_{t-1}}, a_t).$$

Clearly, given any fixed round $t$ and action $i$, $X_{1,i}, \ldots, X_{t,i}$ form a martingale difference sequence adapted to $\mathcal{F}'_1, \ldots, \mathcal{F}'_t$ (since both $B_{s,i}$ and $I_{s,i}$ are deterministic conditioned on $\mathcal{F}'_s$), with

$$\max_{s \leqslant t} X_{s,i} \leqslant 2B_{t,i}/u = 2\left(\frac{N_{t,i}}{\log(2T/\zeta)}\right)^{\frac{1}{1+v}}$$

and

$$\begin{aligned}
\sum_{s=1}^{t} \mathbb{E}\left[(X_{s,i})^2 \big| \mathcal{F}'_s\right] &= \sum_{s:I_{s,i}=1} \mathbb{E}\left[(X_{s,i})^2 \big| \mathcal{F}'_s\right] \\
&\leqslant \sum_{s:I_{s,i}=1} \mathbb{E}\left[|\ell_{s,i}|^{1+v}(B_{s,i})^{1-v}/u^2 \big| \mathcal{F}'_s\right] \\
&\leqslant N_{t,i}(B_{t,i})^{1-v} u^{1+v}/u^2 = \left(\frac{N_{t,i}}{\log(2T/\zeta)}\right)^{\frac{1-v}{1+v}} N_{t,i}, \qquad (51)
\end{aligned}$$

where in the last inequality we utilize the fact that $B_{s,i}$ is non-decreasing in $s$.

Noting that

$$2\left(\frac{N_{t,i}}{\log(2T/\zeta)}\right)^{\frac{1}{1+v}} \leqslant 2\sqrt{\left(\frac{N_{t,i}}{\log(2T/\zeta)}\right)^{\frac{1-v}{1+v}} N_{t,i}} \leqslant 2\left(N_{t,i}\right)^{\frac{1}{1+v}} \leqslant 2T, \qquad (52)$$

we apply Lemma 12 and get with probability at least $1 - \zeta$ that, for any fixed $t$,

$$\sum_{s=1}^{t} X_{s,i} \leqslant 3\sqrt{(N_{t,i})^{\frac{2}{1+v}} (\log(2T/\zeta))^{\frac{v-1}{v+1}} \log(2T/\zeta)} + 2\left(\frac{N_{t,i}}{\log(2T/\zeta)}\right)^{\frac{1}{1+v}} \log(2T/\zeta)$$

$$= 5(N_{t,i})^{\frac{1}{1+v}} (\log(2T/\zeta))^{\frac{v}{1+v}} . \tag{53}$$

Taking a (two-sided) union bound over all rounds and actions, we get with probability at least $1 - \zeta$ that

$$\left|\sum_{s=1}^{t} X_{s,i}\right| \leqslant 5(N_{t,i})^{\frac{1}{1+v}} \left(\log(4KT^2/\zeta)\right)^{\frac{v}{1+v}} , \forall t \leqslant t_{\text{sw}}, i \in [K]. \tag{54}$$

Finally, we have

$$\left|\sum_{s=1}^{t} \left(\mathbb{E}_{\ell_{s,i} \sim P_{s,i}} [\ell_{s,i} \mathbb{I}\{|\ell_{s,i}| \leqslant B_{s,i}\}] - \mu_{s,i}\right) I_{s,i}\right|$$

$$\leqslant \sum_{s \leqslant t: I_{s,i}=1} \left|\mathbb{E}_{\ell_{s,i} \sim P_{s,i}} [\ell_{s,i} \mathbb{I}\{|\ell_{s,i}| \leqslant B_{s,i}\}] - \mu_{s,i}\right|$$

$$\overset{(a)}{\leqslant} u^{1+v} \sum_{N=1}^{N_{t,i}} u^{-v} \left(\frac{N}{\log(2T/\zeta)}\right)^{\frac{-v}{1+v}}$$

$$= u(\log(2T/\zeta))^{\frac{v}{1+v}} \sum_{N=1}^{N_{t,i}} N^{\frac{-v}{1+v}}$$

$$\leqslant u(\log(4KT^2/\zeta))^{\frac{v}{1+v}} (1+v)(N_{t,i}+1)^{\frac{1}{1+v}}$$

$$\leqslant 4u(\log(4KT^2/\zeta))^{\frac{v}{1+v}} (N_{t,i})^{\frac{1}{1+v}} , \tag{55}$$

where step (a) follows from Lemma 8 and the last step relies on the fact that $v \in (0,1]$.

Applying triangle inequality to Eqs. (54) and (55) and then dividing both sides by $N_{t,i}$ completes the proof. $\qquad\square$

## C.2 Logarithmic Regret in the Stochastic Regime

In this subsection, we provide the complete proof for logarithmic regret of Algorithm 2 in the stochastic regime.

Taking a union bound over Lemmas 5, 6, and 7, with probability at least $1 - \zeta$, for any $i \in [K]$ and $t \leqslant t_{\text{sw}}$, all of the following holds in either stochastic or adversarial regime (recall that $\beta = 12T^2 K \log(T)$):

$$\left|\widehat{L}_{t,i} - \sum_{s=1}^{t} \mu_{s,i}\right| \leqslant 6u \left(\sqrt{(t/K)^{\frac{1-v}{1+v}} \left(\sum_{s=1}^{\min\{\tau_i,t\}} \frac{1}{w_{s,i}} + \frac{t \max\{t - \tau_i, 0\}}{q_i \tau_i}\right)} + K^{\frac{v}{1+v}} t^{\frac{1}{1+v}} \max\{t/\tau_i, 1\}\right)$$

$$\cdot (\log(\beta/\zeta))^{\frac{3v}{3v+1}} , \tag{56}$$

$$N_{t,i} \leqslant q_i \tau_i (1 + \log t) + 4\sqrt{q_i \tau_i (1 + \log t) \log(\beta/\zeta)} + 2 \log(\beta/\zeta). \tag{57}$$

$$\left|\widehat{\mu}_{t,i} - \frac{\sum_{s=1}^{t} \mu_{s,i} I_{s,i}}{N_{t,i}}\right| \leqslant 9u \left(\frac{\log(\beta/\zeta)}{N_{t,i}}\right)^{\frac{v}{1+v}} . \tag{58}$$

Before we start to derive the regret bound, we simplify Eq. (56) a bit for convenience.

For any $t$ such that $\tau_i > t$ (i.e., action $i$ is still active by round $t$), we have $w_{s,i} \geqslant 1/K, \forall s \leqslant t$, and Eq. (56) implies that

$$
\begin{aligned}
\left| \widehat{L}_{t,i} - \sum_{s=1}^{t} \mu_{s,i} \right| &\leqslant 6u \left( \sqrt{(t/K)^{\frac{1-v}{1+v}} \sum_{s=1}^{t} \frac{1}{w_{s,i}}} + K^{\frac{v}{1+v}} t^{\frac{1}{1+v}} \right) (\log(\beta/\zeta))^{\frac{3v}{3v+1}} \\
&\leqslant 6u \left( \sqrt{(t/K)^{\frac{1-v}{1+v}} tK} + K^{\frac{v}{1+v}} t^{\frac{1}{1+v}} \right) (\log(\beta/\zeta))^{\frac{3v}{3v+1}} \\
&= 12uK^{\frac{v}{1+v}} t^{\frac{1}{1+v}} (\log(\beta/\zeta))^{\frac{3v}{3v+1}} = \mathrm{Width}(t),
\end{aligned}
\tag{59}
$$

otherwise (when $\tau_i \leqslant t$), we have

$$
\begin{aligned}
\left| \widehat{L}_{t,i} - \sum_{s=1}^{t} \mu_{s,i} \right| &\leqslant 6u \left( \sqrt{(t/K)^{\frac{1-v}{1+v}} K \frac{t^2}{\tau_i}} + K^{\frac{v}{1+v}} t^{1+\frac{1}{1+v}}/\tau_i \right) (\log(\beta/\zeta))^{\frac{3v}{3v+1}} \\
&= 6u \left( tK^{\frac{v}{1+v}} \sqrt{\frac{t^{\frac{1-v}{1+v}}}{\tau_i}} + K^{\frac{v}{1+v}} t^{1+\frac{1}{1+v}}/\tau_i \right) (\log(\beta/\zeta))^{\frac{3v}{3v+1}} \\
&\leqslant 12uK^{\frac{v}{1+v}} t^{1+\frac{1}{1+v}} (\log(\beta/\zeta))^{\frac{3v}{3v+1}}/\tau_i = \mathrm{Width}(t) \cdot t/\tau_i,
\end{aligned}
\tag{60}
$$

where in the first inequality we utilize the facts that $q_i \geqslant 1/K$ and

$$
\tau_i + \frac{t(t-\tau_i)}{\tau_i} \leqslant \frac{t^2}{\tau_i},
\tag{61}
$$

and the second inequality is because

$$
\sqrt{\frac{t^{\frac{1-v}{1+v}}}{\tau_i}} \leqslant \sqrt{\frac{t^{\frac{1-v}{1+v}}}{\tau_i} \cdot \frac{t}{\tau_i}} = \frac{t^{\frac{1}{1+v}}}{\tau_i}.
\tag{62}
$$

Combining two cases (whether $\tau_i > t$ or not), Eq. (56) implies that

$$
\left| \widehat{L}_{t,i} - \sum_{s=1}^{t} \mu_{s,i} \right| \leqslant \mathbb{I}\{i \in A_t\}\mathrm{Width}(t) + \mathbb{I}\{i \notin A_t\}\mathrm{Width}(t)\frac{t}{\tau_i}.
\tag{63}
$$

To show the regret bound, we first show that, in the stochastic regime, when all the three above events hold, we will never start to run Algorithm 1, i.e., tests (6)-(8) always fail (for all $t < T$).

**Test (6) always fails.** This is simply implied by Eqs. (63) and (58) together with a triangle inequality.

**Test (7) always fails.** We first show that test (5) is never satisfied for action $i^*$, so we have $i^* \in A_t, \forall t \leqslant t_{\mathrm{sw}}$. To see this, for actions $i, i^* \in A_{t-1}$, by Eq. (63) we must have

$$
\begin{aligned}
\widehat{L}_{t,i^*} - \widehat{L}_{t,i} &= (\widehat{L}_{t,i^*} - t \cdot \mu(i^*)) - (\widehat{L}_{t,i} - t \cdot \mu(i)) + t \cdot (\mu(i^*) - \mu(i)) \\
&\leqslant 2\mathrm{Width}(t) - t\Delta_i < c_1\mathrm{Width}(t),
\end{aligned}
\tag{64}
$$

which means that $i^*$ is never eliminated from $A_t$ and hence stays active (due to test (5)).

Moreover, for any action $i \notin A_t$, it must be deactivated at some round no later than $t$, so we have $\tau_i \leqslant t$ and that test (5) is satisfied at round $\tau_i$ (and is not satisfied at round $\tau_i - 1$). Let $j_t^* \in \mathrm{argmin}_{j \in A_{t-1}} \widehat{L}_{t,j}$.

Therefore, looking at round $\tau_i$ (and any action $i \neq i^*$), we have

$$
\begin{aligned}
c_1\mathrm{Width}(\tau_i) &\overset{(a)}{<} \widehat{L}_{\tau_i,i} - \widehat{L}_{\tau_i,j_{\tau_i}^*} \\
&= (\widehat{L}_{\tau_i,i} - \tau_i \cdot \mu(i)) - (\widehat{L}_{\tau_i,j_{\tau_i}^*} - \tau_i \cdot \mu(j_{\tau_i}^*)) + \tau_i \cdot (\mu(i) - \mu(j_{\tau_i}^*)) \\
&\overset{(b)}{\leqslant} 2\mathrm{Width}(\tau_i) + \tau_i \cdot (\mu(i) - \mu(i^*)) = 2\mathrm{Width}(\tau_i) + \tau_i\Delta_i,
\end{aligned}
\tag{65}
$$

where step (a) is from test (5) (which should hold now) and step (b) is due to Eq. (63) and the optimality of $i^*$ (i.e., $\mu(i) - \mu(j^*_{\tau_i}) \leqslant \mu(i) - \mu(i^*) = \Delta_i$).

Moreover, looking at round $\tau_i - 1$ (and any action $i \neq i^*$), we have

$$
\begin{aligned}
(\tau_i - 1)\Delta_i - 2\text{Width}(\tau_i - 1) &\overset{(a)}{\leqslant} (\widehat{L}_{\tau_i-1,i} - (\tau_i - 1)\cdot\mu(i)) - (\widehat{L}_{\tau_i-1,i^*} - (\tau_i - 1)\cdot\mu(i^*)) \\
&\quad + (\tau_i - 1)\cdot(\mu(i) - \mu(i^*)) \\
&= \widehat{L}_{\tau_i-1,i} - \widehat{L}_{\tau_i-1,i^*} \\
&\overset{(b)}{\leqslant} \widehat{L}_{\tau_i-1,i} - \widehat{L}_{\tau_i-1,j^*_{\tau_i-1}} \\
&\overset{(c)}{\leqslant} c_1\text{Width}(\tau_i - 1),
\end{aligned}
\tag{66}
$$

where step (a) follows from the good event in Eq. (63), step (b) is because $i^* \in A_{\tau_i-2}$, and step (c) is due to test (5) (which now should not hold for action $i$ since it is still active).

Now we can show that test (7) is never satisfied since in any round $t$ for any $i \notin A_t$:

$$
\begin{aligned}
(\widehat{L}_{t,i} - \min_{j\in A_t}\widehat{L}_{t,j})/t &\leqslant (\widehat{L}_{t,i} - \min_{j\in A_{t-1}}\widehat{L}_{t,j})/t \\
&= (\widehat{L}_{t,i}/t - \mu(i)) - (\widehat{L}_{t,j^*_t}/t - \mu(j^*_t)) + (\mu(i) - \mu(j^*_t)) \\
&\leqslant \text{Width}(t)/\tau_i + \text{Width}(t)/t + \Delta_i \\
&\leqslant 2\text{Width}(t)/(\tau_i - 1) + \Delta_i \\
&\leqslant (2 + c_1 + 2)\text{Width}(t)/(\tau_i - 1),
\end{aligned}
\tag{67}
$$

where the last step is from Eq. (66).

**Test (8) always fails.** Since $i^* \in A_t$, for any action $i \notin A_t$, we have

$$
\begin{aligned}
(\widehat{L}_{t,i} - \min_{j\in A_t}\widehat{L}_{t,j})/t &\geqslant (\widehat{L}_{t,i} - \widehat{L}_{t,i^*})/t \\
&= (\widehat{L}_{t,i}/t - \mu(i)) - (\widehat{L}_{t,i^*}/t - \mu(i^*)) + (\mu(i) - \mu(i^*)) \\
&\geqslant -2\text{Width}(t)/\tau_i + \Delta_i \\
&\geqslant (c_1 - 2 - 2)\text{Width}(t)/\tau_i,
\end{aligned}
\tag{68}
$$

where the first step is due to $i^* \in A_t$ and the last step is from Eq. (65).

**Putting things together.** Now we show two intermediate results, followed by bounding the regret.

First, Eq. (66) implies that

$$
\begin{aligned}
\Delta_i &\leqslant (2 + c_1)\text{Width}(\tau_i - 1)/(\tau_i - 1) \\
&= (2 + c_1)12uK^{\frac{v}{1+v}}(\tau_i - 1)^{\frac{-v}{1+v}}\left(\log(\beta/\zeta)\right)^{\frac{3v}{3v+1}} \\
&\leqslant (2 + c_1)12uK^{\frac{v}{1+v}}(\tau_i/2)^{\frac{-v}{1+v}}\left(\log(\beta/\zeta)\right)^{\frac{3v}{3v+1}} \\
&\leqslant (2 + c_1)24uK^{\frac{v}{1+v}}(\tau_i)^{\frac{-v}{1+v}}\left(\log(\beta/\zeta)\right)^{\frac{3v}{3v+1}},
\end{aligned}
\tag{69}
$$

which after rearranging implies that

$$
\tau_i = O\left(u^{1+\frac{1}{v}}K\left(\log(\beta/\zeta)\right)^{1+\frac{2}{3v+1}}/(\Delta_i)^{1+\frac{1}{v}}\right).
\tag{70}
$$

Due to the definition of $q_i$, we have

$$
\sum_{i=1}^{K} q_i \leqslant \sum_{i=1}^{K}\frac{1}{K - i + 1} \leqslant 1 + \log K.
\tag{71}
$$

To see this, let $i'$ denote the $i$-th earliest action that is deactivated, we have $q_{i'} \leqslant \frac{1}{K-i'+1}$ due to the algorithm design.

Now we have everything needed to arrive at the final result. By the definition of pseudo-regret in stochastic bandits, we have

$$
\begin{aligned}
R_T &= \sum_{i:\Delta_i>0} \Delta_i N_{T,i} \\
&\overset{(a)}{\leqslant} \sum_{i:\Delta_i>0} \Delta_i \left( q_i \tau_i (1 + \log T) + 4\sqrt{q_i \tau_i (1 + \log T) \log(\beta/\zeta)} + 2\log(\beta/\zeta) \right) \\
&\leqslant \sum_{i:\Delta_i>0} \Delta_i \left( 2q_i \tau_i (1 + \log T) + 6\log(\beta/\zeta) \right) \\
&\overset{(b)}{\leqslant} O \left( \sum_{i:\Delta_i>0} \Delta_i q_i \log(T) \cdot u^{1+\frac{1}{v}} K \left( \log(\beta/\zeta) \right)^{1+\frac{2}{3v+1}} /(\Delta_i)^{1+\frac{1}{v}} + \log(\beta/\zeta) \sum_{i:\Delta_i>0} \Delta_i \right) \\
&\overset{(c)}{\leqslant} O \left( u^{1+\frac{1}{v}} K \log(T) \left( \log(\beta/\zeta) \right)^3 /\Delta^{\frac{1}{v}} \cdot \sum_{i:\Delta_i>0} q_i + \log(\beta/\zeta) \sum_{i:\Delta_i>0} \Delta_i \right) \\
&\leqslant O \left( K \log(T) \log(K) \left( \log(\beta/\zeta) \right)^3 (u^{1+\frac{1}{v}}/\Delta)^{\frac{1}{v}} \right),
\end{aligned}
\tag{72}
$$

where step (a) follows from Eq. (57), step (b) is due to Eq. (70), and step (c) is from Eq. (71). Choosing $\zeta = 1/T < 1/e$, we have

$$
\begin{aligned}
\mathbb{E}\left[R_T\right] &\leqslant O \left( K \log(T) \log(K)(\log T)^3 (u^{1+\frac{1}{v}}/\Delta)^{\frac{1}{v}} \right) + \frac{1}{T} uT \\
&= O \left( K \log(K)(\log T)^4 (u^{1+\frac{1}{v}}/\Delta)^{\frac{1}{v}} \right).
\end{aligned}
\tag{73}
$$

### C.3 Optimal Worst-case Regret in the Adversarial Regime

In this subsection, we provide the complete proof for (near-)optimal worst-case regret of Algorithm 2 in the adversarial regime.

Let $i_t^* := \operatorname{argmin}_{i \in [K]} \sum_{s=1}^t \mu_{s,i}$ and $I_t^* := \operatorname{argmin}_{i \in A_t} \sum_{s=1}^t \mu_{s,i}$. We first show that $i_{t_{\text{sw}}-1}^* \in A_{t_{\text{sw}}-1}$.

For any action $i \notin A_{t_{\text{sw}}-1}$, we have $\tau_i \leqslant t_{\text{sw}} - 1$ and test (8) is not satisfied for $i$ at round $t_{\text{sw}} - 1$ (since the algorithm switch has not happened). We get

$$
\begin{aligned}
& \sum_{s=1}^{t_{\text{sw}}-1} \mu_{s,i} - \sum_{s=1}^{t_{\text{sw}}-1} \mu_{s,I_{t_{\text{sw}}-1}^*} \\
&= \left( \sum_{s=1}^{t_{\text{sw}}-1} \mu_{s,i} - \widehat{L}_{t_{\text{sw}}-1,i} \right) + \left( \widehat{L}_{t_{\text{sw}}-1,I_{t_{\text{sw}}-1}^*} - \sum_{s=1}^{t_{\text{sw}}-1} \mu_{s,I_{t_{\text{sw}}-1}^*} \right) + \left( \widehat{L}_{t_{\text{sw}}-1,i} - \widehat{L}_{t_{\text{sw}}-1,I_{t_{\text{sw}}-1}^*} \right) \\
&\overset{(a)}{>} \frac{-(t_{\text{sw}}-1)}{\tau_i} \text{Width}(t_{\text{sw}}-1) - \text{Width}(t_{\text{sw}}-1) + (c_1 - 4) \frac{t_{\text{sw}}-1}{\tau_i} \text{Width}(t_{\text{sw}}-1) \\
&\geqslant (c_1 - 6) \frac{t_{\text{sw}}-1}{\tau_i} \text{Width}(t_{\text{sw}}-1) \geqslant 0,
\end{aligned}
\tag{74}
$$

where step (a) is due to Eq. (63) (applied twice) and test (8). Therefore, we must have $i_{t_{\text{sw}}-1}^* \in A_{t_{\text{sw}}-1}$ (which further implies $i_{t_{\text{sw}}-1}^* \in A_s, \forall s \leqslant t_{\text{sw}} - 1$) since $\sum_{s=1}^{t_{\text{sw}}-1} \mu_{s,i_{t_{\text{sw}}-1}^*} - \sum_{s=1}^{t_{\text{sw}}-1} \mu_{s,I_{t_{\text{sw}}-1}^*} \leqslant 0$ by the definition, otherwise a contradiction is incurred.

Now we can bound the cumulative regret up to the algorithm switch. Specifically, we rewrite the regret as

$$\sum_{s=1}^{t_{\mathrm{sw}}-1} \mu_{s,a_t} - \sum_{s=1}^{t_{\mathrm{sw}}-1} \mu_{s,i_{t_{\mathrm{sw}}-1}^*} = \sum_{s=1}^{t_{\mathrm{sw}}-1} \sum_{i=1}^{K} \mu_{s,i} I_{s,i} - \sum_{s=1}^{t_{\mathrm{sw}}-1} \mu_{s,i_{t_{\mathrm{sw}}-1}^*}$$

$$= \sum_{i=1}^{K} N_{t_{\mathrm{sw}}-1,i} \left( \frac{\sum_{s=1}^{t_{\mathrm{sw}}-1} \mu_{s,i} I_{s,i}}{N_{t_{\mathrm{sw}}-1,i}} - \frac{\sum_{s=1}^{t_{\mathrm{sw}}-1} \mu_{s,i_{t_{\mathrm{sw}}-1}^*}}{t_{\mathrm{sw}}-1} \right)$$

$$= \sum_{i=1}^{K} N_{t_{\mathrm{sw}}-1,i} \left( \underbrace{\frac{\sum_{s=1}^{t_{\mathrm{sw}}-1} \mu_{s,i} I_{s,i}}{N_{t_{\mathrm{sw}}-1,i}} - \frac{\widehat{L}_{t_{\mathrm{sw}}-1,i}}{t_{\mathrm{sw}}-1}}_{\text{Part A}} + \underbrace{\frac{\widehat{L}_{t_{\mathrm{sw}}-1,i} - \widehat{L}_{t_{\mathrm{sw}}-1,i_{t_{\mathrm{sw}}-1}^*}}{t_{\mathrm{sw}}-1}}_{\text{Part B}} \right)$$

$$+ \underbrace{\widehat{L}_{t_{\mathrm{sw}}-1,i_{t_{\mathrm{sw}}-1}^*} - \sum_{s=1}^{t_{\mathrm{sw}}-1} \mu_{s,i_{t_{\mathrm{sw}}-1}^*}}_{\text{Part C}}. \tag{75}$$

In the rewriting above, we use nothing but the simple fact that $\sum_{i=1}^{K} N_{t_{\mathrm{sw}}-1,i} = t_{\mathrm{sw}} - 1$. We now bound each of the three terms (Part A, Part B, and Part C) separately.

**Bounding Part A.** We first rewrite Part A as

$$\text{Part A} = \left( \frac{\sum_{s=1}^{t_{\mathrm{sw}}-1} \mu_{s,i} I_{s,i}}{N_{t_{\mathrm{sw}}-1,i}} - \widehat{\mu}_{t_{\mathrm{sw}}-1,i} \right) + \left( \widehat{\mu}_{t_{\mathrm{sw}}-1,i} - \frac{\widehat{L}_{t_{\mathrm{sw}}-1,i}}{t_{\mathrm{sw}}-1} \right). \tag{76}$$

By Eq. (58), we have

$$\frac{\sum_{s=1}^{t_{\mathrm{sw}}-1} \mu_{s,i} I_{s,i}}{N_{t_{\mathrm{sw}}-1,i}} - \widehat{\mu}_{t_{\mathrm{sw}}-1,i} \leqslant 9u \left( \frac{\log(\beta/\zeta)}{N_{t_{\mathrm{sw}}-1,i}} \right)^{\frac{v}{1+v}}. \tag{77}$$

By test (6) (which does not hold now), we have

$$\widehat{\mu}_{t_{\mathrm{sw}}-1,i} - \frac{\widehat{L}_{t_{\mathrm{sw}}-1,i}}{t_{\mathrm{sw}}-1} \leqslant 9u \left( \frac{\log(\beta/\zeta)}{N_{t_{\mathrm{sw}}-1,i}} \right)^{\frac{v}{1+v}} + \frac{\text{Width}(t_{\mathrm{sw}}-1)}{\tau_i - 1}. \tag{78}$$

Therefore, we can conclude that

$$\text{Part A} \leqslant 18u \left( \frac{\log(\beta/\zeta)}{N_{t_{\mathrm{sw}}-1,i}} \right)^{\frac{v}{1+v}} + \frac{\text{Width}(t_{\mathrm{sw}}-1)}{\tau_i - 1}. \tag{79}$$

**Bounding Part B.** Since $i_{t_{\mathrm{sw}}-1}^* \in A_{t_{\mathrm{sw}}-1}$ as we have shown, for any action $i \notin A_{t_{\mathrm{sw}}-1}$, due to test (7) (which is not satisfied at round $t_{\mathrm{sw}} - 1$), we have

$$\frac{\widehat{L}_{t_{\mathrm{sw}}-1,i} - \widehat{L}_{t_{\mathrm{sw}}-1,i_{t_{\mathrm{sw}}-1}^*}}{t_{\mathrm{sw}}-1} \leqslant \frac{\widehat{L}_{t_{\mathrm{sw}}-1,i} - \min_{j \in A_{t_{\mathrm{sw}}-1}} \widehat{L}_{t_{\mathrm{sw}}-1,j}}{t_{\mathrm{sw}}-1} \leqslant (c_1 + 4) \frac{\text{Width}(t_{\mathrm{sw}}-1)}{\tau_i - 1}. \tag{80}$$

For any other action $i \in A_{t_{\mathrm{sw}}-1}$, due to test (5), we have

$$\frac{\widehat{L}_{t_{\mathrm{sw}}-1,i} - \widehat{L}_{t_{\mathrm{sw}}-1,i_{t_{\mathrm{sw}}-1}^*}}{t_{\mathrm{sw}}-1} \leqslant \frac{\widehat{L}_{t_{\mathrm{sw}}-1,i} - \min_{j \in A_{t_{\mathrm{sw}}-2}} \widehat{L}_{t_{\mathrm{sw}}-1,j}}{t_{\mathrm{sw}}-1} \leqslant c_1 \frac{\text{Width}(t_{\mathrm{sw}}-1)}{t_{\mathrm{sw}}-1} = c_1 \frac{\text{Width}(t_{\mathrm{sw}}-1)}{\tau_i - 1}. \tag{81}$$

Combining these two cases, we can claim that for any action $i \in [K]$,

$$\text{Part B} = \frac{\widehat{L}_{t_{\mathrm{sw}}-1,i} - \widehat{L}_{t_{\mathrm{sw}}-1,i_{t_{\mathrm{sw}}-1}^*}}{t_{\mathrm{sw}}-1} \leqslant (c_1 + 4) \frac{\text{Width}(t_{\mathrm{sw}}-1)}{\tau_i - 1}. \tag{82}$$

**Bounding Part C.** Simply due to Eq. (63), since $i_{t_{\mathrm{sw}}-1}^* \in A_{t_{\mathrm{sw}}-1}$, we have

$$\text{Part C} = \widehat{L}_{t_{\mathrm{sw}}-1,i_{t_{\mathrm{sw}}-1}^*} - \sum_{s=1}^{t_{\mathrm{sw}}-1} \mu_{s,i_{t_{\mathrm{sw}}-1}^*} \leqslant c_1 \text{Width}(t_{\mathrm{sw}}-1). \tag{83}$$

**Putting three parts together**. Putting the three parts together, we get

$$\sum_{s=1}^{t_{\text{sw}}-1} \mu_{s,a_t} - \sum_{s=1}^{t_{\text{sw}}-1} \mu_{s,i^*} \leqslant \sum_{s=1}^{t_{\text{sw}}-1} \mu_{s,a_t} - \sum_{s=1}^{t_{\text{sw}}-1} \mu_{s,i^*_{t_{\text{sw}}-1}}$$

$$= O\left(\sum_{i=1}^{K} N_{t_{\text{sw}}-1,i} \cdot u \left(\frac{\log(\beta/\zeta)}{N_{t_{\text{sw}}-1,i}}\right)^{\frac{v}{1+v}}\right) + O\left(\sum_{i=1}^{K} N_{t_{\text{sw}}-1,i} \frac{\text{Width}(t_{\text{sw}}-1)}{\tau_i - 1}\right)$$

$$+ O\left(\text{Width}(t_{\text{sw}}-1)\right)$$

$$= O\left(u \sum_{i=1}^{K} N_{t_{\text{sw}}-1,i} \left(\frac{\log(\beta/\zeta)}{N_{t_{\text{sw}}-1,i}}\right)^{\frac{v}{1+v}}\right) + O\left(\text{Width}(t_{\text{sw}}-1)\right)$$

$$+ O\left(\sum_{i=1}^{K} (q_i \tau_i (1 + \log T)) \frac{uK^{\frac{v}{1+v}}(t_{\text{sw}}-1)^{\frac{1}{1+v}}(\log(\beta/\zeta))^{\frac{3v}{3v+1}}}{\tau_i - 1}\right)$$

$$+ O\left(\sum_{i=1}^{K} \log(\beta/\zeta) \frac{uK^{\frac{v}{1+v}}(t_{\text{sw}}-1)^{\frac{1}{1+v}}(\log(\beta/\zeta))^{\frac{3v}{3v+1}}}{\tau_i - 1}\right), \qquad (84)$$

where in the last step we apply Lemma 6 to bound $N_{t_{\text{sw}}-1,i}$ in the term $\sum_{i=1}^{K} N_{t_{\text{sw}}-1,i} \frac{\text{Width}(t_{\text{sw}}-1)}{\tau_i - 1}$. Now we bound each of the four terms one by one.

**Bounding the first term.** Applying Jensen's inequality, we get

$$\sum_{i=1}^{K} (N_{t_{\text{sw}}-1,i})^{\frac{1}{1+v}} \leqslant K \cdot \left(\sum_{i=1}^{K} \frac{N_{t_{\text{sw}}-1,i}}{K}\right)^{\frac{1}{1+v}} = K^{\frac{v}{1+v}}(t_{\text{sw}}-1)^{\frac{1}{1+v}}, \qquad (85)$$

and the first term is bounded as

$$u \sum_{i=1}^{K} N_{t_{\text{sw}}-1,i} \left(\frac{\log(\beta/\zeta)}{N_{t_{\text{sw}}-1,i}}\right)^{\frac{v}{1+v}} = O\left(uK^{\frac{v}{1+v}}(t_{\text{sw}}-1)^{\frac{1}{1+v}}(\log(\beta/\zeta))^{\frac{v}{1+v}}\right). \qquad (86)$$

**Bounding the second term.** Simply plugging in the definition of Width$(\cdot)$, we have

$$\text{Width}(t_{\text{sw}}-1) = O\left(uK^{\frac{v}{1+v}}(t_{\text{sw}}-1)^{\frac{1}{1+v}}(\log(\beta/\zeta))^{\frac{3v}{1+3v}}\right).$$

**Bounding the third term.** By the fact that $\sum_{i=1}^{K} q_i = O(\log K)$, we have

$$\sum_{i=1}^{K} (q_i \tau_i (1 + \log T)) \frac{uK^{\frac{v}{1+v}}(t_{\text{sw}}-1)^{\frac{1}{1+v}}(\log(\beta/\zeta))^{\frac{3v}{3v+1}}}{\tau_i - 1}$$

$$= O\left(\log(K) \log(T) uK^{\frac{v}{1+v}}(t_{\text{sw}}-1)^{\frac{1}{1+v}}(\log(\beta/\zeta))^{\frac{3v}{3v+1}}\right). \qquad (87)$$

**Bounding the forth term.** Notice that in the algorithm design, each action will be pulled once in the initialization, and we clearly have $\tau_i \geqslant K + 1$. Therefore, we have

$$\sum_{i=1}^{K} \log(\beta/\zeta) \frac{uK^{\frac{v}{1+v}}(t_{\text{sw}}-1)^{\frac{1}{1+v}}(\log(\beta/\zeta))^{\frac{3v}{3v+1}}}{\tau_i - 1}$$

$$\leqslant \sum_{i=1}^{K} \log(\beta/\zeta) \frac{uK^{\frac{v}{1+v}}(t_{\text{sw}}-1)^{\frac{1}{1+v}}(\log(\beta/\zeta))^{\frac{3v}{3v+1}}}{K}$$

$$= O\left(uK^{\frac{v}{1+v}}(t_{\text{sw}}-1)^{\frac{1}{1+v}}(\log(\beta/\zeta))^{1+\frac{3v}{3v+1}}\right). \qquad (88)$$

Combining the bounds on these four terms, we have

$$\sum_{s=1}^{t_{\text{sw}}-1} \mu_{s,a_t} - \sum_{s=1}^{t_{\text{sw}}-1} \mu_{s,i^*} = O\left(\log(K)\log(T)uK^{\frac{v}{1+v}}(t_{\text{sw}}-1)^{\frac{1}{1+v}}\left(\log(\beta/\zeta)\right)^{1+\frac{3v}{3v+1}}\right). \quad (89)$$

The regret incurred starting from round $t_{\text{sw}}+1$ is taken care of by Algorithm 1 (and the regret guarantee is given by Theorem 1). By taking a union bound, we have with probability at least $1-\zeta$ that

$$R_T = O\left(uK^{\frac{v}{1+v}}T^{\frac{1}{1+v}}\log(K)\log(T)(\log(\beta/\zeta))^{1+\frac{3v}{3v+1}}\right) + O\left(uK^{\frac{v}{1+v}}T^{\frac{1}{1+v}}(\log T)^2\log(\beta/\zeta)\right)$$
$$= O\left(uK^{\frac{v}{1+v}}T^{\frac{1}{1+v}}\log(K)(\log T)^2(\log(\beta/\zeta))^2\right). \quad (90)$$

Choosing $\zeta = 1/T$, we have

$$\mathbb{E}\left[R_T\right] = O\left(uK^{\frac{v}{1+v}}T^{\frac{1}{1+v}}\log(K)(\log T)^4\right) + 2\frac{1}{T}uT = O\left(uK^{\frac{v}{1+v}}T^{\frac{1}{1+v}}\log(K)(\log T)^4\right). \quad (91)$$

## D  Heavy-tailed Adversarial Bandits with Huber Contamination

This section is dedicated to the adversarial regime, with the additional setup that the bandit feedback could be contaminated in the Huber model (Huber, 1996). In the stochastic MAB, Huber contamination has been studied in Guan et al. (2020); Agrawal et al. (2024); Wu et al. (2024). We first formally define the problem setup, and then present the algorithm design and the regret analysis. From the regret analysis one could readily see why removing Assumption 2 is necessary for a near-optimal regret upper bound. Lastly, we provide a matching lower bound which suggests that we obtain the near-optimal worst-case regret guarantee.

### D.1  Problem Setup

The learning algorithm and environment perform the following interactions repeatedly in round $t = 1, \ldots, T$:

1. The algorithm samples action $a_t$ from $[K]$ via $a_t \sim w_t := (w_{t,1}, \ldots, w_{t,K}) \in \Omega$, i.e., the probability of sampling action $i \in [K]$ is $w_{t,i}$. The environment draws loss $\ell_{t,i}$ from "clean" distribution $P_{t,i}$ satisfying Assumption 1 for every action $i \in [K]$.

2. Let $\bar{\ell}_{t,a_t}$ denote the feedback revealed to the algorithm associated with $a_t$. With probability $\alpha \in (0, 1]$, the algorithm observes the contaminated feedback. That is, it observes $\bar{\ell}_{t,a_t} = \widetilde{\ell}_{t,i}$, which is generated from an *arbitrary* "bad" distribution $Q_{t,i}$. With probability $(1 - \alpha)$, it observes the "clean" loss $\bar{\ell}_{t,a_t} = \ell_{t,i}$.

3. The algorithm determines $w_{t+1}$ based on all the revealed history so far.

The goal of the learning algorithm is to minimize the regret in the presence of contaminated feedback. For the ease of presentation, we consider the weaker expected pseudo-regret (rather than the high-probability version), which is defined as:

$$\mathbb{E}[R_T] := \mathbb{E}\left[\sum_{t=1}^T \langle w_t - y^{i^*}, \mu_t \rangle\right], \quad (92)$$

where the optimal action $i^*$ and loss mean $\mu_t$ are both still with respect to the "clean" distributions, i.e., $\mu_{t,i} := \mathbb{E}_{\ell_{t,i} \sim P_{t,i}}[\ell_{t,i}]$ and $i^* \in \operatorname{argmin}_{i \in [K]} \sum_{t=1}^T \mu_{t,i}$, while the expected pseudo-regret additionally includes the randomness from the contamination. Similar to all previous works, we assume that the "contamination level" $\alpha$ is known to the algorithm.

Assumption 2 adapted to the contaminated case (to bound CONTRIMERR) becomes the following.

**Assumption 3** (Truncated non-negative losses in the contaminated case). Given any fixed $M > 0$, the loss distributions of the optimal action $i^*$ satisfy that $\mathbb{E}_{\ell_{t,i^*} \sim P_{t,i^*}^\alpha}[\ell_{t,i^*} \cdot \mathbb{I}\{|\ell_{t,i^*}| > M\}] \geqslant 0, \forall t \in [T]$.

Importantly, even if there is some prior knowledge regarding the "clean" distribution $P_{t,i}$, due to the *arbitrary* "bad" distribution component contained in $P_{t,i*}^\alpha$, the adapted "truncated non-negativity" may not hold any more and one may not obtain any meaningful regret guarantee. However, by taking advantage of the log-barrier regularizer, we naturally handle all possible "bad" distribution $Q_{t,i}$. In the following two subsections, we the details of regret upper bound and lower bound, respectively.

## D.2 Regret Guarantee and Analysis

To obtain near-optimal high-probability regret guarantee again Huber contamination, we simply need to run our Algorithm 1 with different initial learning rate and trimming threshold, both of which now further depend on the contamination level. We formally state the theoretical guarantee below.

**Theorem 3.** *For any failure probability $\zeta$ and contamination level $\alpha > 0$, by choosing initial learning rate*

$$\eta = \frac{1}{u}\left(\frac{K\log(T)}{T}\right)^{\frac{2+v}{2+2v}}\left(\frac{1}{\alpha}\right)^{\frac{v}{2+2v}}$$

*and trimming threshold*

$$M_{t,i} = \min\{\frac{u}{\alpha^{\frac{1}{1+v}}}, \left(\frac{u^{1+v}}{\eta\alpha}\right)^{\frac{1}{2+v}}, \frac{1}{110\eta\log(T)\log(40T^2/\zeta)}\},$$

*Algorithm 1 ensures that with probability at least $1 - \zeta$,*

$$R_T = O\left(u\alpha^{\frac{v}{1+v}}T + uK^{\frac{v}{1+v}}T^{\frac{1}{1+v}}(\log(T))^{1.5}\log(T/\zeta)\right).$$

*By further choosing $\zeta = 1/T$, Algorithm 1 ensures that*

$$\mathbb{E}[R_T] = O\left(u\alpha^{\frac{v}{1+v}}T + uK^{\frac{v}{1+v}}T^{\frac{1}{1+v}}(\log(T))^{2.5}\right).$$

We first rewrite regret as

$$R_T = \underbrace{\sum_{t\in N_Q}(\mu_{t,a_t} - \overline{\ell}'_{t,a_t}) - \sum_{t\in N_Q}(\mu_{t,i*} - \overline{\ell}'_{t,i*})}_{\text{CONTRIMERR I}} + \underbrace{\sum_{t\in N_P}(\mu_{t,a_t} - \overline{\ell}'_{t,a_t}) - \sum_{t\in N_P}(\mu_{t,i*} - \overline{\ell}'_{t,i*})}_{\text{CONTRIMERR II}}$$

$$+ \underbrace{\sum_{t=1}^{T}(\overline{\ell}'_{t,a_t} - \overline{\ell}'_{t,i*})}_{\text{CONTRIMREG}}, \tag{93}$$

where $N_Q$ is the set containing all round indices in which the observation is contaminated (i.e., $\overline{\ell}_{t,a_t} = \widetilde{\ell}_{t,a_t} \sim Q_{t,a_t}$), and $N_P = [T]\backslash N_Q$ (i.e., those rounds in which $\overline{\ell}_{t,a_t} = \ell_{t,a_t} \sim P_{t,a_t}$).

Similar to the uncontaminated case, it is sufficient to choose some fixed trimming threshold $M_{t,i} = M$.

### D.2.1 Bounding CONTRIMERR I

Recall that $|N_Q|$ denotes the total number of rounds when the feedback is contaminated, which is exactly the sum of $T$ independent random variables from $\text{Ber}(\alpha)$. By standard concentration results (e.g., Lemma 10), we have with probability at least $1 - \zeta$ that

$$|N_Q| \leqslant \alpha T + \sqrt{2T\alpha(1-\alpha)\log(1/\zeta)} + \frac{2\log(1/\zeta)}{3} \leqslant 2\alpha T + \frac{4\log(1/\zeta)}{3}, \tag{94}$$

where the last step is again due to the fact that $\sqrt{x_1 x_2} \leqslant \frac{x_1+x_2}{2}, \forall x_1, x_2 \geqslant 0$.

For any fixed trimming threshold $M$, it holds almost surely that

$$\mu_{t,i} - \overline{\ell}'_{t,i} = \mu_{t,i} - \mu'_{t,i} + \mu'_{t,i} - \overline{\ell}_{t,i} \leqslant u^{1+v}M^{-v} + 2M, \tag{95}$$

which implies that, with probability at least $1 - \zeta$, we have

$$\text{CONTRIMERR I} \leqslant 2\left(2\alpha T + \frac{4\log(1/\zeta)}{3}\right)\cdot\left(u^{1+v}M^{-v} + 2M\right). \tag{96}$$

### D.2.2 Bounding CONTRIMERR II

In round $t \in N_P$, the losses are not contaminated. By Lemma 1, we have probability with at least $1 - \zeta$ that,

$$
\begin{aligned}
\text{CONTRIMERR II} &\leqslant 2\left|N_P\right| u^{1+v} M^{-v} + 8\sqrt{\left|N_P\right| u^{1+v} M^{1-v} \log(2\log(T)/\zeta)} \\
&\quad + 8M\log(2\log(T)/\zeta) \\
&\leqslant 6Tu^{1+v}M^{-v} + 12M\log(2\log(T)/\zeta).
\end{aligned}
\tag{97}
$$

### D.2.3 Bounding CONTRIMREG

We first rewrite this part as

$$
\begin{aligned}
\text{CONTRIMREG} &= \sum_{t=1}^{T}\left(\langle w_t, \widehat{\ell}_t\rangle - \langle y^{i^*}, \ell'_t\rangle\right) \\
&= \underbrace{\sum_{t=1}^{T}\langle w_t - y', \widehat{\ell}_t\rangle}_{\text{CONTRIMREGI}} + \underbrace{\sum_{t=1}^{T}\langle y' - y^{i^*}, \overline{\ell}'_t\rangle}_{\text{CONTRIMREGII}} + \underbrace{\sum_{t=1}^{T}\langle y', \widehat{\ell}_t - \overline{\ell}'_t\rangle}_{\text{CONTRIMREGIII}}.
\end{aligned}
\tag{98}
$$

### D.2.4 Bounding CONTRIMREGI

As long as it holds that $\eta \leqslant \frac{1}{10M}$, we have

$$
\sum_{t=1}^{T}\langle w_t - y', \widehat{\ell}_t\rangle \leqslant \frac{2K\log T}{\eta} - \frac{\langle y', \rho_T\rangle}{10\eta\log T} + 5\eta M^{1-v}\sum_{t=1}^{T}\left|\overline{\ell}'_{t,a_t}\right|^{1+v},
\tag{99}
$$

and it is left to bound $\sum_{t=1}^{T}\left|\overline{\ell}'_{t,a_t}\right|^{1+v}$.

We first rewrite it as

$$
\sum_{t=1}^{T}\left|\overline{\ell}'_{t,a_t}\right|^{1+v} = \sum_{t\in N_Q}\left|\widetilde{\ell}'_{t,a_t}\right|^{1+v} + \sum_{t\in N_P}\left|\ell'_{t,a_t}\right|^{1+v}.
\tag{100}
$$

Similarly, with probability at least $1 - \zeta$, it holds that

$$
\sum_{t\in N_Q}\left|\widetilde{\ell}'_{t,a_t}\right|^{1+v} \leqslant \left(2\alpha T + \frac{4\log(1/\zeta)}{3}\right)M^{1+v}.
\tag{101}
$$

As in the uncontaminated case, by Lemma 11, we have with probability at least $1 - \zeta$ that

$$
\begin{aligned}
\sum_{t\in N_P}\left|\ell'_{t,a_t}\right|^{1+v} &\leqslant \sum_{t\in N_P}\mathbb{E}\left[\left|\ell'_{t,a_t}\right|^{1+v}\right] + 4\sqrt{\left|N_P\right| u^{1+v} M^{1+v}\log(\log(T)/\zeta)} + 2M^{1+v}\log(\log(T)/\zeta) \\
&\leqslant Tu^{1+v} + 4\sqrt{Tu^{1+v}M^{1+v}\log(\log(T)/\zeta)} + 2M^{1+v}\log(\log(T)/\zeta).
\end{aligned}
\tag{102}
$$

Taking a union bound, we get with probability at least $1 - \zeta$ that

$$
\begin{aligned}
\text{CONTRIMREGI} &\leqslant \frac{2K\log T}{\eta} - \frac{\langle y', \rho_T\rangle}{10\eta\log T} + 5\eta\left(2\alpha T + \frac{4\log(2/\zeta)}{3}\right)M^2 + 5\eta M^{1-v}Tu^{1+v} \\
&\quad + 20\eta\sqrt{Tu^{1+v}M^{3-v}\log(2\log(T)/\zeta)} + 10\eta M^2\log(2\log(T)/\zeta) \\
&\leqslant \frac{2K\log T}{\eta} - \frac{\langle y', \rho_T\rangle}{10\eta\log T} + 5\eta\left(2\alpha T + \frac{4\log(2/\zeta)}{3}\right)M^2 + 15\eta M^{1-v}Tu^{1+v} \\
&\quad + 20\eta M^2\log(2\log(T)/\zeta).
\end{aligned}
\tag{103}
$$

### D.2.5 Bounding CONTRIMREGII

By Hölder's inequality, we have

$$
\begin{aligned}
\sum_{t=1}^{T} \langle y' - y^{i^*}, \bar{\ell}'_t \rangle &\leqslant \sum_{t=1}^{T} \left\| y' - y^{i^*} \right\|_{\infty} \left\| \bar{\ell}'_t \right\|_1 \\
&= \sum_{t=1}^{T} \max\{\lambda/K, \lambda(1 - 1/K)\} \cdot \left\| \bar{\ell}'_t \right\|_1 \\
&\leqslant \lambda \sum_{t=1}^{T} \sum_{i=1}^{K} \left| \bar{\ell}'_{t,i} \right| \\
&= \lambda \left( \sum_{t \in N_Q} \sum_{i=1}^{K} \left| \widetilde{\ell}'_{t,i} \right| + \sum_{t \in N_P} \sum_{i=1}^{K} |\ell'_{t,i}| \right).
\end{aligned}
\tag{104}
$$

We clearly have with probability at least $1 - \zeta$ that

$$
\sum_{t \in N_Q} \sum_{i=1}^{K} \left| \widetilde{\ell}'_{t,i} \right| \leqslant KM \left( 2\alpha T + \frac{4 \log(1/\zeta)}{3} \right),
\tag{105}
$$

and with probability with at least $1 - \zeta$ that

$$
\sum_{t \in N_Q} \sum_{i=1}^{K} |\ell'_{t,i}| \leqslant uKT + \sqrt{2KTu^{1+v}M^{1-v} \log(1/\zeta)} + \frac{4}{3} M \log(1/\zeta).
\tag{106}
$$

By taking a union bound, we get with probability at least $1 - \zeta$ that

$$
\text{CONTRIMREGII} \leqslant \lambda \left( (2\alpha M + u)KT + \sqrt{2KTu^{1+v}M^{1-v} \log(2/\zeta)} + 2KM \log(2/\zeta) \right).
\tag{107}
$$

### D.2.6 Bounding CONTRIMREGIII

We first rewrite CONTRIMREGIII as

$$
\text{CONTRIMREGIII} = \sum_{t \in N_Q} \langle y', \widehat{\ell}_t - \widetilde{\ell}'_t \rangle + \sum_{t \in N_P} \langle y', \widehat{\ell}_t - \ell'_t \rangle.
\tag{108}
$$

To bound the first part, recall that for any $t \in N_Q$, it holds that

$$
\left| \langle y', \widehat{\ell}_t - \widetilde{\ell}'_t \rangle \right| / M \leqslant 2 \langle y', \rho_T \rangle
\tag{109}
$$

and

$$
\begin{aligned}
\mathbb{E} \left[ (\langle y', \widehat{\ell}_t - \ell'_t \rangle / M)^2 \Big| \mathcal{F}_t \right] &\leqslant \frac{1}{M^2} \mathbb{E} \left[ (\langle y', \widehat{\ell}_t \rangle)^2 \Big| \mathcal{F}_t \right] \\
&= \frac{1}{M^2} \mathbb{E} \left[ \frac{(y'_{a_t})^2 (\widetilde{\ell}'_{t,a_t})^2}{(w_{t,a_t})^2} \Big| \mathcal{F}_t \right] \\
&= \frac{1}{M^2} \sum_{i=1}^{K} w_i \frac{(y'_i)^2}{(w_{t,i})^2} \mathbb{E} \left[ (\widetilde{\ell}'_{t,i})^2 \Big| \mathcal{F}_t, a_t = i \right] \\
&\leqslant \frac{1}{M^2} \sum_{i=1}^{K} \frac{(y'_i)^2}{w_{t,i}} M^2 \\
&\leqslant \langle y', \rho_T \rangle.
\end{aligned}
\tag{110}
$$

Therefore, with probability at least $1 - \zeta$, it holds that

$$
\sum_{t \in N_Q} \langle y', \widehat{\ell}_t - \widetilde{\ell}'_t \rangle \leqslant 3M \sqrt{\left( 2\alpha T + \frac{4 \log(2/\zeta)}{3} \right) \langle y', \rho_T \rangle \iota'} + 2M \max\{1, 2\langle y', \rho_T \rangle\} \iota'
$$

$$
\leqslant \frac{4.5M \left( 2\alpha T + \frac{4 \log(2/\zeta)}{3} \right)}{2} + \frac{2M \langle y', \rho_T \rangle \iota'}{2} + 2M \max\{1, 2\langle y', \rho_T \rangle\} \iota'
$$

$$
\leqslant 3M \left( 2\alpha T + \frac{4 \log(2/\zeta)}{3} \right) + 3M \max\{1, 2\langle y', \rho_T \rangle\} \iota', \tag{111}
$$

where $\iota' := \log(4 \max\{ \sqrt{\left( 2\alpha T + \frac{4 \log(2/\zeta)}{3} \right) \langle y', \rho_T \rangle}, 1, 2\langle y', \rho_T \rangle\}/\zeta)$.

For the uncontaminated part, with probability, by Lemma 12, we have with probability at least $1 - \zeta$ that,

$$
\sum_{t \in N_P} \langle y', \widehat{\ell}_t - \ell'_t \rangle \leqslant 3u \sqrt{T u^{-1+v} M^{1-v} \langle y', \rho_T \rangle \iota''} + 2u \max\{1, 2M \langle y', \rho_T \rangle / u\} \iota''
$$

$$
\leqslant \frac{4.5 u T u^v M^{-v}}{2} + \frac{2u u^{-1} M \langle y', \rho_T \rangle \iota'}{2} + 2u \max\{1, 2M \langle y', \rho_T \rangle / u\} \iota''
$$

$$
\leqslant 3T u^{1+v} M^{-v} + 5 \max\{u, M \langle y', \rho_T \rangle\} \iota'', \tag{112}
$$

where $\iota'' := \log(2 \max\{ \sqrt{T u^{-1+v} M^{1-v} \langle y', \rho_T \rangle}, 1, 2M \langle y', \rho_T \rangle / u\}/\zeta)$. Taking a union bound, we get with probability at least $1 - \zeta$ that

$$
\textsc{ConTrimRegiii} \leqslant 3M \left( 2\alpha T + \frac{4 \log(3/\zeta)}{3} \right) + 3T u^{1+v} M^{-v}
$$

$$
+ 3M \max\{1, 2\langle y', \rho_T \rangle\} \log(6 \max\{ \sqrt{\left( 2\alpha T + \frac{4 \log(3/\zeta)}{3} \right) K/\lambda}, 2K/\lambda\}/\zeta)
$$

$$
+ 3 \max\{1, 2M \langle y', \rho_T \rangle\} \log(6 \max\{ \sqrt{2T u^{1+v} M^{1-v} K/\lambda}, 4MK/\lambda\}/\zeta). \tag{113}
$$

### D.2.7 Putting All Pieces Together

Combining all the parts, we have with probability at least $1 - \zeta$ that

$$
R_T \leqslant 4 \left( \alpha T + \frac{2 \log(5/\zeta)}{3} \right) \cdot \left( u^{1+v} M^{-v} + 2M \right) + 6T u^{1+v} M^{-v} + 12M \log(10 \log(T)/\zeta)
$$

$$
+ \frac{2K \log T}{\eta} - \frac{\langle y', \rho_T \rangle}{10\eta \log T} + 10\eta \left( \alpha T + \frac{2 \log(10/\zeta)}{3} \right) M^2 + 15\eta M^{1-v} T u^{1+v}
$$

$$
+ 20\eta M^2 \log(10 \log(T)/\zeta) + (2\alpha M + u) \lambda K T + \lambda \sqrt{2KT u^{1+v} M^{1-v} \log(10/\zeta)}
$$

$$
+ 2\lambda K M \log(10/\zeta) + 3M \left( 2\alpha T + \frac{4 \log(15/\zeta)}{3} \right)
$$

$$
+ 3M \max\{1, 2\langle y', \rho_T \rangle\} \log(15 \max\{ \sqrt{\left( 2\alpha T + \frac{4 \log(15/\zeta)}{3} \right) K/\lambda}, 2K/\lambda\}/\zeta)
$$

$$
+ 3T u^{1+v} M^{-v}
$$

$$
+ 5 \max\{u, M \langle y', \rho_T \rangle\} \log(10 \max\{ \sqrt{2T u^{-1+v} M^{1-v} K/\lambda}, 4MK/(u\lambda)\}/\zeta). \tag{114}
$$

We first plug in $\lambda$ and simplify the upper bound as

$$R_T \leqslant 4\left(\alpha T + \frac{2\log(5/\zeta)}{3}\right) \cdot \left(u^{1+v}M^{-v} + 2M\right) + 9Tu^{1+v}M^{-v} + 12M\log(10\log(T)/\zeta)$$

$$+ \frac{2K\log T}{\eta} - \frac{\langle y', \rho_T\rangle}{10\eta\log T} + 10\eta\left(\alpha T + \frac{2\log(10/\zeta)}{3}\right)M^2 + 15\eta M^{1-v}Tu^{1+v}$$

$$+ 20\eta M^2\log(10\log(T)/\zeta) + 2\alpha MT^{\frac{1}{1+v}}K^{\frac{v}{1+v}} + uT^{\frac{1}{1+v}}K^{\frac{v}{1+v}} + uT^{\frac{1}{1+v}}K^{\frac{v}{1+v}}\sqrt{2\log(10/\zeta)}$$

$$+ 2T^{\frac{-v}{1+v}}K^{\frac{v}{1+v}}M\log(10/\zeta) + 3M\left(2\alpha T + \frac{4\log(15/\zeta)}{3}\right)$$

$$+ 6M\langle y', \rho_T\rangle\log(15\max\{\sqrt{\left(2\alpha T + \frac{4\log(15/\zeta)}{3}\right)KT}, 2KT\}/\zeta)$$

$$+ 5\max\{u, M\langle y', \rho_T\rangle\}\log(10T\max\{\sqrt{2u^{-1+v}M^{1-v}K}, 4MK/u\}/\zeta). \tag{115}$$

Assuming $\alpha T \geqslant \frac{4}{3}\log(15\log(T)/\zeta)$ and $T \geqslant \sqrt{T\log(15\log(T)/\zeta)}$, we further simplify the upper bound as

$$R_T \leqslant 8\alpha T \cdot \left(u^{1+v}M^{-v} + 2M\right) + 9Tu^{1+v}M^{-v} + 21M\alpha T$$

$$+ \frac{2K\log T}{\eta} - \frac{\langle y', \rho_T\rangle}{10\eta\log T} + 20\eta\alpha TM^2 + 35\eta M^{1-v}Tu^{1+v}$$

$$+ 20\eta M^2\alpha T + 2\alpha MT^{\frac{1}{1+v}}K^{\frac{v}{1+v}} + uT^{\frac{1}{1+v}}K^{\frac{v}{1+v}} + uT^{\frac{1}{1+v}}K^{\frac{v}{1+v}}\sqrt{2\log(10/\zeta)}$$

$$+ 2\alpha MT^{\frac{1}{1+v}}K^{\frac{v}{1+v}} + 6M\langle y', \rho_T\rangle\log(15\max\{\sqrt{3\alpha KT^2}, 2KT\}/\zeta)$$

$$+ 5\max\{u, M\langle y', \rho_T\rangle\}\log(10T\max\{\sqrt{2u^{-1+v}M^{1-v}K}, 4MK/u\}/\zeta)$$

$$\leqslant 32\alpha MT + 17Tu^{1+v}M^{-v} + 40\eta\alpha TM^2 + 35\eta M^{1-v}Tu^{1+v} + \frac{2K\log T}{\eta}$$

$$- \frac{\langle y', \rho_T\rangle}{10\eta\log T} + uT^{\frac{1}{1+v}}K^{\frac{v}{1+v}} + uT^{\frac{1}{1+v}}K^{\frac{v}{1+v}}\sqrt{2\log(10/\zeta)}$$

$$+ 6M\langle y', \rho_T\rangle\log(30KT/\zeta)$$

$$+ 5\max\{u, M\langle y', \rho_T\rangle\}\log(10T\max\{\sqrt{2u^{-1+v}M^{1-v}K}, 4MK/u\}/\zeta), \tag{116}$$

where in the last inequality we apply $K \leqslant T$ to upper bound terms of form $\Theta(\alpha MT^{\frac{1}{1+v}}K^{\frac{v}{1+v}})$.

For any $\eta > 0$, choosing $M = \min\{\frac{u}{\alpha^{\frac{1}{1+v}}}, \left(\frac{u^{1+v}}{\eta\alpha}\right)^{\frac{1}{2+v}}, \frac{1}{10\eta}\}$, we have

$$R_T = O\left((\eta\alpha)^{\frac{v}{2+v}}u^{1+\frac{v}{2+v}}T + (\eta)^v u^{1+v}T + \frac{K\log T}{\eta}\right)$$

$$- \frac{\langle y', \rho_T\rangle}{10\eta\log T} + 6M\langle y', \rho_T\rangle\log(30KT/\zeta)$$

$$+ 5\max\{u, M\langle y', \rho_T\rangle\}\log(10T\max\{\sqrt{2u^{-1+v}M^{1-v}K}, 4MK/u\}/\zeta)$$

$$+ O\left(u\alpha^{\frac{v}{1+v}}T + uT^{\frac{1}{1+v}}K^{\frac{v}{1+v}}\sqrt{\log(1/\zeta)}\right). \tag{117}$$

Now we are going to choose $\eta = \frac{1}{u} \cdot \min\{\left(\frac{K\log(T)}{T}\right)^{\frac{1}{1+v}}, \left(\frac{K\log(T)}{T}\right)^{\frac{2+v}{2+2v}}\left(\frac{1}{\alpha}\right)^{\frac{v}{2+2v}}\}$. We further assume that $\alpha^{\frac{v}{(1+v)(2+v)}}T \geqslant K\log(T)$, which implies that

$$\left(\frac{K\log(T)}{T}\right)^{\frac{1}{1+v}} \geqslant \left(\frac{K\log(T)}{T}\right)^{\frac{2+v}{2+2v}}\left(\frac{1}{\alpha}\right)^{\frac{v}{2+2v}} \tag{118}$$

and thus $\eta = \frac{1}{u}\left(\frac{K\log(T)}{T}\right)^{\frac{2+v}{2+2v}}\left(\frac{1}{\alpha}\right)^{\frac{v}{2+2v}}$.

Finally, we need to cancel the terms containing $\langle y', \rho_T \rangle$ to obtain the high-probability guarantee.

**Case 1:** $u \geqslant M\langle y', \rho_T \rangle$. If $\max\{u, M\langle y', \rho_T \rangle\} = u$, it would just introduce a $\widetilde{O}(u)$ term which can be ignored. In this case, $\eta$ must satisfy that

$$-\frac{1}{10\eta \log(T)} + 6M \log(30KT/\zeta) \leqslant 0, \tag{119}$$

which implies that

$$\eta \leqslant \frac{1}{60M \log(T) \log(30KT/\zeta)}. \tag{120}$$

**Case 2:** $u < M\langle y', \rho_T \rangle$. With the current choice of $M$ and $\eta$, we have

$$M \leqslant \frac{1}{10\eta} \leqslant 0.1 u\alpha^{\frac{v}{2+2v}} \left( \frac{T}{K \log(T)} \right)^{\frac{2+v}{2+2v}} \leqslant uT/K, \tag{121}$$

which implies that

$$\log(10T \max\{\sqrt{2u^{-1+v}M^{1-v}K}, 4MK/u\}/\zeta) \leqslant \log(10T \max\{\sqrt{2T}, 4T\}/\zeta)$$
$$\leqslant \log(40T^2/\zeta). \tag{122}$$

Now it is sufficient to have $\eta$ satisfying that

$$-\frac{1}{10\eta \log(T)} + 11M \log(40T^2/\zeta) \leqslant 0, \tag{123}$$

meaning that

$$\eta \leqslant \frac{1}{110M \log(T) \log(40T^2/\zeta)}. \tag{124}$$

In summary, in either case, it is sufficient to have $\eta M \leqslant \frac{1}{110 \log(T) \log(40T^2/\zeta)}$. Note that this is stronger than (and implies) the stability condition $\eta M \leqslant 0.1$.

Starting again from Eq. (117), with any $\eta > 0$, choosing

$$M = \min\{ \frac{u}{\alpha^{\frac{1}{1+v}}}, \left( \frac{u^{1+v}}{\eta\alpha} \right)^{\frac{1}{2+v}}, \frac{1}{110\eta \log(T) \log(40T^2/\zeta)} \},$$

we have

$$R_T = O\left( (\eta\alpha)^{\frac{v}{2+v}} u^{1+\frac{v}{2+v}} T + (\eta)^v u^{1+v} T \left( \log(T) \log(40T^2/\zeta) \right)^v + \frac{K \log T}{\eta} \right)$$
$$- \frac{\langle y', \rho_T \rangle}{10\eta \log T} + 6M\langle y', \rho_T \rangle \log(30KT/\zeta)$$
$$+ 5 \max\{u, M\langle y', \rho_T \rangle\} \log(10T \max\{\sqrt{2u^{-1+v}M^{1-v}K}, 4MK/u\}/\zeta)$$
$$+ O\left( u\alpha^{\frac{v}{1+v}} T + uT^{\frac{1}{1+v}} K^{\frac{v}{1+v}} \sqrt{\log(1/\zeta)} \right). \tag{125}$$

Lastly, still choosing

$$\eta = \frac{1}{u} \cdot \min\{ \left( \frac{K \log(T)}{T} \right)^{\frac{1}{1+v}}, \left( \frac{K \log(T)}{T} \right)^{\frac{2+v}{2+2v}} \left( \frac{1}{\alpha} \right)^{\frac{v}{2+2v}} \} = \frac{1}{u} \left( \frac{K \log(T)}{T} \right)^{\frac{2+v}{2+2v}} \left( \frac{1}{\alpha} \right)^{\frac{v}{2+2v}}$$

yields that

$$R_T = O\left( uK^{\frac{v}{2+2v}} T^{\frac{2+v}{2+2v}} (\log T)^{\frac{v}{2+2v}} \alpha^{\frac{v}{2+2v}} + uK^{\frac{v}{1+v}} T^{\frac{1}{1+v}} (\log(T))^{1+\frac{v}{1+v}} \left( \log(40T^2/\zeta) \right)^v \right)$$
$$+ O\left( u\alpha^{\frac{v}{1+v}} T + uT^{\frac{1}{1+v}} K^{\frac{v}{1+v}} \sqrt{\log(1/\zeta)} \right)$$
$$\leqslant O\left( u\alpha^{\frac{v}{1+v}} T + uK^{\frac{v}{1+v}} T^{\frac{1}{1+v}} \left( (\log(T))^{1+\frac{v}{1+v}} \left( \log(40T^2/\zeta) \right)^v + \sqrt{\log(1/\zeta)} \right) \right)$$
$$\leqslant O\left( u\alpha^{\frac{v}{1+v}} T + uK^{\frac{v}{1+v}} T^{\frac{1}{1+v}} (\log T)^{1.5} \log(40T^2/\zeta) \right). \tag{126}$$

### D.3 Regret Lower Bound with Contamination

**Theorem 4** (Lower Bound for Heavy-tailed Adversarial Bandits with Huber Contamination). *For any bandit algorithm, there must exist one problem instance in which the algorithm suffers regret*

$$\mathbb{E}[R_T] = \Omega\left(uT^{\frac{1}{1+v}}K^{\frac{v}{1+v}} + uT\alpha^{\frac{v}{1+v}}\right).$$

*Proof of Theorem 4.* In the presence of heavy tails, every algorithm suffers $\Omega\left(uT^{\frac{1}{1+v}}K^{\frac{v}{1+v}}\right)$ regret in the worst case, *regardless of contamination level* $\alpha$ (Bubeck et al., 2013).

Therefore, to show the lower bound with contamination, it suffices to show a lower bound of

$$\Omega\left(uT\alpha^{\frac{v}{1+v}}\right)$$

given any $\alpha \in (0, 1]$.

This proof is a direct modification based on Wu et al. (2024, Appendix B). We use $\pi$ to denote a bandit algorithm. We construct two environments, denoted by $\nu_1$ and $\nu_2$, respectively. And then we show that any algorithm suffers the claimed regret in one of these two environments.

**Environment $\nu_1$.** In $\nu_1$, the loss of action 1 in every round $t \in [T]$ is given by

$$\ell_{t,1} = \begin{cases} u/\gamma, & \text{with probability } \frac{1}{2}\gamma^{1+v}, \\ 0, & \text{with probability } 1 - \frac{1}{2}\gamma^{1+v}, \end{cases}$$

where $\gamma \leqslant 1$ is some free parameter to choose at the last step of the proof. One can verify that $\mathbb{E}[|\ell_1|^{1+v}] \leqslant u^{1+v}$ via direct calculations.

For any suboptimal action $i \neq 1$, the loss in every round $t \in [T]$ is given by

$$\ell_{t,i} = \begin{cases} u/\gamma, & \text{with probability } \frac{3}{10}\gamma^{1+v}, \\ 0, & \text{with probability } 1 - \frac{3}{10}\gamma^{1+v}. \end{cases}$$

One can verify that $\mathbb{E}[|\ell_{t,i}|^{1+v}] \leqslant u^{1+v}, \forall i \neq 1$. Moreover, action 1 is the optimal one and we have the "sub-optimality gap" $\Delta := \mathbb{E}[\ell_{t,1} - \ell_{t,i}] = \frac{u}{5}\gamma^v, \forall i \neq 1$.

Given algorithm $\pi$ and environment $\nu_1$, we define $i' = \underset{i \in \{2,...,K\}}{\operatorname{argmin}} \underset{\pi,\nu_1}{\mathbb{E}}[N_{T,i}]$, and hence we have $\underset{\pi,\nu_1}{\mathbb{E}}[N_{T,i'}] \leqslant \frac{T}{K-1}$. Now we are able to construct the second environment.

**Environment $\nu_2$.** In this environment, everything is the same as in $\nu_1$, except that for action $i'$, now the loss follows

$$\ell_{t,i'} = \begin{cases} u/\gamma, & \text{with probability } \frac{7}{10}\gamma^{1+v}, \\ 0, & \text{with probability } 1 - \frac{7}{10}u^{1+v}. \end{cases}$$

One can verify that $\mathbb{E}[\ell_{i'}] = \frac{7}{10}\gamma^{1+v}, \mathbb{E}[|\ell_{i'}|^{1+v}] \leqslant u^{1+v}$ and now the optimal action is $i'$. We use $\widetilde{\nu}_2$ to denote the contaminated version of $\nu_2'$ (where the bad distributions are determined later).

And then, we use $\widetilde{\nu}_1$ ($\widetilde{\nu}_2$) to denote the contaminated version of $\nu_1$ ($\nu_2$). The bad distributions will be determined later.

**Environments $\widetilde{\nu}_1$ and $\widetilde{\nu}_2$.** We choose $\gamma = \alpha^{\frac{1}{1+v}} \in (0, 1]$. Then for any $i \in [K]$, we have $\mathrm{TV}(P_{t,i}\|P'_{t,i}) \leqslant \frac{2}{5}\gamma^{1+v} = \frac{2}{5}\alpha \leqslant \frac{\alpha}{1-\alpha}$. According to Lemma 13, for any action $i \in [K]$, there exist bad distributions $G_i$ and $G'_i$ such that

$$(1-\alpha)P_{t,i} + \alpha G_i = (1-\alpha)P'_{t,i} + \alpha G'_i,$$

where $P_{t,i}$ ($P'_{t,i}$) denotes the loss distribution of action $i$ in round $t$ under environment 1 (2). We construct $\widetilde{\nu}_1$ and $\widetilde{\nu}_2$ by

$$\widetilde{\nu}_1 = \{x_i = (1-\alpha)P_{t,i} + \alpha G_i : i \in [K]\},$$
$$\widetilde{\nu}_2 = \{x'_i = (1-\beta)P'_{t,i} + \beta G'_i : i \in [K]\},$$

where $x_i$ and $x'_i$ denote the loss distributions for action $i$ in these two environments, respectively.

Following from the regret definition, we first have

$$R_T(\pi, \widetilde{\nu}_1) = \Delta \left( T - \underset{\pi, \widetilde{\nu}_1}{\mathbb{E}} [N_{T,1}] \right) \geqslant \frac{\Delta T}{2} \mathbb{P}_{\pi, \widetilde{\nu}_1} \left( N_{T,1} \leqslant \frac{T}{2} \right),$$

$$R_T(\pi, \widetilde{\nu}_2) = \Delta \underset{\pi, \widetilde{\nu}_2}{\mathbb{E}} [N_{T,1}] + \sum_{i \notin \{1, i'\}} 2\Delta \underset{\pi, \widetilde{\nu}_2}{\mathbb{E}} [N_{T,i}] \geqslant \frac{\Delta T}{2} \mathbb{P}_{\pi, \widetilde{\nu}_2} \left( N_{T,1} \geqslant \frac{T}{2} \right).$$

By adding them together, we have

$$R_T(\pi, \widetilde{\nu}_1) + R_T(\pi, \widetilde{\nu}_2) \geqslant \frac{\Delta T}{2} \left( \mathbb{P}_{\pi, \widetilde{\nu}_1} \left( N_{T,1} \leqslant \frac{T}{2} \right) + \mathbb{P}_{\pi, \widetilde{\nu}_2} \left( N_{T,1} \geqslant \frac{T}{2} \right) \right)$$

$$\overset{(a)}{\geqslant} \frac{\Delta T}{4} \exp \left( -\text{KL}(\mathbb{P}_{\pi, \widetilde{\nu}_1} \| \mathbb{P}_{\pi, \widetilde{\nu}_2}) \right)$$

$$\overset{(b)}{=} \frac{\Delta T}{4} \exp(0),$$

where step (a) follows from the Bretagnolle–Huber inequality (Lattimore & Szepesvári, 2020, Theorem 14.2), and step (b) is due to the fact that $\widetilde{\nu}_1$ and $\widetilde{\nu}_2$ are identical under our construction.

Recall that $\Delta = \frac{u}{5}\gamma^v$ and $\gamma = \alpha^{\frac{1}{1+v}}$, we arrive at

$$\max\{R_T(\pi, \nu_1), R_T(\pi, \nu_2)\} \geqslant \Omega(uT\alpha^{\frac{v}{1+v}}),$$

which completes the proof. $\qquad\square$

## E  BOBW Guarantees with Bounded Losses and Local Differential Privacy

We first give the definition of Differential Privacy followed by the learning setup.

**Definition 1** (Differential Privacy (DP)). For any given privacy budget $\varepsilon > 0, \delta \geqslant 0$, a mechanism $\mathcal{M} : \mathcal{D} \to \mathbb{R}^m$ is said to be $(\varepsilon, \delta)$-differentially private (DP) if for all datasets $X, X'$ in $\mathcal{D}$ that differ on only one element and measurable subset $\mathcal{E} \subset \mathbb{R}^m$, it holds that $\mathbb{P}(\mathcal{M}(X) \in \mathcal{E}) \leqslant \exp(\varepsilon) \cdot \mathbb{P}(\mathcal{M}(X') \in \mathcal{E}) + \delta$. When $\delta = 0$, we refer to $(\varepsilon, \delta)$-DP as $\varepsilon$-DP (pure DP), which is stronger than $(\varepsilon, \delta)$-DP for some $\delta > 0$ (approximate DP).

**Definition 2** (Bandits with Bounded Losses and Local DP (LDP)). All loss distributions $(P_{t,i})_{t \in [T], i \in [K]}$ have support bounded in $[0, 1]$. Given any privacy budget $\varepsilon \in (0, 1], \delta > 0$, the bandit model is said to be $(\varepsilon, \delta)$-LDP if $a_{t+1}$ lies in the sigma-algebra generated by $\{a_{t'}, \mathcal{M}(\ell_{t, a_{t'}})\}_{t' \in [t]}$ in any round $t \in [T]$ where $\mathcal{M}$ is an $(\varepsilon, \delta)$-DP mechanism.

Roughly speaking, the algorithm should not touch true losses, and it observes privatized losses only. Here, we adopt the widely-used Laplace mechanism (Dwork et al., 2014). Specifically, when data are bounded in $[0, 1]$, adding noise drawn from $\text{Lap}(\varepsilon)$[7] to them ensures $\varepsilon$-DP. By adopting it, the observed loss is the true loss plus an i.i.d. sample from $\text{Lap}(\varepsilon)$. That is, this setup could be viewed as a specific way of generating heavy-tailed losses (i.e., bounded true loss + Laplace noise for privacy).

In the literature, Agarwal & Singh (2017) and Tossou & Dimitrakakis (2017) investigated the adversarial regime and proposed algorithms that achieve $\widetilde{O}(\frac{\sqrt{KT}}{\varepsilon})$ worst-case regret (in expectation) with $\varepsilon$-LDP protection, which matches the lower bound (Garcelon et al., 2021) up to some $\log T$ factor and implies that their algorithms are (nearly) minimax-optimal. In the stochastic regime, Ren et al.

---

[7]Laplace distribution with parameter $\eta$ is denoted by $\text{Lap}(\eta)$ and has Probability Density Function $f_\eta(z) = \eta \cdot \exp(-\eta|z|)/2, \forall z \in \mathbb{R}$.

(2020) proposed privatized Upper Confidence Bound-based algorithms with $O(\sum_{i:\Delta_i>0} \frac{\log T}{\varepsilon^2 \Delta_i})$ gap-dependent regret and $O(\frac{\sqrt{KT\log T}}{\varepsilon})$ worst-case regret. They also provided a matching gap-dependent lower bound, which scales as $\Omega(\sum_{i:\Delta_i>0} \frac{\log T}{\varepsilon^2 \Delta_i})$ when $\varepsilon \leqslant 1$.

In the BOBW setting, Zheng et al. (2020) showed that by privatizing the Tsallis-INF (an optimal algorithm in the (non-private) BOBW setting when losses are bounded) with Gaussian noise, they achieved $O(\frac{\sqrt{KT}}{\varepsilon})$ regret in the adversarial regime and $O\left(\sum_{i:\Delta_i>0} \frac{\log(T)\log(1/\delta)}{\varepsilon^2 \Delta_i}\right)$ regret in the stochastic regime with approximate $(\varepsilon, \delta)$-LDP protection, which is weaker than pure $\varepsilon$-LDP. And these regret bounds hold in expectation only.

In what follows, we present the BOBW guarantee provided by our Algorithm 2 when the losses are privatized by noise from $\mathrm{Lap}(\varepsilon)$.

Recall that the privatized loss $\overline{\ell}_{t,i} = \ell_{t,i} = \ell_{t,i} + z$, where $\ell_{t,i}$ is bounded in $[0,1]$ and $z \sim \mathrm{Lap}(\varepsilon)$, we have

$$\mathbb{E}[|\overline{\ell}_{t,i}|^2] = \mathbb{E}[|\ell_{t,i} + z|^2] \leqslant 2\mathbb{E}[(\ell_{t,i})^2] + 2\mathbb{E}[z^2] \leqslant 2 + 2(1/\varepsilon)^2 = O(1/\varepsilon^2), \qquad (127)$$

where in the last inequality we utilize the fact that distribution $\mathrm{Lap}(\varepsilon)$ has second moment bounded by $1/\varepsilon^2$.

Therefore, plugging $u = O(1/\varepsilon)$ and $v = 2$ into Theorem 2 yields the following high-probability BOBW regret guarantee with (pure) $\varepsilon$-LDP protection: $O(\frac{\sqrt{KT}\log(K)(\log T)^3}{\varepsilon})$ worst-case regret in the adversarial regime and $O(\frac{K\log(K)(\log T)^4}{\varepsilon^2 \Delta})$ regret in the stochastic regime with high probability. To the best of our knowledge, this is the first BOBW regret guarantee in MAB with pure LDP protection. We believe that our BOBW results can also be generalized to the case when true losses (to be protected) are heavy-tailed rather than bounded (by properly trimming the true losses before privatization). Related work by Tao et al. (2022) investigated solely the stochastic regime.

# F   Auxiliary Lemmas

**Lemma 8.** *For any fixed $M_{t,i} > 0$, it holds almost surely that $\left|\mu_{t,i} - \mu'_{t,i}\right| \leqslant u^{1+v}(M_{t,i})^{-v}$.*

*Proof of Lemma 8.* Expanding the definitions of $\mu_{t,i}$ and $\mu'_{t,i}$, we have

$$\begin{aligned}
\left|\mu_{t,i} - \mu'_{t,i}\right| &= \left| \mathop{\mathbb{E}}_{\ell_{t,i} \sim P_{t,i}}[\ell_{t,i}] - \mathop{\mathbb{E}}_{\ell_{t,i} \sim P_{t,i}}[\ell_{t,i} \cdot \mathbb{I}\{|\ell_{t,i}| \leqslant M_{t,i}\}] \right| \\
&= \left| \mathop{\mathbb{E}}_{\ell_{t,i} \sim P_{t,i}}[\ell_{t,i} \cdot \mathbb{I}\{|\ell_{t,i}| > M_{t,i}\}] \right| \\
&\leqslant \mathop{\mathbb{E}}_{\ell_{t,i} \sim P_{t,i}}[|\ell_{t,i}| \cdot \mathbb{I}\{|\ell_{t,i}| > M_{t,i}\}] \\
&\leqslant \mathop{\mathbb{E}}_{\ell_{t,i} \sim P_{t,i}}[|\ell_{t,i}|^{1+v}(M_{t,i})^{-v} \cdot \mathbb{I}\{|\ell_{t,i}| > M_{t,i}\}] \\
&\leqslant \mathop{\mathbb{E}}_{\ell_{t,i} \sim P_{t,i}}[|\ell_{t,i}|^{1+v}(M_{t,i})^{-v}] \\
&\leqslant u^{1+v}(M_{t,i})^{-v}.
\end{aligned}$$

$\square$

**Lemma 9** (Adapted from Lemma 19 in Wei & Luo (2018)). *In Algorithm 1, for any fixed $\lambda \in (0,1)$, let $n_i$ be the number of times the learning rate of arm $i$ changes (and in fact increases) in total, i.e., $\eta_{T+1,i} = \kappa^{n_i}\eta_{1,i} = \kappa^{n_i}\eta$. We have $n_i \leqslant \log_2(1/\lambda)$ and $\eta_{t,i} \leqslant e^{\frac{\log_2(1/\lambda)}{\log T}}\eta, \forall t \in [T], i \in [K]$.*

*Proof of Lemma 9.* Let $t_1, \ldots, t_{n_i}$ be the (ordered) rounds when the learning rate of action $i$ changes, we have

$$\frac{K}{\lambda} \geqslant \frac{1}{w_{t_{n_i},i}} > \rho_{t_{n_i},i} > 2\rho_{t_{n_i}-1,i} > \cdots > 2^{n_i-1}\rho_{t_1,i} = 2^{n_i}K.$$

Clearly, we have $n_i \leqslant \log_2(1/\lambda)$ and therefore $\eta_{t,i} \leqslant \kappa^{n_i}\eta_{1,i} \leqslant e^{\frac{\log_2(1/\lambda)}{\log T}}\eta$. $\square$

**Lemma 10** (Bernstein's inequality for independent random variables (Vershynin, 2018)). *Let $X_1, \ldots, X_T$ be zero-mean independent random variables such that $|X_t| \leqslant b, \forall t \in [T]$ for some fixed constant $b$ and $V_T = \sum_{i=1}^{T} \mathbb{E}\left[(X_t)^2\right]$. Then, for any $\zeta > 0$, we have with probability at least $1 - \zeta$ that,*

$$\sum_{i=1}^{T} X_i \leqslant \sqrt{2V_T \log(1/\zeta)} + \frac{2b \log(1/\zeta)}{3}.$$

**Lemma 11** (Freedman's inequality for martingales, Lemma 3 in Rakhlin et al. (2011)). *Let $X_1, \ldots, X_T$ be a martingale difference sequence adapted to filtration $\mathcal{F}_1 \subset \cdots \subset \mathcal{F}_T$ such that $|X_t| \leqslant b$ almost surely for some fixed constant $b$ and $V_t = \sum_{s=1}^{t} \mathbb{E}\left[(X_s)^2 | \mathcal{F}_s\right]$. Then, for any $\zeta < 1/(e \log T)$ and $T \geqslant 4$, we have with probability at least $1 - \zeta$ that for any $t \in [T]$,*

$$\sum_{i=1}^{t} X_i \leqslant 2 \max\{2\sqrt{V_t}, b\sqrt{\log(\log(T)/\zeta)}\} \sqrt{\log(\log(T)/\zeta)} \leqslant 4\sqrt{V_t \log(\log(T)/\zeta)} + 2b \log(\log(T)/\zeta).$$

**Lemma 12** (Adaptive Freedman's inequality, Theorem 9 in Zimmert & Lattimore (2022)). *Let $X_1, \ldots, X_T$ be a martingale difference sequence adapted to filtration $\mathcal{F}_1 \subset \cdots \subset \mathcal{F}_T$ such that $\mathbb{E}\left[X_t | \mathcal{F}_t\right]$ is finite almost surely. Then, for any $\zeta > 0$, it holds with probability at least $1 - \zeta$ that*

$$\sum_{i=1}^{T} X_i \leqslant 3\sqrt{V_T \log\left(\frac{2 \max\{U_T, \sqrt{V_T}\}}{\zeta}\right)} + 2U_T \log\left(\frac{2 \max\{U_T, \sqrt{V_T}\}}{\zeta}\right),$$

*where $V_T = \sum_{i=1}^{T} \mathbb{E}\left[(X_t)^2 | \mathcal{F}_t\right]$ and $U_T = \max\{1, \max_{t \in [T]} X_t\}$.*

**Lemma 13** (Theroem 5.1 of Chen et al. (2018)). *Let $R_1$ and $R_2$ be two distributions on $\mathcal{X}$. If for some $\alpha \in [0, 1]$ it holds that $\mathrm{TV}(R_1 \| R_2) \leqslant \frac{\alpha}{1 - \alpha}$, then there exist two distributions on the same probability space $G_1$ and $G_2$ such that*

$$(1 - \alpha)R_1 + G_1 = (1 - \alpha)R_2 + G_2.$$

