# OpenReview forum: "Taming Heavy-Tailed Losses in Adversarial Bandits and the Best-of-Both-Worlds Setting"
_NeurIPS.cc/2024/Conference — NeurIPS 2024 poster_

### Official Review · Reviewer_q85G · 2024-07-06

**Soundness:** 3
**Presentation:** 3
**Contribution:** 2
**Rating:** 6
**Confidence:** 4

**Summary:**

This paper studies multi-armed bandits (MAB) with heavy-tailed losses.
In the heavy-tailed bandits literature, two common assumptions are made about the losses: either a known upper bound $u$ on (1+$v$)-th raw moments (where both/either $v$ and/or $u$ are known) or truncated non-negativity (or non-positivity) assumption.
This paper considers the former and proposes a Best-of-Both-Worlds (BOBW) policy based on the Online Mirror Descent (OMD) and detect-switch framework.
The authors present high-probability near-optimal regret bounds that can provide meaningful expected regret analysis.
The main techniques used in their proofs rely on variants of Freedman's inequality (Lemma 11 and 12).

- - -
After rebuttal:

The proof for the modified algorithm seems correct, though it introduces an additional logarithmic factor.
While I believe a better result could potentially be achieved using the general FTRL framework in BOBW research instead of the switch-detect framework, further improving the current analysis could be explored as a separate research direction.

As a result, I am changing my score from 3 to 6 (and the soundness rating from 1 to 3).

**Strengths:**

### Quality and Clarity
The main manuscript is clearly written and easy to follow.

### Originality and Significance
This paper provides high-probability regret bounds for both stochastic and adversarial settings.
The proposed policy is novel as it utilizes both OMD and a switch-detect policy specifically for (stochastic) heavy-tailed bandits.
Additionally, the results imply the achievability of BOBW performance in a pure local differential privacy setting.

**Weaknesses:**

### Assumption on losses
Although the authors claim that their assumption, which requires knowledge of both $v$ and $u$, relaxes the undesirable truncated non-negative losses considered in Huang et al. (2022) or non-positivity assumption in Genalti et al. (2024), I disagree with this claim.
Genalti et al. (2024) showed that there is **no** $u,v$-adaptive algorithm that can achieve the lower bound for known $u,v$ setting.
I believe that these assumptions cannot be directly compared to determine which is generally weaker **in practice**.
However, the truncated assumption appears to be more challenging problem in this context.

### Proof of Lemma 2 - step 2 (p. 17)
In the equation between (25) and (26), the authors use $\bar{w} = \rho /2$, which is **incorrect** since $\rho = 2/w$ not $2/\bar{w}$ as shown in Algorithm 1.
Therefore, the results in (26) and (27) are not necessarily true.
This implies that the authors should modify the choice of $\eta$, which was designed to cancel terms related to the results in (27).
This issue seems very critical and needs to be addressed, even if it might not change the order of regret.

### Proof for adversarial setting (Alg. 2)
By definition, for $s \leq t\_{sw}-1$, none of the tests (6), (7), and (8) should be satisfied.
This implies that for all $i\in [K]$, (6) should not be satisfied, i.e., for all $i \in [K]$,
$$
\begin{equation*}
       | {\frac{\hat{L}\_{s,i}}{s} - \hat{\mu}\_{s,i}} | > 9u(\cdot)^{v/(1+v)} + 1[i \in A\_s] \text{Width}(s)/s + 1[i \not\in A\_s] \text{Width}(\tau\_i)/\tau\_i.
\end{equation*}
$$
However, equation (69) used exactly the opposite condition.
Therefore, the arguments related to bounding Part A seem incorrect.

Also, what is $\hat{L}\_{t\_{sw}-1}'$?
Is it different from the cumulative importance-weighted estimate?

### Stochastic results
From (64), the result is of order $K\log K \log T (\log(T/\xi))^3$, not $K\log K \log T (\log(T/\xi))^2$ stated in Theorem 2.

**Questions:**

1. What is $\hat{y}\_t$ in (2)?

1. What is $t\_{quit}$ in (13)? (just $t\_{sw}$?)

2. I cannot understand why (55) holds with $\tau\$ terms inside when $t > \tau$. To be specific $(\tau\_i / K)^{(1-v)/(1+v)}$ term.

3. Why (59) holds for all $i \in [K]$ instead of $i \in A\_t$? The last step used the test (5), where only $i\in A\_t$ satisfies $\leq$.

4. Can you explain more on (63)?

5. Although Algorithm 2 just used fixed constant, $c\_1 = 6$, why it should be defined as a variable? It seems not necessary.

**Limitations:**

I follow the whole proofs up to Appendix C.

### Minor comments
1. Step 1 in p. 16: in Algorithm 1, $\bar{w}_1$ is not defined since some necessary terms like $\eta_0$ and $\hat{\ell}_0$ are not defined. Therefore, you should define $\bar{w}_1 = w_1$ first.
Also, the last inequality of step 1 holds only when $K \leq T$. This is a very minor one, but it should be specified in somewhere in Lemma.

2. Line 629: we first some -> we first fix some?

3. Line 631: why $w\_{s,i}$ is non-decreasing? It would be true only if $i$ is activated.

4. unfinished log term appears in (48).

5. It would be better to specify that Lemma 8 is applied to derive the second inequality in (50).

6. It would be better to specify that the result of (62) is applied to derive the third inequality in (64).

---

> ### Author Rebuttal · Authors · 2024-08-05
>
> We first would like to thank Reviewer q85G for carefully reading our work and providing constructive comments, which significantly help us improve our work. We are also glad to further engage in interactive discussions with the reviewer.
>
> > Cmt 1: Why Eq. (55) holds
>
> Re: This is indeed a mis-calculation. After correction, Eq. (55) should be
>
> $$\\left| \\widehat{L}\_{t,i} - \\sum_{s=1}^t \\mu\_{s,i} \\right| \\leq \\text{Width}(t)\\cdot t/ \\tau\_i ,$$ where basically the original $\text{Width}(\tau_i)$ is now replaced with $ \text{Width}(t)$.
> Eq. (51) now implies that
> $$ \\left|\\hat{L}\_{t,i} - \\sum\_{s=1}^t\mu\_{s,i}\\right| \\leq  \\mathbb{I}\\{i\\in A\_t\\} \text{Width}(t) +\mathbb{I}\\{i\notin A_t\\} \text{Width}(t)\frac{t}{\tau_i}. $$
> In other words, before the deactivation ($t\leq \tau_i$), the tightness of our control on $\\left|\\widehat{L}\_{t,i} - \sum_{s=1}^t \mu_{s,i}\\right|$ is unchanged, and after that, it becomes looser (with $\\text{Width}(\\tau\_i)$ replaced by $\text{Width}(t)$).
>
> While this correction leads to changes in tests (6)-(8), we justify that we will still get the claimed regret guarantee (albeit an extra log factor in the adversarial regime). The **key intuition** is that, to fix this gap, we are replacing $\text{Width}(\tau_i)$ with $\text{Width}(t)$ in some steps, which is the same thing we did in the last step of Eq. (75) in the current proof. In other words, these $\text{Width}(\tau_i)$ in the analysis eventually becomes $\text{Width}(t_{\text{sw}})$, sooner or later.
>
> We provide the complete roadmap below.
>
> **A. New tests (6)-(8)**
>
> The corrected tests (6)-(8) should be like:
>
> $$\\text{test (6):} \\exists i\\in[K] \\text{\~such that\~}\\left| \\frac{\\widehat{L}\_{t,i}}{t} - \\widehat{\\mu}\_{t,i} \\right|> 9u\\left(\\log(\\beta/\\zeta)/ N\_{t,i} \\right)^{\\frac{v}{1+v}} +  \\mathbb{I}\\{i\\in A\_t\\}\\frac{\\text{Width}(t)}{t} +  \\mathbb{I}\\{i\\notin A\_t\\}\\frac{\\text{Width}(t)}{\\tau\_i},$$
> $$\\text{test (7):} \\exists i\\notin A\_t \text{\~such that\~}  (\\hat{L}\_{t,i} - \\min\_{j\\in A_t} \\hat{L}\_{t,j})/t > (c_1 + 4)\\text{Width}(t)/(\\tau_i -1),$$
> $$\\text{test (8):} \\exists i\\notin A\_t \\text{\~such that\~}  (\\hat{L}\_{t,i} - \\min\_{j\\in A_t} \hat{L}_{t,j})/t \leq (c_1 - 4)\text{Width}(t)/\tau_i,$$
>
> **B. Stochastic regime**
>
> After correcting Eq. (55) and tests (6)-(8), the bound in stochastic regime is unchanged, following the three steps in Sec. 4.2.1 (lines 274-280). That is, we are still able to show that (modified) tests (6)-(8) never fail. Then, steps 2 and 3 are not affected (step 2 is only about the round up to $\tau_i$, not after, and step 3 is only about the sampling strategy), so we get the same bound.
>
> **C. Adversarial regime**
>
> The key is whether the modified tests will impact the analysis and the guarantee in the adversarial regime, which we will show below is not the case.
>
> **C.1. Regret decomposition**
>
> To get the regret decomposition in Eq. (13), it is crucial to show that Eq. (66) is still true. With the modified tests, we can still show that (again with $\text{Width}(\tau_i-1)$ replaced with $\text{Width}(t_{\text{sw}}-1)$):
> $$\sum_{s=1}^{t_{\text{sw}}-1} \mu_{s,i} - \sum_{s=1}^{t_{\text{sw}}-1} \mu_{s,I^*_{t_{\text{sw}}-1}} > (c_1-6) \frac{t_{\text{sw}}-1}{\tau_i-1} \text{Width}(t_{\text{sw}}-1) \geq 0.$$
> Therefore, the regret decomposition in Eq. (13) still holds. The next step is to bound parts A, B, and C therein.
>
> **C.2. Parts A and B**
>
> With the modified tests, now we have
> $\text{part A} = O\left(\left(\frac{\log(\beta/\zeta)}{N_{t_{\text{sw}}-1,i}}\right)^{\frac{v}{1+v}} + \frac{\text{Width}(t_{\text{sw}}-1)}{\tau_i-1} \right),$
> and
> $\text{part B} = O\left(\text{Width}(t_{\text{sw}}-1)/(\tau_i -1) \right).$
> Both of them have the original $\text{Width}(\tau_i-1)$ replaced with $\text{Width}(t_{\text{sw}}-1)$.
>
> **C.3. Part C**
>
> Since part C is related to action $i^*_{t_{\text{sw}}-1}$ only, which is active in round $t_{\text{sw}}-1$ as shown in C.1 above, it is not affected, and we still have
> $$\text{part C} = O\left(\text{Width}(t_{\text{sw}}-1)\right)$$
> **C.4. Final calculation**
>
> Combing three terms, now Eq. (75) becomes
> $$\\sum\_{s=1}^{t\_{\\text{sw}}-1}\\mu\_{s,a\_t} - \\sum\_{s=1}^{t\_{\\text{sw}}-1}\\mu\_{s,i^*} = O\\left(\\sum\_{i=1}^K N\_{t\_{\\text{sw}}-1,i} \\cdot u\\left( \\frac{\\log(\\beta/\\zeta)}{N\_{t\_{\\text{sw}}-1,i}} \\right)^{\\frac{v}{1+v}}   \\right)+ O\\left( \\underbrace{\\sum\_{i=1}^K N\_{t\_{\\text{sw}}-1,i}\\frac{\\text{Width}(t\_{\\text{sw}}-1)}{\\tau_i -1}}\_{\\text{* term}} \\right) + O\\left( \\text{Width}(t\_{\\text{sw}}-1) \\right).$$
> The first and third term are the same as original ones (and can be well bounded), and in the second term (* term), the previous $\text{Width}(\tau_i-1)$ becomes $\text{Width}(t_{\text{sw}}-1)$, which will be our focus below.
>
> Applying Lemma 6 to $N\_{t\_{\\text{sw}}-1,i}$ and expanding $\text{width}(\cdot)$, the * term is bounded by
>
> $$O\\left( \\sum_{i=1}^K\\left( q_i \\tau_i(1+\\log T)\\right)\\frac{u K^{\\frac{v}{1+v}} (t_{\\text{sw}}-1)^{\frac{1}{1+v}} \\left(\\log(\\beta/\\zeta) \\right)^{\\frac{3v}{3v+1}}}{\\tau_i -1}  + \\sum_{i=1}^K\\log(\\beta/\\zeta)\\frac{u K^{\\frac{v}{1+v}} (t_{\\text{sw}}-1)^{\\frac{1}{1+v}} \\left(\\log(\\beta/\\zeta)\\right)^{\\frac{3v}{3v+1}}}{\\tau_i -1}\\right).$$
>
> The first part is not new and is bounded using $\\sum_{i=1}^K q_i = O(\\log K)$, and the second part (which used to be a lower-order term) is also under control since $\\tau_i \\geq K+1$ due to the initialization in Algorithm 2.
>
> Finally, now the bound becomes
>
> $$\\sum_{s=1}^{t_{\\text{sw}}-1} \\mu_{s,a_t} - \\sum_{s=1}^{t_{\\text{sw}}-1} \\mu_{s,i^*} = O\\left( (\\log K) (\\log T) u K^{\frac{v}{1+v}} (t_{\\text{sw}}-1)^{\\frac{1}{1+v}} \\left(\\log(\\beta/\\zeta)\\right)^{1 + \\frac{3v}{3v+1}} \\right).$$
>
> After correction, it has an extra $ \\left( \\log(\\beta/\\zeta) \\right)$ factor, which is coming from the second part of the * term.

---

> ### Author Response · Authors · 2024-08-05
> **Response to Reviewer q85G (2/3)**
>
> > Cmt 2: Comparision with Huang et al. (2022) and Genalti et al. (2024)
>
> Re: We agree that: these two assumptions (i.e., 1) truncated non-negative losses and 2) the knowledge of $u,v$) are in general incomparable to say which one is weaker. However, we would like to clarify that the setup we study in this work is to achieve the **BOBW** regret guarantee with the knowledge of $u,v$. Under this setup, [1] achieved the optimal BOBW guarantee with the additional truncated non-negative loss assumption, and the goal in our work is to still achieve the BOBW guarantee, while removing this assumption.
>
> While there are $(u,v)$-adaptive algorithms proposed in [1, 2], they handle **one single regime only** (adversarial regime in [1] and stochastic regime [2]) and even require the truncated non-negative loss assumption. In other words, there is no result achieving $(u,v)$-adaptive BOBW guarantee. Therefore, these adaptive results are not comparable to ours.
>
> > Cmt 3: Proof of Lemma 2 - step 2 (p. 17)
>
> Re: Thanks for catching this. This is indeed a gap in our analysis, as $\\bar{w}_t$, obtained from the OMD update, is not controlled by $\\rho_t$.
>
> We propose a fix to this, which is an update rule change in Line 7 of our Algorithm 1. That is, instead of first performing OMD update over the entire probability simplex to get $\\bar{w}_t$ and then getting $w_t$, we directly perform OMD update over the **truncated** simplex $\\Omega’$ as in the original paper [3], that is
>
> $$w\_{t+1} = \\text{argmin}\_{x\\in \\Omega'} \\left( \\langle x,\\hat{\\ell}\_t \\rangle + D\_{\\phi\_t} (x, w_t)\\right),$$
> where $\\Omega’:=\\{x\\in \\Omega: x(i) \\geq \\lambda/K, \\forall i\\in [K]\\}.$
>
> By doing this, the $\bar{w}\_t$ in the the analysis will become $w\_t$ (which is controlled by $\rho\_t$) and the entire proof will become the same as that of [3], where there’s no longer mismatching between $w\_t$ and $\bar{w}\_t$, and the additional regret due to truncating the simplex in the update can be bounded as in our Lemma 3 (line 588), given that $\lambda$ is small enough. One potential disadvantage of doing so is that it’s unclear to us how to perform the efficient implementation (as mentioned in line 219) when the OMD update is over the **truncated** simplex $\Omega’$ (which was also our motivation to modify the original update rule in [3]).
>
> > Cmt 4: Proof for adversarial setting (Alg. 2)
>
> Re: Thanks for catching this typo. The $\leq$ in Eq. (6) (Line 12 in Algorithm 2) should be $>$.
>
> > Cmt 5: What is $\\widehat{L}_{t\_{\\text{sw}-1}}'$?
>
> Re: It should be $\\widehat{L}\_{t\_{\\text{sw}-1}}$, the cumulative IW estimate.
>
> > Cmt 6: The bounds do not match between Eq. (64) and Theorem 2.
>
> Re: Thanks for catching this. The correct bound should follow Eq. (64) from the proof, which has an additional logarithmic factor.
>
> > Cmt 7: $\\hat{y}_t$ in (2)
>
> Re: It should be $\hat{\ell}_t$, the loss estimate sequence.
>
> > Cmt 8: $t_{\text{quit}}$ in (13)
>
> Re: $t_{\text{quit}}$ was meant to be $t_{\text{sw}}$.
>
> > Cmt 9: Why Eq. (59) holds for all actions in $[K]$
>
> Re: According to lines 657-658, for any action $i$ deactivated at round $\\tau\_i$, test (5) must fail in round $\\tau\_i-1$ (and is satisfied in round $\\tau\_i$). To derive Eq. (59), we are looking at round $\tau_i-1$ for each action $i$, not round $t$. We will improve the writing of this part.
>
> > Cmt 10: $c\_1=6$ seems unnecessary
>
> Re: Our intention was to separate free constants (namely, $c\_1\\geq 6$ which can be freely chosen) from the others (some “2” coming from the proof). As a revision in Alg. 2, we will write $c\_1 \\geq 6$ in the **"Input:"** rather than $c\_1 = 6$ in the **"Define:"**
>
> > Cmt 11: Line 629: first some -> first fix some?
>
> Re: Yes, thanks for spotting this typo.
>
> > Cmt 12: Line 631: why $w\_{s,i}$ is non-decreasing?
>
> Re: Correct. $w\_{s,i}$ is non-decreasing only up to round $s =\\tau\_i$. We will refine this statement. The proof after it does not rely on increasing $w\_{s,i}$ after round $\tau\_i$.
>
> > Cmt 13: unfinished log term appears in (48).
>
> Re: Thanks for catching this. It was meant to be $\log(2T/\zeta)$ according to the last line in Page 22.
>
> > Cmt 14: It would be better to specify that Lemma 8 (resp. (62)) is applied to derive (50) (resp. (64)).
>
> Re: Thanks for the suggestion. We will expand the explanations of key steps in our proof (including these two and more) to improve the readability.
>
> > Cmt 15: Some notations are not well-defined in Algorithm 1. $K\leq T$ is not specified.
>
> Re: Thanks for the suggestions. Yes, initializing $\\bar{w}\_1$ is necessary in the current Algorithm 1, and we assume that $K\\leq T$. We will explicitly specify this assumption when we introduce the problem setup.

---

> > ### Comment · Reviewer_q85G · 2024-08-09
> >
> > First, I would like to express my appreciation to the authors for their detailed explanations and modifications to the proofs.
> >
> > Overall, the revised algorithms and their proofs appear correct. Although the introduction of an additional logarithmic factor may seem concerning, it is acceptable within the context of the original definition of BOBW, where such a factor is also allowed.
> >
> > My only remaining concern is the computational efficiency of calculating the refined $w\_{t+1}$ which now includes the new constraint $w\_{t+1,i} \geq \lambda /K$ due to truncated simplex.

---

> ### Author Response · Authors · 2024-08-05
> **Response to Reviewer q85G (3/3)**
>
> > Cmt 16: Explain more on (63).
>
> Re: According to Line 14 of Algorithm 2, the probability of pulling a deactivated arm keeps decaying since its deactivation, and all active arms equally share the remaining probability mass. For the first action to be deactivated (denoted by $1'$), $q_{1'}$ is exactly $1/K$. For the second one $(2')$, we clearly have $q_{2'}\leq \frac{1}{K-1}$. So in general, we have $q_{i'}\leq \frac{1}{K-i'+1}$, and
> $$\\sum\_{i=1}^K q\_i = \\sum\_{i'=1}^K q\_{i'} \\leq \\sum\_{i'=1}^K \\frac{1}{K-i'+1}.$$
> We will attach the explanation to it.
>
> Referrences
>
> [1] Huang, Jiatai, Yan Dai, and Longbo Huang. "Adaptive best-of-both-worlds algorithm for heavy-tailed multi-armed bandits." international conference on machine learning. PMLR, 2022. https://proceedings.mlr.press/v162/huang22c.html
>
> [2] Genalti, Gianmarco, Lupo Marsigli, Nicola Gatti, and Alberto Maria Metelli. "$(ε, u) $-Adaptive Regret Minimization in Heavy-Tailed Bandits." In The Thirty Seventh Annual Conference on Learning Theory, pp. 1882-1915. PMLR, 2024. https://proceedings.mlr.press/v247/genalti24a.html
>
> [3] Lee, Chung-Wei, Haipeng Luo, Chen-Yu Wei, and Mengxiao Zhang. "Bias no more: high-probability data-dependent regret bounds for adversarial bandits and mdps." Advances in neural information processing systems 33 (2020): 15522-15533. https://proceedings.neurips.cc/paper_files/paper/2020/hash/b2ea5e977c5fc1ccfa74171a9723dd61-Abstract.html

---

> ### Author Response · Authors · 2024-08-10
> **Regarding the computational efficiency**
>
> We are glad to see that our responses successfully addressed the concerns and questions raised in the initial review, and we thank the reviewer again for carefully reading our submission (as well as the rebuttal) and providing valuable feedback. Please feel free to let us know if there are any other questions/comments.
>
> Regarding the computational efficiency of the new update rule over the truncated simplex (i.e., the one proposed in [3]), we would like to clarify that it can still be solved efficiently (i.e., in polynomial time): since both the objective function and the domain (namely, the truncated simplex) are convex, the update is (still) a convex optimization problem, for which there are well-developed solvers.
>
> What we meant in the initial response to Comment 3 is that, when the simplex is not truncated, there is an even more efficient way to perform the OMD update (which is to find the Lagrangian multiplier corresponding to the equality constraint via line search), which (to our understanding) is no longer applicable due to the additional constraints/multipliers introduced by the simplex truncation. We didn't mean that updating over the truncated simplex becomes "computationally inefficient" in the general sense.
>
> We hope this addresses your concern.

---

### Official Review · Reviewer_LBNn · 2024-07-13

**Soundness:** 3
**Presentation:** 3
**Contribution:** 3
**Rating:** 7
**Confidence:** 2

**Summary:**

This paper considers the bandit problem for heavy-tailed losses and proposes an algorithm that achieves a nearly tight high-probability regret bounds.
The proposed algorithm has a best-of-both-worlds guarantee, i.e., it achieves nearly tight bounds in both adversarial and stochastic settings.
The proposed approach is also shown to be useful in terms of local differential privacy.

**Strengths:**

- This study bypasses the assumption of truncated non-negative losses, which was required in the previous study by Huang et al. (2022).
- This paper shows high-probability regret bounds, which is rare in the contexts of heavy-tailed bandits and best-of-both-worlds algorithms.
- Obtained regret upper bounds are almost tight.

**Weaknesses:**

- The proposed algorithm requires prior knowledge of the parameters $u$ and $v$. This is a weakness when compared to algorithms that are adaptive to these parameters, e.g., by Huang et al.(2022) and  Genalti et al. (2024).
- There is a lack of mention or analysis of intermediate settings between stochastic and adversarial settings, e.g., corrupted environments.
A number of best-of-both-worlds algorithms are also effective in these settings (e.g., (Lee et al., 2020), (Zimmert and Seldin, 2021), (Dann et al., 2023)).

**Questions:**

How tight is the bound of the proposed algorithm in terms of its dependence on $\log T$?
Are there any known comparable lower bound?

**Limitations:**

The limitations and potential negative societal impact are adequately addressed.

---

> ### Author Rebuttal · Authors · 2024-08-05
>
> > Comment 1: The proposed algorithm requires prior knowledge of the parameters $u$ and $v$. This is a weakness when compared to algorithms that are adaptive to these parameters, e.g., by Huang et al.(2022) and Genalti et al. (2024).
>
> Re: While there are $(u,v)$-adaptive algorithms proposed in [1, 2], they handle **one single regime only** (adversarial regime in [1] and stochastic regime [2]) and hence are not directly comparable to our BOBW result. Moreover, these adaptive algorithms require the truncated non-negative loss assumption.
>
> The setup we consider in this work is the BOBW setting with the knowledge of $u$ and $v$. Under this setup, the **only** existing result is from [1], which is optimal in both regimes and requires the truncated non-negative loss assumption. Compared to that, we still achieve BOBW guarantee but try to get rid of the additional assumption.
>
> > Comment 2: There is a lack of mention or analysis of intermediate settings between stochastic and adversarial settings, e.g., corrupted environments. A number of best-of-both-worlds algorithms are also effective in these settings (e.g., (Lee et al., 2020), (Zimmert and Seldin, 2021), (Dann et al., 2023)).
>
> Re: That is a good point. Online Learning-based algorithms naturally ensure regret guarantee in the corrupted regime through advanced and elegant analysis, and there is no existing result on achieving regret guarantee in the corrupted regime using the detect-switch framework. It is unclear whether/how the detect-switch framework can do that (by, e.g., explicitly detecting the degree of corruption $C\in[0, \Theta(T)]$). This is indeed an informative remark we may consider adding to future versions.
>
> > Comment 3: How tight is the bound of the proposed algorithm in terms of its dependence on $\log T$? Are there any known comparable lower bounds?
>
> Re: There are two main sources of $\log T$ factors in our work, including 1) the log-barrier regularizer and 2) high-probability guarantees (from concentrations). For the latter one, we mean that the $\log T$ factor is inevitable for a high-probability guarantee, even considering the adversarial regime only, as shown in [3].
>
> In terms of in-expectation regret, the worst-case lower bound in the adversarial regime is $\Omega(u K^{\frac{v}{1+v}} T^{\frac{1}{1+v}})$, and the gap-dependent lower bound in the stochastic regime is $\Omega(\sum_{i\neq i^*}\frac{\log T}{(\Delta_i)^{1/v}})$ as we mentioned in the paper.
>
> In terms of high-probability regret, when heavy tails are involved, (to our knowledge) there is no regret lower bound showing the refined dependency on $\log T$.
>
> References
>
> [1] Huang, Jiatai, Yan Dai, and Longbo Huang. "Adaptive best-of-both-worlds algorithm for heavy-tailed multi-armed bandits." international conference on machine learning. PMLR, 2022. https://proceedings.mlr.press/v162/huang22c.html
>
> [2] Genalti, Gianmarco, Lupo Marsigli, Nicola Gatti, and Alberto Maria Metelli. "$(ε, u) $-Adaptive Regret Minimization in Heavy-Tailed Bandits." In The Thirty Seventh Annual Conference on Learning Theory, pp. 1882-1915. PMLR, 2024. https://proceedings.mlr.press/v247/genalti24a.html
>
> [3] Gerchinovitz, Sébastien, and Tor Lattimore. "Refined lower bounds for adversarial bandits." Advances in Neural Information Processing Systems 29 (2016). https://proceedings.neurips.cc/paper_files/paper/2016/hash/2f37d10131f2a483a8dd005b3d14b0d9-Abstract.html

---

### Official Review · Reviewer_t2SX · 2024-07-16

**Soundness:** 3
**Presentation:** 3
**Contribution:** 3
**Rating:** 6
**Confidence:** 3

**Summary:**

This work studied the adversarial bandit problem with heavy-tailed distribution. It also studied the best-of-both-world setting. It relaxed the non-negative assumption and analyzed near-optimal algorithms.



===================

I would like to keep the score after reading the rebuttal.

**Strengths:**

1. This work discussed the earlier literature in detail.
1. It highlighted the technical challenges.
1. Overall, I think this work provided a good set of results and explained them well.

**Weaknesses:**

1. Is it possible to bound the expected regret of the proposed algorithm? What is the limitation?
1. As mentioned in Section 2, there are two possible definitions of regret in the adversarial setting. The author(s) should clarify which definition is used in the related works.
1. A table comparing all results is appreciated.

**Questions:**

Please refer to the 'Weaknesses' part.

---

> ### Author Rebuttal · Authors · 2024-08-05
>
> > Comment 1: Is it possible to bound the expected regret of the proposed algorithm? What is the limitation?
>
> Re: We believe that the reviewer is asking whether it is possible to bound the stronger regret $\mathbb{E}[\bar{R}_T]$ defined in line 147 in the adversarial regime. We now state the challenge of that in heavy-tailed bandits.
>
> By definition, we would like to bound
> $$\\mathbb{E}[\\bar{R}\_T] = \\mathbb{E}[ \\mathbb{E}[\\sum\_{t=1}^Y \\langle w\_t - y\_{i^*}, \\ell\_t \\rangle]|\\ell\_1,\\dots,\\ell\_T].$$
>
> Note that here $i^*:=\\text{argmin}\_i \\sum\_{i=1}^T \\ell\_{t,i}$, which depends on the realization of loss sequence. Looking at the inner expectation, the quantity $\sum_{t=1}^Y \langle w_t - y_{i^*}, \ell_t \rangle$ depends on not only the policy ($w_t$), **but also** the realization (the scale of losses). In other words, as long as the losses have large scale, the absolute quantity of regret is also large, even if the learning algorithm is good.
>
> Therefore, one potential solution is to bound $\mathbb{E}[\bar{R}_T]$ hopefully thanks to the outer-level expectation (i.e., the realizations cannot be always very large due to the heavy tail definition). It is unclear to us whether this can be well bounded. Notably, this was also our intuition for why it is reasonable to consider pseudo-regret in heavy-tailed bandits: strong regret could be no longer meaningful as a performance metric.
>
> > Comment 2: As mentioned in Section 2, there are two possible definitions of regret in the adversarial setting. The author(s) should clarify which definition is used in the related works.
>
> Re: Thanks for the suggestion. We will clarify this in future versions. Roughly speaking, in the heavy-tailed case, both [1] and us consider pseudo-regret (as we explained in our Remark 2), and in the bounded-loss case, the stronger regret is typically considered and can be handled.
>
> > Comment 3: A table comparing all results is appreciated.
>
> Re: We thank the reviewer for the suggestion. We will consider adding a table to summarize the existing results for a clear comparison between our work and previous ones.
>
> References
>
> [1] Huang, Jiatai, Yan Dai, and Longbo Huang. "Adaptive best-of-both-worlds algorithm for heavy-tailed multi-armed bandits." international conference on machine learning. PMLR, 2022. https://proceedings.mlr.press/v162/huang22c.html

---

> > ### Comment · Reviewer_t2SX · 2024-08-12
> >
> > Thanks for your response. I would like to keep the score.

---

### Official Review · Reviewer_FpXy · 2024-07-16

**Soundness:** 3
**Presentation:** 3
**Contribution:** 3
**Rating:** 7
**Confidence:** 3

**Summary:**

In this work, the authors consider a best-of-both-worlds multi-armed bandits problem where the losses are not bounded and instead are heavy-tailed.
To be precise, in the stochastic setting, losses are generated from fixed distributions.
In the oblivious adversarial setting, the losses are drawn from distributions that are generated arbitrarily and can change from one round to the next.
In both cases, these distributions can sample heavy-tailed losses, which are defined such that the $1 + v$^{th} moment of the losses are bounded by $u^{1 + v}$ for some constants $u > 0$ and $v \in (0, 1]$.
This setting is more challenging than the standard BOBW approach because of the generation process of the losses, which can lead to negative unbounded losses but also because the performance is evaluated in terms of expected regret rather than in terms of pseudo-regret, which is a more challenging measure.

The authors propose and analyze an algorithm based on the FTRL framework. They discuss that simply using regularization with Tsallis entropy (which is the state of the art for the standard BOBW MAB problem) cannot handle large negative losses and instead rely on the log barrier regularizer, which is a standard approach to handle problems where the stability is difficult to bound.
While this approach itself is sufficient to derive bounds in the adversarial regime, achieving BOBW bounds requires supplementary tricks:

They then use a detect-switch strategy to monitor whether the environment is stochastic, in which case the arms are sampled at a rate that ensures stochastic guarantees, and otherwise make a definitive switch to the adversarial regime and use the previously discussed FTRL with log-barrier regularization.

**Strengths:**

The authors tackle a challenging problem of BOBW bandits with heavy tail losses and propose a solution that achieves near-optimal results in both the adversarial and the stochastic regime.

The proposed methods combine well-studied methods in the BOBW literature and this paper highlights another topic of interest for the FTRL with log barrier framework.
The authors provide a detailed analysis of their methods and provide a very detailed explanation of their choices of methods.

**Weaknesses:**

While the presented results are novel and interesting, the proposed method is suboptimal by several logarithmic factors both in the stochastic and in the adversarial framework. Both the detect-switch method and FTRL with log barrier are known to be suboptimal in the standard BOBW MAB problem with bounded losses, which means that improving upon the existing results might require a completely different approach.

**Questions:**

Do you think that the detect-switch framework is necessary to achieve these BOBW methods or whether some more straightforward method (like FTRL with 1/2 Tsallis-Inf for the standard BOBW MAB with bounded losses)?

**Limitations:**

This work is purely theoretical and the limitations of the applicability of the results are properly detailed in the conditions of each theorem.

---

> ### Author Rebuttal · Authors · 2024-08-05
>
> > Comment 1: Both the detect-switch method and FTRL with log barrier are known to be suboptimal in the standard BOBW MAB problem with bounded losses, which means that improving upon the existing results might require a completely different approach.
>
> Re: We agree that both log-barrier (for adversarial bandits) and detect-switch method (for BOBW) do not achieve the optimal regret (i.e., suffering extra $\\log T$ factors) even with bounded losses, although to our knowledge, currently they are the only known approaches that are promising to handle heavy-tailed bandits in BOBW (and previous approaches need additional assumptions), and how to narrow the gap towards the optimal regret is still largely open, for which a totally different algorithm design seems to be necessary.
>
> > Comment 2: Do you think that the detect-switch framework is necessary to achieve these BOBW methods or whether some more straightforward method (like FTRL with 1/2 Tsallis-Inf for the standard BOBW MAB with bounded losses)?
>
> Re: As we discussed in the last paragraph in the main body, it is largely unknown whether purely online algorithms (e.g., FTRL) alone can achieve BOBW regrets in heavy-tailed bandits. Even considering the adversarial regime only, log-barrier seems to be necessary, so one promising direction is to show that OMD/FTRL alone (without detect-switch framework) with log-barrier can achieve BOBW guarantee, although involved theoretical analysis is expected.

---

> > ### Comment · Reviewer_FpXy · 2024-08-13
> >
> > Thank you for your comments, particularly the detailed discussion you had with reviewer q85G. I don't have any further questions at this point.

---

### Decision · Program_Chairs · 2024-09-25

**Decision:**

Accept (poster)

**Comment:**

This paper tackles the best-of-both-world problem for multi-armed bandits with heavy-tailed losses. This work removes the restrictive assumption made in Huang et al. (2022).  Although there are concerns about the sub-optimality of the regret bounds, the consensus is that the ideas developed in this paper are valuable and interesting.